# Evaluating the Community Land Model (CLM 4.5) at a Coniferous Forest Site in Northwestern United States Using Flux and Carbon-Isotope Measurements

Henrique F. Duarte[1], Brett M. Raczka[2], Daniel M. Ricciuto[3], John C. Lin[1], Charles D. Koven[4], Peter E. Thornton[3], David R. Bowling[2], Chun-Ta Lai[5], Kenneth J. Bible[6], James R. Ehleringer[2]

[1]Department of Atmospheric Sciences, University of Utah, Salt Lake City, 84112, USA
[2]Department of Biology, University of Utah, Salt Lake City, 84112, USA
[3]Oak Ridge National Laboratory, Oak Ridge, 37831, USA
[4]Lawrence Berkeley National Laboratory, Berkeley, 94720, USA
[5]Department of Biology, San Diego State University, San Diego, 92182, USA
[6]United States Forest Service, Pacific Northwest Research Station, Corvallis, 97331, USA

*Correspondence to*: Henrique F. Duarte (h.duarte@utah.edu)

**Abstract.** Droughts in the western United States are expected to intensify with climate change. Thus, an adequate representation of ecosystem response to water stress in land models is critical for predicting carbon dynamics. The goal of this study was to evaluate the performance of the Community Land Model, Version 4.5 (CLM) against observations at an old-growth coniferous forest site in the Pacific Northwest region of the United States (Wind River AmeriFlux site), characterized by a Mediterranean climate that subjects trees to water stress each summer. CLM was driven by site-observed meteorology and calibrated primarily using parameter values observed at the site or at similar stands in the region. Key model adjustments included parameters controlling specific leaf area and stomatal conductance. Default values of these parameters led to significant underestimation of gross primary production, overestimation of evapotranspiration, and consequently overestimation of photosynthetic $^{13}C$ discrimination, reflected on reduced $^{13}C$:$^{12}C$ ratios of carbon fluxes and pools. Adjustments in soil hydraulic parameters within CLM were also critical, preventing significant underestimation of soil water content and unrealistic soil moisture stress during summer. After calibration, CLM was able to simulate energy and carbon fluxes, leaf area index, biomass stocks, and carbon isotope ratios of carbon fluxes and pools in reasonable agreement with site observations. Overall, the calibrated CLM was able to simulate the observed response of canopy conductance to atmospheric vapor pressure deficit (VPD) and soil water content, reasonably capturing the impact of water stress on ecosystem functioning. Both simulations and observations indicate that stomatal response from water stress at Wind River was primarily driven by VPD and not soil moisture. The calibration of the Ball-Berry stomatal conductance slope ($m_{bb}$) at Wind River aligned with findings from recent CLM experiments at sites characterized by the same plant functional type (needleleaf evergreen temperate forest), despite significant differences in stand composition/age and climatology, suggesting that CLM could benefit from a revised $m_{bb}$ value of 6, rather than the default value of 9, for this plant functional type. On the other hand, Wind River required a unique calibration of the hydrology submodel to simulate soil moisture, suggesting the

default hydrology has a more limited applicability. This study demonstrates that carbon isotope data can be used to constrain stomatal conductance and intrinsic water use efficiency in CLM, as an alternative to eddy covariance flux measurements. It also demonstrates that carbon isotopes can expose structural weaknesses in the model and provide a key constraint that may guide future model development.

## 1 Introduction

The frequency, duration, and severity of droughts are expected to increase in the 21$^{st}$ century with climate change (Burke et al., 2006; Sheffield and Wood, 2008; Dai, 2013; Prein et al., 2016). In the western United States in particular, the combination of warmer temperature, larger vapor pressure deficit, reduced snowfall and snow pack, earlier snow melt, and extended growing season length is expected to lead to an intensification of water stress during the summer (Boisvenue and Running, 2010; Spies et al., 2010; Swain and Hayhoe, 2015; Fyfe et al., 2017). In this drying scenario, an accurate representation of ecosystem response to water stress in land models is critical for projecting carbon dynamics (and climate) into the future.

The land carbon and water cycles are coupled by the plant stomata through $CO_2$ uptake (photosynthesis) and water vapor loss (transpiration). While stomatal conductance responds to atmospheric vapor pressure deficit, soil moisture, and various other environmental factors, its modeling still represents a major challenge for the scientific community (Damour et al., 2010). Many stomatal conductance models have been proposed, including different approaches to account for water stress, but each model is subject to its own limitations (Damour et al., 2010; Miner et al., 2017; Sperry et al., 2017). Traditionally, stomatal conductance models have been calibrated through leaf to canopy-level observations of water exchange.

Stable carbon isotopes provide an alternative observation to constrain stomatal conductance and offer an opportunity for model evaluation and improvement. During photosynthesis, plants discriminate against the heavier stable isotope of carbon ($^{13}$C) in favor of the lighter, more abundant $^{12}$C stable isotope. This discrimination in C$_3$ plants, expressed as $\Delta = \left[ \left( R_{air}/R_{plant} \right) - 1 \right] \times 1000$ (‰), where $R_{air}$ and $R_{plant}$ are the $^{13}$C:$^{12}$C isotope ratios of atmospheric $CO_2$ and plant assimilated carbon, respectively, can be estimated according to the model proposed by Farquhar and Richards (1984) as

$$\Delta = a + (b - a)c_i/c_a , \tag{1}$$

where $c_i/c_a$ is the ratio of intracellular $CO_2$ concentration to atmospheric $CO_2$ concentration, $a$ is the $^{13}$C discrimination associated with the process of $CO_2$ diffusion through the stomata, and $b$ is the $^{13}$C discrimination associated with the process of assimilation of $CO_2$ via Rubisco ($a \approx 4.4$‰ and $b \approx 27$‰; Farquhar et al., 1989). The $c_i/c_a$ ratio correlates negatively with leaf intrinsic water use efficiency (iWUE), defined as the ratio of net leaf assimilation to stomatal conductance (Farquhar et al., 1989). Under water stress, C$_3$ plants tend to reduce stomatal conductance and increase water use efficiency, leading to reductions in $c_i/c_a$ and $^{13}$C discrimination, affecting the carbon isotope ratio ($\delta^{13}$C) of photosynthesis and consequently of carbon pools and respiration. Experimental studies have shown, for instance, correlations between the $\delta^{13}$C

of ecosystem respiration and soil water content, atmospheric vapor pressure deficit, and precipitation. Bowling et al. (2008) and Brüggemann et al. (2011) present extensive reviews of experimental results on the link between environmental factors and the isotopic signature of carbon pools and fluxes, demonstrating that isotopic measurements provide insights into the response of stomatal conductance and iWUE to water stress. Furthermore, stable carbon isotopes have been used to partition

photosynthetic and respiration fluxes from flux tower data (e.g., Wehr and Saleska, 2015) and to identify the strength of land and ocean sinks (e.g., Alden et al., 2010; van der Velde et al., 2013).

Photosynthetic $^{13}$C discrimination is represented in biospheric models including the stable isotope-enabled Land Surface Model, ISOLSM (Riley et al., 2002), the Simple Biosphere Model, SiB (Suits et al., 2005), the Lund-Potsdam-Jena dynamic global vegetation model, LPJ (Scholze et al., 2003; 2008), the Land Surface Processes and Exchanges model of the

University of Bern, LPX-Bern (Spahni et al., 2013; Stocker et al., 2013), the hybrid SiB-CASA (combining biophysics from SiB and biochemistry from the Carnegie-Ames-Stanford Approach model) (van der Velde et al., 2013; 2014), and the Community Land Model, CLM (Oleson et al., 2013).

Modeling studies have shown that stable carbon isotopes provide a constraint upon stomatal conductance (Aranibar et al., 2006; Raczka et al., 2016; Mao et al., 2016). Aranibar et al. (2006) evaluated the performance of ISOLSM at the

Metolius Old Pine AmeriFlux site and were able to calibrate the slope of the stomatal conductance equation ($m_{bb}$ in the Ball-Berry stomatal conductance model; see Eq. 2) with the aid of foliar $\delta^{13}$C data measured at the site. Raczka et al. (2016) evaluated photosynthetic $^{13}$C discrimination in CLM version 4.5 (CLM4.5) against $\delta^{13}$C observations of photosynthesis and biomass at the Niwot Ridge AmeriFlux site and found the model to perform poorly with its default nitrogen limitation approach, resulting in overestimation of stomatal conductance and $^{13}$C discrimination. By using an alternative approach in

which a nitrogen downscaling factor is directly applied to $V_{cmax25}$ (maximum rate of carboxylation at 25$^o$C), they found significant improvement in the simulations, but with results still suggesting that a smaller $m_{bb}$ value (they used the default C$_3$ value, $m_{bb} = 9$) would better simulate the site observations. Mao et al. (2016) evaluated CLM4.0 at a loblolly pine site in Tennessee, USA and were able to adequately simulate the observed biomass $\delta^{13}$C values with an optimized $m_{bb}$ value of 5.6. Keller et al. (2017) used a global tree-ring $\delta^{13}$C dataset to evaluate the 20$^{th}$-century trend in photosynthetic $^{13}$C discrimination

and iWUE as modeled by CLM4.5 and LPX-Bern. LPX-Bern was found to perform well, while CLM simulated a significantly stronger increase (decrease) in iWUE ($^{13}$C discrimination) than that indicated by the tree-ring data. The default CLM parameterization and configuration were used in their study. Keller et al. (2017) suggested that the model-data mismatch was associated with the stomatal conductance parameterization ($m_{bb}$ in particular) and the shortcomings of the nitrogen limitation scheme.

The present study focuses on CLM —the land component of the Community Earth System Model (CESM), a fully-coupled global climate model widely used by the scientific community (http://www.cesm.ucar.edu/publications/)— and further evaluates the performance of its latest release (CLM4.5 – hereafter referred simply as "CLM") against observations at a coniferous forest site in the Pacific Northwest region of the United States, with particular attention to the simulation of stomatal conductance and its response to water stress. The study site, Wind River, is part of the AmeriFlux eddy covariance

network. Wind River is an old-growth forest (~500 years) characterized by a Mediterranean climate, due to which trees are naturally subject to water stress each summer. The combination of long-term measurements of energy/carbon fluxes, meteorology, biological variables, and stable carbon isotope ratios, makes the site a good choice for evaluating carbon cycle and carbon isotope components of CLM. In addition to energy flux observations which allow for the estimation of canopy conductance, this study leverages the recent inclusion of photosynthetic [13]C discrimination within CLM and also uses $\delta^{13}$C observations to better diagnose the simulation of stomatal conductance at the site. We test whether a reduced stomatal conductance at similar needleleaf evergreen temperate forest sites (Mao et al., 2016; Raczka et al., 2016) is appropriate for Wind River. This study also provides further investigation on the nitrogen limitation issue identified by Raczka et al. (2016) and the adequacy of the default parameters used in CLM, especially those regulating stomatal conductance. We test whether the calibration scheme (optimized parameters, nitrogen limitation) proposed by Raczka et al. (2016) for Niwot Ridge, is appropriate for Wind River. By comparing the results at Wind River against those at different sites characterized by the same plant functional type (needleleaf evergreen temperate tree) but with different stand composition/age and climatology (Mao et al., 2016; Raczka et al., 2016), this study also seeks to identify general improvements in model parameterization.

## 2 Material and Methods

This Section provides a description of CLM, focusing on key formulations of relevance to the present study (Sect. 2.1), followed by a description of the study site (Sect. 2.2), the eddy-covariance and meteorological data sets used to drive and assess the model (Sect. 2.3), the carbon isotope data sets used to assess the photosynthetic [13]C discrimination in CLM (Sect. 2.4), and also a description of the CLM configuration, simulations performed, and calibration of model parameters (Sects. 2.5 and 2.6). Section 2.7 describes the methodology used in the calculation of canopy conductance values from eddy-covariance observations, which are compared against simulated values as a way to assess the model skill in simulating leaf stomatal conductance.

### 2.1 Model Description

This Section focuses on describing CLM's approach to the simulation of stomatal conductance and photosynthetic [13]C discrimination, key aspects of this study. For a full description of the model, the reader is referred to Oleson et al. (2013).

In CLM, leaf stomatal conductance ($g_s$) is calculated based on the Ball-Berry model as described by Collatz et al. (1991) and implemented by Sellers et al. (1996) in the SiB2 model:

$$g_s = m_{bb} \frac{A_n(\beta_t)}{c_s / P_{atm}} h_s + b_{bb} \beta_t \tag{2}$$

where $A_n(\beta_t)$ is the potential net leaf photosynthesis (without nitrogen limitation) as a function of a soil moisture stress factor ($\beta_t$), $c_s$ is the $CO_2$ partial pressure at the leaf surface, $P_{atm}$ is the atmospheric pressure, $h_s$ is the relative humidity at the leaf surface (defined as the ratio of vapor pressure at the leaf surface to saturation vapor pressure inside the leaf at

vegetation temperature $T_v$), $m_{bb}$ is a slope coefficient, and $b_{bb}$ corresponds to the minimum stomatal conductance in the original Ball-Berry model. The soil moisture stress factor $\beta_t$ is defined as:

$$\beta_t = \sum_i w_i r_i \tag{3}$$

where $r_i$ is the root fraction at soil layer $i$ and $w_i$ is a corresponding plant wilting factor. The former is defined as (Oleson et al., 2013):

$$r_i = 0.5(e^{-r_a z_{h,i-1}} + e^{-r_b z_{h,i-1}}) - 0.5\alpha(e^{-r_a z_{h,i}} + e^{-r_b z_{h,i}}) \tag{4}$$

where $z_{h,i}$ (m) is the depth from the soil surface to the interface between layers $i$ and $i + 1$ ($z_{h,0} = 0$ corresponds to the soil surface), $r_a$ and $r_b$ are root distribution parameters (m$^{-1}$), $\alpha = 1$ for $1 \le i < N_{levsoi}$, and $\alpha = 0$ for $i = N_{levsoi}$ ($N_{levsoi}$ is the number of soil layers). The plant wilting factor for soil layer $i$ is defined as (Oleson et al., 2013):

$$w_i = \begin{cases} \frac{\Psi_c - \Psi_i}{\Psi_c - \Psi_o}\left[\frac{\theta_{sat,i} - \theta_{ice,i}}{\theta_{sat,i}}\right] \le 1 & \text{for } T_i > T_f - 2 \text{ and } \theta_{liq,i} > 0 \\ 0 & \text{for } T_i \le T_f - 2 \text{ or } \theta_{liq,i} = 0 \end{cases} \tag{5}$$

where $\Psi_i$ is the soil water matric potential, $\Psi_c$ and $\Psi_o$ are the soil water potential when stomata are fully closed or fully open, respectively ($\Psi_c = -255000$ mm and $\Psi_o = -66000$ mm for the needleleaf evergreen temperate tree plant functional type, hereafter referred simply as "NETT PFT"), $\theta_{sat,i}$ is the saturated volumetric water content, $\theta_{ice,i}$ is the volumetric ice content, $\theta_{liq,i}$ is the volumetric liquid water content, $T_i$ is the soil layer temperature, and $T_f = 273.15$ K is the freezing temperature of water. The sum in Eq. (3) is defined over the entire soil column, resulting in $\beta_t$ values from 0 (maximum soil moisture stress) to 1 (no soil moisture stress). In CLM's implementation of the Ball-Berry model (Eq. 2), $\beta_t$ is used to downscale $b_{bb}$, directly impacting $g_s$. $\beta_t$ also indirectly impacts $g_s$ through the $A_n$ term, as $\beta_t$ is used to downscale the maximum rate of carboxylation ($\beta_t V_{cmax}$) and also leaf respiration ($\beta_t R_d$) (Oleson et al., 2013).

Stomatal conductance ($g_s$) and $A_n$ are solved separately for sunlit and shaded leaves. Canopy conductance ($G_c$) is given by

$$G_c = \frac{1}{r_b + r_s^{sun}}\text{LAI}^{sun} + \frac{1}{r_b + r_s^{sha}}\text{LAI}^{sha} \tag{6}$$

and potential gross primary production (GPP$_{pot}$, without nitrogen limitation) by

$$\text{GPP}_{pot} = (A_n^{sun} + R_d^{sun})\text{LAI}^{sun} + (A_n^{sha} + R_d^{sha})\text{LAI}^{sha} \tag{7}$$

where $r_b$ is the leaf boundary layer resistance, $r_s = 1/g_s$ is the leaf stomatal resistance, LAI is the leaf area index, and $R_d$ is the leaf-level respiration ($sun$ and $sha$ superscripts denote sunlit and shaded leaves, respectively). Photosynthetic parameters such as $V_{cmax25}$ are solved separately for sunlit and shaded leaves and their canopy scaling scheme is detailed in Oleson et al. (2013, Sect. 8.3).

Based on nitrogen availability and nitrogen requirements for allocation of new carbon tissue, CLM calculates actual GPP as

$$\text{GPP} = \text{GPP}_{pot}(1 - d) \tag{8}$$

The nitrogen down-regulation factor ($d$) is defined as

$$d = \frac{CF_{avail\_alloc} - CF_{alloc}}{GPP_{pot}} \qquad (9)$$

where $CF_{avail\_alloc}$ is the carbon flux from photosynthesis which is available to new growth allocation and $CF_{alloc}$ is the actual carbon allocation to new growth (limited by nitrogen availability). This implementation of nitrogen down-regulation makes CLM a partially-coupled model with respect to net leaf photosynthesis and stomatal conductance. Note that actual plant gross carbon uptake (GPP) is calculated via down-regulation (Eq. 8) after the solution for $A_n$ and $g_s$ is obtained. Modeled $g_s$ remains consistent with $A_n$ (potential, not actual net leaf photosynthesis).

The original implementation of $^{13}$C in CLM was developed in consultation with Neil Suits (Suits et al., 2005) and is described in Oleson et al. (2013). Photosynthetic $^{13}$C discrimination in CLM for $C_3$ plants follows the model proposed by Farquhar and Richards (1984) (cf. Eq. 1):

$$\Delta = 4.4 + 22.6\, c_i/c_a. \qquad (10)$$

CLM calculates the intracellular-to-atmospheric $CO_2$ concentration ratio, $c_i/c_a$, in Eq. (10) as:

$$\frac{c_i}{c_a} = 1 - \frac{A_n(1-d)}{c_a}\left[\frac{1.4}{g_b} + \frac{1.6}{g_s}\right] \qquad (11)$$

where $g_b$ is the leaf boundary layer conductance. CLM does not account for mesophyll conductance (intracellular $CO_2$ is assumed to be the same as intercellular $CO_2$). Assuming $g_b \gg g_s$ (typically true for coniferous needles), Eq. (11) can be approximated by:

$$\frac{c_i}{c_a} \cong 1 - \frac{1.6(1-d)}{c_a}\left[\frac{A_n}{g_s}\right]$$

$$\cong 1 - \frac{1.6(1-d)}{c_a}\,\text{iWUE} \qquad (12)$$

where $\text{iWUE} = A_n/g_s$ is the intrinsic water use efficiency. Note that $c_i/c_a$ and consequently $\Delta$ correlate negatively with iWUE. All other terms being constant in Eq. (12), an increase in iWUE is expected to result in a reduction of the photosynthetic $^{13}$C discrimination, i.e., an increase in the assimilation of the heavier $^{13}$C stable isotope relative to the lighter, more abundant $^{12}$C stable isotope. Note also that $A_n$ is multiplied by $(1-d)$ in Eqs. (11) and (12), making $c_i$ consistent with the actual, nitrogen-limited GPP. However, it is important to highlight that $g_s$ is consistent with $A_n$ (potential net assimilation), not $A_n(1-d)$ (actual net assimilation). The implications of this mismatch to the simulation of $\Delta$ are discussed in Raczka et al. (2016) and later in the present paper.

The carbon isotope ratio of the GPP flux ($\delta^{13}C_{GPP}$) is calculated in CLM based on the prescribed $\delta^{13}$C of atmospheric $CO_2$, the carbon assimilation and photosynthetic $^{13}$C discrimination by sunlit and shaded leaves, and their respective leaf area indices. The $\delta^{13}$C of newly allocated carbon is the same as $\delta^{13}C_{GPP}$. The $\delta^{13}$C of the leaf carbon pool, for instance, depends on the allocation flux and its $\delta^{13}$C ($\delta^{13}C_{GPP}$) and the turnover time of the pool. CLM does not include any representation of post-photosynthetic $^{13}$C discrimination.

## 2.2 Site Description

The site for this study (Wind River) is part of the AmeriFlux eddy covariance network (Baldocchi et al., 2001) with a long record of meteorological, biological, surface flux (energy and carbon), and carbon isotope measurements for model assessment (1998–present). The site is located in the Pacific Northwest region of the United States, in the state of Washington (45.8205 Lat, −121.9519 Lon, 371-m elevation — see Fig. 1). Wind River is characterized by an old-growth conifer forest dominated by Douglas-fir (*Pseudotsuga menziesii*) and western hemlock (*Tsuga heterophylla*) trees, with a mean canopy height of 56 m. Douglas-fir trees are about 40–65-m high, corresponding to about 50% of the wood volume of the stand and 33% of the leaf area, while western hemlock trees are more numerous and smaller, corresponding to about 53% of the leaf area of the stand (Unsworth et al., 2004; Parker et al., 2004). No significant disturbances have occurred at the site in the past ~450–500 years. The local climate is strongly seasonal, marked by dry summers and wet winters. The climate summary reported by Shaw et al. (2004) indicates a mean annual precipitation of 2223 mm, with only ≈5% falling during June, July, and August. During winter, much of the precipitation falls as snow, and the average snowpack depth exceeds 100 mm. The mean annual, January, and July air temperatures are 8.7±6.5 $^{o}$C, 0.1±2.3 $^{o}$C, and 17.7±1.7 $^{o}$C, respectively.

## 2.3 Eddy-Covariance and Meteorological Data

Air temperature, relative humidity, wind speed, incident short-wave radiation, incident long-wave radiation, atmospheric pressure, and precipitation observed at the Wind River site from 1998 to 2006 were used to drive CLM. The time series were gap-filled using data from nearby towers and climate stations or interpolated in case of missing data. The gap-filled data product used to drive CLM in this study was created at Oak Ridge National Laboratory following the methodology described in Barr et al. (2013).

The L4 data set based on the eddy-covariance observations was downloaded from the AmeriFlux repository (version V002, daily averages). This data set contains friction-velocity-filtered, gap-filled, and partitioned fluxes and was used to assess the simulated surface fluxes of sensible heat ($H$), latent heat ($LE$), and carbon, including gross primary production (GPP) and ecosystem respiration (ER). The ER product was estimated according to the short-term temperature response of measured nighttime net ecosystem exchange (NEE) (Reichstein et al., 2005), and GPP as the difference between ER and NEE: i.e., ER − NEE. The gap-filled NEE values (and derived GPP and ER) using the Marginal Distribution Sampling method (Reichstein et al., 2005) were used in this study.

Eddy-covariance and meteorological data from the AmeriFlux L2 data product (version V007, 30-min averages) were used to calculate canopy conductance ($G_c$, see Sect. 2.7) and atmospheric vapor pressure deficit (VPD). In the analysis, 30-min surface flux data were rejected during periods when the wind direction was in the [45$^{o}$: 135$^{o}$] sector (same criterion used by Wharton et al., 2012), as the northeast-to-southeast wind sector is characterized by heterogeneous (age-fragmented) land cover. The data were hourly averaged prior to $G_c$ and VPD calculation. L2 soil water content (SWC) data were also

used in the analysis. Missing SWC data from the L2 dataset in the year 2002 were replaced by respective L1 data (version Apr2013).

The AmeriFlux L2 data product (version V007, 30-min averages) was also used to assess the energy balance closure at the site. The energy balance ratio, $EBR = (H + LE)/(R_n - G)$, where $R_n$ is net radiation and $G$ is soil heat flux, was calculated for dry season months (June to September) using 10:00–14:00 PST data and rejecting periods with rain or unfavorable wind direction ($[45^o : 135^o]$ sector). With the available data, EBR could be calculated for the years of 1998–2001, 2004, and 2006.

## 2.4 Carbon Isotope Data

Estimated $\delta^{13}C$ values of ER (Lai et al., 2005) and observed $\delta^{13}C$ values of leaf tissue and soil organic matter (Fessenden and Ehleringer, 2003) at Wind River were used to assess the photosynthetic $^{13}C$ discrimination in CLM. Lai et al. (2005) used an automated air sampling system, with inlets at 0.5 m above ground level and at 0.5 canopy height, collecting 15 flasks weekly during the growing season. Most of the flasks (13 out of 15) were dedicated to nighttime sampling (over a single night). The Keeling-plot method was used to infer the weekly $\delta^{13}C_{ER}$ using the $CO_2$ and $\delta^{13}CO_2$ observations (for simplicity, the resulting $\delta^{13}C_{ER}$ values are referred to as "observations" in the text). The monthly averages (June–November) from 2001 to 2003 reported by Lai et al. (2005) were used as reference in the present study. Fessenden and Ehleringer (2003) conducted measurements of $\delta^{13}C$ of bulk organic tissue from current-year needles of *Tsuga heterophylla* trees and seedlings at the top (55 m), middle (25 m), and bottom (2 m) of the canopy. They also conducted vertical profile measurements of $\delta^{13}C$ of bulk soil organic carbon down to 20-cm depth. The measurements were performed on a 1-month to 2-month time interval. The values reported by Fessenden and Ehleringer (2003) for the growing season in 1999 and 2000 were used as reference.

In the present study, both observed and modeled carbon isotope ratios were expressed as $\delta^{13}C = \left(\frac{R_x}{R_{std}} - 1\right) \times 1000$ (‰), where $R_x$ is the $^{13}C{:}^{12}C$ isotope ratio of the carbon pool/flux of interest and $R_{std}$ is the $^{13}C{:}^{12}C$ isotope ratio of a standard reference material (Vienna Pee Dee Belemnite standard).

## 2.5 CLM Configuration and Simulations

CLM was run at site level using the PTCLM scripting framework (see Kluzek, 2013), as in recent studies (e.g., Mao et al., 2016; Raczka et al., 2016). Land cover was defined as the needleleaf evergreen temperate tree plant functional type. The model was configured to use CLM v.4.5 physics and CLM v.4.5 CN biogeochemistry. The vertical soil carbon profile option was turned on, and the CENTURY Carbon model was selected for the decomposition parameters. The nitrification and de-nitrification sub-model was switched off, as preliminary simulations indicated an excess of nitrogen availability and forest productivity when the respective module was active. Given that the Wind River site is characterized by an old-growth mature forest, no land-cover disturbance was considered in the simulations.

The model was spun-up in a two-stage process, using a pre-industrial component set with a constant, pre-industrial atmospheric $CO_2$ concentration and $\delta^{13}CO_2$ of 285 ppmv and $-6.5$‰, respectively. The model was run in accelerated decomposition mode for 600 years (first stage) and then in normal decomposition mode for 1000 years (second and final stage), using the local observed meteorological data (Sect. 2.3) from 1998 to 2006 to drive the model (continuously cycled). Following the spin-up process, a transient run (1850–2006) was performed with prescribed nitrogen deposition, atmospheric $CO_2$ concentration, and atmospheric $\delta^{13}CO_2$.

The transient atmospheric $CO_2$ concentrations used in this study were based on the CMIP5 recommendations for annual global mean values (Meinshausen et al., 2011). The transient atmospheric $\delta^{13}CO_2$ values used here were based on ice-core and flask measurements reported by Francey et al. (1999) (annual values in their spline fitting from 1850 to 1981) and flask measurements in Mauna Loa (annual averages from 1981 to 2006) by the Scripps $CO_2$ program (Keeling et al., 2005), following a similar methodology as in Raczka et al. (2016) (note that, unlike in Raczka et al. (2016), here a seasonal cycle was not superimposed onto the time series). As in the spin-up process, the local observed meteorological data from 1998 to 2006 were cycled during the transient run. The driver-data and model years were aligned in a way to guarantee a perfect match between them during the final 9 years of the simulation (1998–2006).

## 2.6 CLM Calibration

Initial simulations using the default parameters from CLM resulted in a poor representation of the carbon dynamics at the Wind River site (Figs. A5 and A6). GPP and forest biomass were significantly underestimated. The seasonality of ER was poorly represented and the simulated late-summer GPP was impacted by an underestimation of SWC, resulting from a poor representation of soil hydrology at the site. Furthermore, the modeled evapotranspiration values were significantly overestimated for the given values of GPP, i.e., the simulated water use efficiency was much lower than the observed. As a result of CLM's poor performance in the simulation of GPP and evapotranspiration, the modeled photosynthetic $^{13}C$ discrimination was found to be overestimated, reflected on reduced $^{13}C{:}^{12}C$ ratios of carbon fluxes and pools.

In order to improve the representation of the carbon dynamics at the site, key model parameters were calibrated as detailed in Appendix A and summarized in Table 1. The adjusted parameters were primarily based on biological measurements at Wind River or at similar stands in the Pacific Northwest. Parameters controlling specific leaf area and stomatal conductance were found to be critical to the simulation of GPP and evapotranspiration and were manually adjusted in a way to minimize the differences between model output and site observations (eddy covariance fluxes). The default soil hydraulic parameters used in CLM version 4.5 were found to be inadequate at Wind River, leading to severe underestimation of SWC and unrealistic soil moisture stress in the model during late summer. These parameters were reverted back to their default values in CLM version 4.0, with significant improvement in the representation of soil hydrology at the site. In an additional measure to reduce the unrealistic late-summer soil moisture stress in the model, root distribution was adjusted based on CLM's default parameter values for the broadleaf evergreen temperate tree plant functional type, shifting roots towards deeper soil layers (justified based on physical understanding of the site – see Appendix A8).

Bayesian parameter calibration is a common approach used in modelling studies to account for both the prior parameter distributions and more recent observations. In this case, a Bayesian calibration approach would be complicated by the current lack of prior parameter distributions within CLM in order to create a model ensemble and the computational expense of running a calibration. Commonly used techniques such as Markov Chain Monte Carlo (MCMC) are prohibitively expensive with long CLM simulations, and more advanced techniques for calibration (e.g. using surrogate modeling approaches) are still under development. The simpler approach used here proved to be an effective method to improve model performance at the Wind River AmeriFlux site. The reader is referred to Appendix A for a more complete description of the parameters that were adjusted and the calibration approach used. All model results presented and discussed in Sects. 3 and 4, unless noted otherwise, are based on the optimized model.

## 2.7 Canopy Conductance

Observed canopy conductance ($G_c$, m s$^{-1}$) was calculated by combining hourly tower data (see Sect. 2.3) with the Penman-Monteith equation (Monteith, 1964) as in Wharton et al. (2012):

$$G_c = \left[\frac{\rho c_p \text{VPD}}{\gamma LE} + \frac{\left(\frac{\Delta_{sat}}{\gamma}\right)\left(\frac{H}{LE}\right) - 1}{G_a}\right]^{-1} \tag{13}$$

where $\rho$ and $c_p$ are the density and specific heat of air, respectively (kg m$^{-3}$, J kg$^{-1}$ K$^{-1}$), VPD is the atmospheric vapor pressure deficit (kPa), $LE$ is the latent heat flux (W m$^{-2}$), $\Delta_{sat}$ is the slope of the saturation vapor pressure curve as a function of air temperature (kPa K$^{-1}$), $\gamma$ is the psychrometric constant (kPa K$^{-1}$), $H$ is the sensible heat flux (W m$^{-2}$), and $G_a = u_*^2/U$ is the aerodynamic conductance for momentum transfer (m s$^{-1}$), where $u_*$ is the friction velocity and $U$ is the wind speed. Atmospheric pressure and air temperature data and the ideal gas law were later used to convert the $G_c$ values to mmol m$^{-2}$ s$^{-1}$. The calculation of $G_c$ was restricted to daytime hours (10:00–16:00 PST) and to the months of June to September (dry season). Rain events and periods with $LE < 5$ W m$^{-2}$ or relative humidity $> 80\%$ were disregarded. $G_c$ values outside the interval of 0 to 1000 mmol m$^{-2}$ s$^{-1}$ were also disregarded.

For comparison against observations, modeled canopy conductance values were calculated using the same methodology described above, but using hourly CLM output ($H, LE, u_*$) instead. An alternative would be to calculate canopy conductance directly by upscaling CLM's leaf stomatal conductance and leaf boundary-layer conductance using leaf area index (Eq. 6). Canopy conductance values derived from both approaches were found to be strongly correlated. The Penman-Monteith method was ultimately selected for the calculation of $G_c$ in order to allow a more direct comparison between modeled and observed values. This comparison was done as a way to assess the performance of CLM in the simulation of leaf stomatal conductance.

# 3 Results

## 3.1 Carbon Pools and Isotopic Signatures

Figure 2 shows modeled LAI, carbon stocks (leaf, fine root, coarse root, tree wood, and soil organic matter (SOM) carbon), and $\delta^{13}C$ of leaf and SOM pools throughout the transient run (1850–2006). Before the transient run, the model was spun-up and successfully equilibrated under the defined pre-industrial scenario, with LAI, carbon stocks, and leaf/SOM carbon isotope ratios reaching steady state (results not shown). The cyclic behavior exhibited in Fig. 2 is related to the driving meteorological data set, which was cycled throughout the simulation period (Sect. 2.5).

From 1850 to 2006, modeled LAI and carbon stocks (Fig. 2a–f) increased due to $CO_2$ fertilization and increasing nitrogen deposition. Average values of LAI, leaf carbon, and tree wood carbon were in agreement with the reference values reported in the AmeriFlux database for the Wind River site. Modeled fine root and coarse root carbon were underestimated, but within 2 standard deviations from the reference values.

The $\delta^{13}C$ of leaves and SOM was initialized in the model with a value of −6‰ (default value in CLM, close to the pre-industrial atmospheric $\delta^{13}CO_2$ value of −6.5‰ used in this study). During the model spin-up, in which constant pre-industrial atmospheric $\delta^{13}CO_2$ and $CO_2$ concentration values were prescribed, the $\delta^{13}C$ values stabilized at $\approx$ −26‰. During the transient simulation, the $\delta^{13}C$ of both leaves (Fig. 2g) and SOM (Fig. 2h) decreased (the pools became isotopically "lighter"), mostly due to the decreasing atmospheric $\delta^{13}CO_2$ values associated with the "Suess effect" (Keeling, 1979) but also due to the increasing atmospheric $CO_2$ concentration values. The $\delta^{13}C$ of leaves declined faster over the years than the $\delta^{13}C$ of SOM, given the fact that leaves have a significantly shorter turnover time than SOM and therefore present a faster response to the changes in atmospheric $\delta^{13}CO_2$ and $CO_2$ concentration. Modeled $\delta^{13}C$ of leaves compared well against the site observations for top and mid-canopy leaves (−0.8‰ and +0.8‰ difference, respectively), and modeled $\delta^{13}C$ of SOM (top 1 m of soil) compared well against site observations for SOM at 20 cm below ground (−0.4‰ difference).

It is important to clarify that CLM has leaf properties that vary continuously with canopy depth, and that two leaf categories (sunlit and shaded leaves) are estimated dynamically on every time step, as a function of canopy structure and solar elevation angle (Thornton and Zimmerman, 2007). The modeled leaf $\delta^{13}C$ output corresponds to the isotopic signature of the entire leaf carbon pool, which is calculated from both sunlit and shaded portions of the leaf canopy (see Sect. 2.1). The observed leaf $\delta^{13}C$ values in Fig. 2g correspond to measurements at canopy top (55 m), middle (25 m), and bottom (2 m). As pointed out by Fessenden and Ehleringer (2003), the decrease in the observed leaf $\delta^{13}C$ values (i.e., increase in photosynthetic $^{13}C$ discrimination) with canopy depth can be explained by light reduction within the canopy. In principle, the observed mid-canopy values are expected to better represent the isotopic composition of leaves for the whole canopy, in comparison with the observed values at the two canopy extremes, especially given the larger amount of leaf biomass in mid canopy. However, considering how light is reduced within the canopy, the top-canopy $\delta^{13}C$ value should still be representative of a significant fraction of the canopy as well, so the whole canopy $\delta^{13}C$ is expected to lay somewhere in

between the top- and mid-canopy values. As shown in Fig. 2g, the modeled $\delta^{13}C$ of the leaf carbon pool was the average between the observed values at canopy top and middle.

The overall agreement between the observed and modeled carbon isotope ratios indicates that CLM had skill in simulating the balance between assimilation and stomatal conductance and the associated photosynthetic $^{13}C$ discrimination. The adjustment of the parameters controlling stomatal conductance in the model ($m_{bb}$ and $b_{bb}$ – see Sect. 2.6, Table 1 and Appendix A9) to improve the simulation of evapotranspiration had a significant impact on the simulation of photosynthetic $^{13}C$ discrimination. When using the default parameter values (resulting in significantly higher stomatal conductance values), the modeled values of $\delta^{13}C$ in leaves and SOM were generally 2–3‰ lower (Fig. A1), departing from site observations.

## 3.2 Energy and Carbon Fluxes

Modeled energy and carbon fluxes are compared against daily-averaged observations in Fig. 3 for the period between 1998 and 2006. "Observed" GPP and ER were obtained from applying a partitioning model to NEE measurements (Sect. 2.3), but are referred to as "observations" in the text.

Modeled $LE$ values were close to observations, with a mean bias error (MBE) of $\approx -3$ W m$^{-2}$ and a root mean square error (RMSE) of $\approx 20$ W m$^{-2}$. The adjustment of the stomatal conductance parameters $m_{bb}$ and $b_{bb}$ (Table 1) was fundamental in modifying the $LE$ simulations. When using the default parameter values the modeled evapotranspiration was significantly overestimated, with summer values exceeding observations by almost 100% (Fig. A2a).

In 1998–2003 the model overestimated $H$ (MBE $\approx 32$ W m$^{-2}$, RMSE $\approx 40$ W m$^{-2}$), while in 2004–2006 the modeled values were closer to observations (MBE $\approx 10$ W m$^{-2}$, RMSE $\approx 36$ W m$^{-2}$). The modeled $H$ values did not present significant interannual variability in 1998–2006; however, the observations showed significantly smaller fluxes in 1998–2003 than in 2004–2006. Such changes in the magnitude of $H$ were reported as a potential data issue in the Wind River site documentation available in the AmeriFlux repository.

The overall mean EBR calculated from site observations was 0.88 (see Sect. 2.3 for calculation approach). The energy balance closure for years 2004 and 2006 was high (mean EBR = 1.01 and 1.09, respectively). The model bias of $H$ and $LE$ was relatively small in those years (Figs. 3a, b). In years 1998, 2000 and 2001, mean EBR was significantly lower (0.63, 0.69 and 0.76, respectively). Modeled $H$ presented a relatively large positive bias in those years (Fig. 3a). As discussed above, the observed $H$ values in 1998–2003 were significantly smaller than in 2004–2006, while the $LE$ observations showed approximately the same pattern over the years. The low EBR for years 1998, 2000 and 2001 supports the remark included in the AmeriFlux documentation regarding a potential data issue with $H$ and suggests that the observed values were biased low in 1998–2003. Mean EBR in 1999 was relatively high (0.92), where the reduced $H$ values (Fig. 3a) were compensated by larger $LE$ values (Fig. 3b). In that year, modeled $H$ ($LE$) had a positive (negative) bias in respect to the observations.

Modeled GPP resembled observed values, with small differences (MBE $\approx 0.23$ gC m$^{-2}$ day$^{-1}$, RMSE $\approx 1.60$ gC m$^{-2}$ day$^{-1}$). Modeled ER exhibited closer correspondence with measurements during the spring and summer months in general (MBE $\approx 0.82$ gC m$^{-2}$ day$^{-1}$, RMSE $\approx 1.85$ gC m$^{-2}$ day$^{-1}$), with summer peaks especially close to measured values. In the colder months, modeled ER was significantly overestimated (MBE $\approx 1.46$ gC m$^{-2}$ day$^{-1}$, RMSE $\approx 1.77$ gC m$^{-2}$ day$^{-1}$).

Despite the significant improvement in the seasonal behavior of ER after the $Q_{10}$ adjustments discussed in Appendix A6, the results indicate that further adjustments also including the base rate of maintenance respiration and the base decomposition rates for each litter and SOM pool within CLM would be necessary to better simulate the observed ER at Wind River. The results suggest that lower base rates and higher $Q_{10}$ values would improve the simulations at the site.

## 3.3 Isotopic Signatures of GPP and ER

### 3.3.1 Diurnal Cycle

Modeled $\delta^{13}C_{GPP}$ exhibited a well-defined diurnal cycle (Fig. 4), with minimum values in the early morning and late afternoon and a peak value typically in mid-afternoon, reflecting diurnal changes in the simulated iWUE (see Eqs. 10 and 12). Modeled $\delta^{13}C$ values of the heterotrophic component of ecosystem respiration (HR) were approximately constant, with a $\approx 0.2‰$ change over the entire period of study (1998–2006). On the other hand, modeled $\delta^{13}C$ values of the autotrophic component (AR) were found to be virtually equal to modeled $\delta^{13}C_{GPP}$ during daytime. At nighttime, modeled $\delta^{13}C_{AR}$ was found to change abruptly towards values closer to modeled $\delta^{13}C_{HR}$. Because AR was the major component of the total ecosystem respiration (ER=AR+HR; see Fig. 4a), modeled $\delta^{13}C_{ER}$ exhibited a similar behavior compared to modeled $\delta^{13}C_{AR}$ (Fig. 4b).

In CLM, newly assimilated carbon is first allocated to meet the total maintenance respiration demand of live plant tissues (top priority). When this demand exceeds the supply of carbon via photosynthesis (e.g., during nocturnal periods, wintertime, stress periods), carbon is drawn from a storage pool (excess maintenance respiration pool; $CS_{xs}$), which is allowed to run a deficit state. The reason CLM allows this deficit state is to avoid the requirement of knowing the size of the total storage pool available to plants and thus the possibility of vegetation dying in a given location if the storage pool is depleted. When negative, $CS_{xs}$ is gradually replenished with newly assimilated carbon at a potential rate of $-CS_{xs}/\tau_{xs}$, where $\tau_{xs}$ is a time constant (set to 30 days in CLM). The carbon allocation flux to replenish $CS_{xs}$ receives second priority in the model, while the carbon allocation fluxes to support plant growth have third priority. Given this allocation structure, $\delta^{13}C_{AR}$ will follow $\delta^{13}C_{GPP}$ during daytime (assuming GPP is enough to meet the maintenance respiration demand), and the $\delta^{13}C$ of the "excess maintenance respiration flux" ($\delta^{13}C_{XSMR}$) during nighttime. CLM does not calculate the isotopic signature of XSMR from $CS_{xs}$, but from bulk vegetation tissues (total vegetation carbon, TOTVEGC). This is done because $CS_{xs}$ is not a physical quantity, but a construct of CLM. Note that XSMR "borrows" carbon from the $CS_{xs}$ pool, which is allowed to run a deficit state. This "debt" is paid in the future with the replenishment of the $CS_{xs}$ pool with newly assimilated carbon.

This construct makes the $\delta^{13}C$ of $CS_{xs}$ non-physical, therefore, the approximation that $\delta^{13}C_{XSMR}=\delta^{13}C_{TOTVEGC}$ is more physically realistic. This approximation makes the nocturnal $\delta^{13}C_{AR}$ to follow $\delta^{13}C_{TOTVEGC}$, explaining the low sensitivity of the nocturnal $\delta^{13}C_{AR}$ to recent $^{13}C$ discrimination in the results shown in Fig. 4b.

Autotrophic respiration at Wind River is likely fueled by a mixture of stored and recently-fixed carbon, as indicated by $^{14}C$ measurements from root respiration at the site (Taylor et al., 2015). This process cannot be appropriately modeled by CLM with the current carbon allocation scheme, impacting the simulation of $\delta^{13}C_{ER}$. An explicit representation of carbohydrate storage pools within CLM to support the maintenance respiration demand would improve the simulation of $\delta^{13}C_{ER}$. The need for a better representation of carbohydrate storage pools within CLM was also highlighted by the $^{13}CO_2$-labeling study conducted by Mao et al. (2016).

It is important to highlight that, unlike models such as SiB (Sellers et al., 1996; Vidale and Stöckli, 2005), CLM does not have a prognostic canopy airspace where $\delta^{13}CO_2$ is impacted by photosynthetic and respiratory fluxes, so the simulation of $\delta^{13}C_{GPP}$ is not affected by the above described limitations in the simulation of $\delta^{13}C_{ER}$.

### 3.3.2 Seasonal Cycle

Modeled $\delta^{13}C_{GPP}$ exhibited a well-defined seasonal pattern, peaking during the summer as a result of a decrease in the photosynthetic $^{13}C$ discrimination associated with higher iWUE values (Fig. 5; see also Eqs. 10 and 12). The summer peak in iWUE was linked to changes in stomatal conductance in response to increased VPD and reduced SWC during the dry summer season.

On a monthly scale, roughly indicated by the smoothed curve in Fig. 5, the modeled $\delta^{13}C_{GPP}$ values presented a similar seasonal pattern in comparison with the $\delta^{13}C_{ER}$ observations at the site by Lai et al. (2005). Differences between $\delta^{13}C_{GPP}$ and $\delta^{13}C_{ER}$ are obviously expected, as $\delta^{13}C_{ER}$ depends on the contribution of recently-assimilated carbon to AR, the AR:ER ratio, and also post-photosynthetic fractionation (Bowling et al., 2008; Brüggemann et al., 2011). The seasonal pattern in the observed $\delta^{13}C_{ER}$ (Fig. 5) could be partially attributed to an eventual spring-to-summer decrease in AR:ER ratio (assuming $\delta^{13}C_{HR} > \delta^{13}C_{AR}$). $^{14}C$ measurements from below-ground respiration components at Wind River reported by Taylor et al. (2015) do indicate a spring-to-summer decrease in the contribution of root respiration (RR) towards total soil respiration (SR=RR+HR). The similarity of the seasonal patterns of observed $\delta^{13}C_{ER}$ and modeled $\delta^{13}C_{GPP}$ suggests that stomatal response to water stress could also be driving the seasonal pattern in the observed $\delta^{13}C_{ER}$ at the site. The broader implication is that $\delta^{13}C_{ER}$, which can be more easily measured than $\delta^{13}C_{GPP}$, can be reasonably used as a surrogate to indicate forest response to water stress at Wind River.

Due to the limitations in the carbon allocation scheme used in CLM (Sect. 3.3.1), the simulated $\delta^{13}C_{ER}$ values were found to be inconsistent with the site observations, with nocturnal values approximately constant throughout the entire period of study (1998–2006), exhibiting little sensitivity to recent photosynthetic $^{13}C$ discrimination. Diurnal values, on the other hand, were found to be strongly correlated with $\delta^{13}C_{GPP}$, given the fact that in CLM current photosynthate directly fuels AR (results not shown).

The adjustment of the stomatal conductance parameters $m_{bb}$ and $b_{bb}$ to improve the simulation of evapotranspiration (Sect. 2.6, Table 1 and Appendix A9) led to a significant change in the simulation of $\delta^{13}C_{GPP}$. When the default parameter values were used, modeled $\delta^{13}C_{GPP}$ was generally 2–3‰ lower due to higher photosynthetic $^{13}C$ discrimination (Fig. A2b), also presenting a considerable reduction in the amplitude of the seasonal cycle. The difference between modeled $\delta^{13}C_{GPP}$ and observed $\delta^{13}C_{ER}$ was significantly larger. As discussed in Sect. 3.1, site observations of leaf and SOM $\delta^{13}C$ support the notion that the default stomatal conductance parameters are inadequate at Wind River, resulting in excessive photosynthetic $^{13}C$ discrimination.

## 3.4 Ecosystem Response to Water Stress

Overall, CLM was able to reasonably capture the observed interannual variability of GPP at the study site (Fig. 3c). The behavior of observed GPP in 2002 and 2006 stands out, showing an early-season peak followed by a quick reduction, suggesting strong water stress in those years, especially in 2002. Among the years studied here, 2002 and 2006 had the lowest summer precipitation. Spring precipitation was also low in 2006 but normal in 2002. Observed canopy conductance during the spring and summer of 2006 was smaller than in 2002, but a stronger attenuation of GPP was observed in 2002, suggesting that water stress was not the main reason for the attenuated GPP values in 2002. CLM was able to simulate the observed GPP behavior in 2006 but not in 2002. The reason for the model-data mismatch in the spring/summer of 2002 is currently unclear. Despite the fact that meteorological forcing data from 1998–2006 were continuously cycled throughout the transient run (1850–2006), meaning that the impact of any slow secular change in the forcing data was not captured in the simulation, the simulated GPP still compared reasonably well against observations.

Throughout the simulation period (1998–2006), CLM predicted a few periods where the ecosystem was under the influence of soil moisture stress (Fig. 6). As indicated by the $\beta_t$ parameter (Eq. 3), which varies from 0 (maximum soil moisture stress) to 1 (no soil moisture stress) (see Sect. 2.1), those periods included the summers of 1998, 2006, 2003, and 2002, in decreasing order of stress intensity. The departures from $\beta_t = 1$ typically occurred when modeled SWC (top 5 soil layers, 0–27 cm) decreased below $\approx 20\%$. Note that, at Wind River, SWC at permanent wilting point and at field capacity is 14% and 30%, respectively (Wharton et al., 2009).

With the adjustment of soil hydraulic parameters (Appendix A7), CLM was able to adequately simulate SWC throughout most of the years within the study period (Fig. 6), especially during the summer months, with an overall summer MBE of 3.24%. However, the simulated SWC significantly departed from observations in 1999–2002. CLM, which was driven by observed precipitation at the site, indicated higher SWC than observations in 1999–2002, particularly during the summer months, with a summer MBE of 8.05%. For the remaining years, summer MBE was −0.27%. The SWC observations starting on the second year of the site records (1999) up to the data gap in 2002 presented a different pattern in comparison with the remaining years, showing an apparent negative offset of near 10%. It is likely that the apparent shift in the time series of observed SWC was instrument-related. In 1999–2002, soil moisture monitoring at the site consisted of 2, 2-pronged TDR probes instead of 6, 3-pronged TDR probes, likely resulting in less-accurate data collection.

Observed canopy conductance was found to be strongly dependent on VPD, following a decreasing exponential relationship (Fig. 7). In order to investigate the additional dependence on soil moisture stress, the data points were divided into 4 bins according to the observed values of SWC (Fig. 7a). The linear regression fit between $\log G_c$ and VPD for the points corresponding to the lowest SWC bin (SWC<17.5%, $\approx 22\%$ of all data points) was virtually the same as the linear regression considering all data points. If the forest were under soil moisture stress at those low SWC levels, the former regression curve with data points from the lowest SWC bin would be expected to be found below the latter. Instead, the SWC<17.5% regression curve was very similar—even slightly above the regression curve using all data points.

The lack of sensitivity of observed $G_c$ to observed SWC (Fig. 7a) was likely associated with a negative bias in SWC in 1999–2002. Observed $G_c$ was found to respond to modeled SWC (driven by observed precipitation) (Fig. 7b). As discussed above, the observed SWC values in 1999–2002 were suspected to have a negative bias, i.e., drier than reality. $G_c$ values in Fig. 7a corresponding to the summer of 1999–2002 were tagged as belonging to the lowest SWC bin, but in reality, they could be associated with wetter, non-moisture stress conditions. Assuming CLM's summer simulated SWC (driven by observed precipitation) was not as biased as the observed SWC might be, we instead used the modeled SWC values to probe the $G_c$ vs. VPD relationship under different SWC regimes in Fig. 7b. Interestingly, with this approach, a distinct pattern emerged for the data points within the lowest SWC bin. The regression curve considering all data points was $\log G_c = -0.59\text{VPD} + 6.06$ ($r = -0.60$) and when considering only the data points from the lowest bin (modeled SWC<21.25%, $\approx 24\%$ of all points), the regression curve was $\log G_c = -0.50\text{VPD} + 5.71$ ($r = -0.56$). The latter regression curve corresponded to reasonably lower $G_c$ values, especially at low VPD levels, which is compatible with a moisture stress scenario. The result supports the suspicion of a negative bias in the observed SWC data in 1999–2002.

Similar to observations, modeled canopy conductance was also found to be strongly dependent on VPD (Fig. 8). This is expected given the Ball-Berry stomatal conductance model used in CLM (Eq. 2). The Ball-Berry model has a direct dependence on leaf relative humidity (leaf RH), not leaf VPD, but these variables are strongly correlated. The correlation between modeled $G_c$ and RH was found to be slightly higher than between modeled $G_c$ and VPD, while observed $G_c$ correlated slightly better with VPD than RH (results not shown). The results indicate that a direct dependence on leaf VPD in CLM's stomatal conductance model, rather than leaf RH, would lead to a more accurate representation of stomatal functioning at Wind River, but overall, for the period analyzed in the present study, such improvement is expected to be small. The general dependence of modeled canopy conductance on VPD was very similar in comparison with observations, as indicated by the linear regression curve between $\log G_c$ and VPD in Fig. 8 using all data points ($\log G_c = -0.59\text{VPD} + 6.04$; compare with $\log G_c = -0.59\text{VPD} + 6.06$ in Fig. 7b). The correlation between observed $\log G_c$ and VPD, however, was lower than for the model results ($r = -0.60$ and $r = -0.91$, respectively).

The impact of soil moisture stress on $G_c$ was reasonably captured in CLM (Fig. 8b; cf. Fig. 7b). The impact of soil moisture stress on modeled $G_c$ is clearly visible in Fig. 8a, in which the data points were binned according to $\beta_t$. With increasing soil moisture stress (decreasing $\beta_t$ values), the modeled $G_c$ values still maintained a strong dependence on VPD,

but were shifted downward, particularly at low VPD levels. In order to allow a more direct comparison against Fig. 7b, the data points were binned according to modeled SWC in Fig. 8b. The points in the lowest SWC bin (SWC<21.25%, ≈ 22% of all points) roughly corresponded to the periods under soil moisture stress ($\beta_t < 1$). The regression curve for the SWC<21.25% group laid reasonably below the regression curve considering all data points ($\log G_c = -0.53\text{VPD} + 5.80$, $r = -0.90$ and $\log G_c = -0.59\text{VPD} + 6.04$, $r = -0.91$, respectively). The regression curves associated with SWC<21.25% were similar for the observed and modeled results (Figs. 7b and 8b), indicating that CLM could reasonably simulate soil moisture stress at Wind River, although with a small underestimation (i.e., a small overestimation of $G_c$; note the $G_c$ intercepts at 301 and 331 mmol m$^{-2}$ s$^{-1}$ in Figs. 7b and 8b, respectively). It is important to point out, however, that modeled SWC was used to segregate the observations in Fig. 7b due to the potential bias in the SWC observations discussed above.

Modeled $\delta^{13}\text{C}_{\text{GPP}}$ and $G_c$ were highly correlated ($r = -0.88$, $p < 0.001$; Fig. 9b). Modeled $G_c$ generally decreased into the summer season, leading to an increase in water use efficiency and a decrease in photosynthetic $^{13}\text{C}$ discrimination, resulting in higher $\delta^{13}\text{C}_{\text{GPP}}$ values. Observed $\delta^{13}\text{C}_{\text{ER}}$ was found to have a low negative correlation with observed $G_c$, but not statistically significant ($r = -0.27$, $p = 0.396$; Fig. 9a). The low correlation was likely a result of $\delta^{13}\text{C}_{\text{ER}}$ reflecting constraints of prior environmental drivers in comparison with the more rapid response of $G_c$ to more recent environmental drivers. Another possible explanation is that the monthly $\delta^{13}\text{C}_{\text{ER}}$ values in Fig. 9a were obtained by averaging up to 4 discrete weekly observations (see Sect. 2.4), in contrast with the calculation of monthly $G_c$, which used daytime values for each day of the month. It is important to mention that the observed $\delta^{13}\text{C}_{\text{ER}}$ show a clear seasonal pattern (Fig. 5), with values peaking during summer likely in response to changes in $g_s$ and iWUE associated with increasing water stress (see discussion in Sect. 3.3.2), but the present results indicate a lag in this response.

## 4 Discussion

### 4.1 Ecosystem Response to Water Stress

We found that the major cause of water stress leading to stomatal response at Wind River during summer was the elevated VPD, and not the reduced soil moisture (Section 3.4). Observed canopy conductance values at the site strongly decreased at moderate VPD levels, regardless of soil moisture conditions (Fig. 7b). The high sensitivity of stomatal response to changes in VPD was also shown and discussed in Wharton et al. (2009). As pointed out in their study, "even under moderate VPD levels, foliage at the tops of tall evergreen conifer trees often reach near critical values for cavitation due to a long path distance between the water table and the hydraulic capacity of the xylem, and as a result shut their stomata frequently (Ryan and Yoder 1997)". They also point out that soil moisture depletion is usually not limiting at the site because the mature trees are capable to tap water from deeper soil layers. This is generally consistent with our findings (Sect. 3.4), however, we also found that stomatal conductance responded to soil moisture stress during periods of more severe SWC depletion and low VPD (Fig. 7b).

Overall, CLM was able to simulate the observed response of canopy conductance to VPD and SWC, reasonably capturing the impact of water stress on ecosystem functioning (Fig. 8). Similarly to observations, VPD exerted a strong limitation on modeled $G_c$, while SWC was usually not limiting. Note that $\beta_t$ was equal to 1 (no soil moisture stress) throughout most of the period of study (Fig. 6), in alignment with the explanation by Wharton et al. (2009) that the mature trees at the site are capable to access water from deeper soil layers. Note also that the default NETT PFT root distribution in CLM was shifted towards deeper soil layers (Appendix A8), aiming to improve the simulation of $\beta_t$. Despite the good overall model-data agreement ( $G_c$ dependency on VPD and SWC) after calibration, the results indicate a small underestimation of soil moisture stress in CLM, as discussed in Sect. 3.4. Calibration of the parameters controlling the plant wilting factor (Eq. 5) and additional calibration of the root distribution parameters could improve the results but are out of scope here.

An obvious but important point that must be highlighted is that in order to adequately simulate soil moisture stress, CLM must first adequately simulate SWC. Even when the model is driven by observed precipitation data (the case of the present study), this task is not trivial. As discussed in Appendix A7, CLM's hydrology submodel performed poorly at Wind River when the default soil hydraulic parameters were used, leading to a strong dry bias in SWC. The original parameters used in the previous version of CLM (Version 4.0) were found to perform much better at the site, likely due to a reduction of subsurface runoff and consequent increase in water retention in the soil column. As the default parameter values are intended for global simulations, it is natural to expect site-to-site variation in model performance (see Sect. 4.2). Raczka et al. (2016), for instance, did not find issues with the default soil hydraulic parameters in their CLM 4.5 simulation at the Niwot Ridge AmeriFlux site. This difference in impact between the sites may have resulted from unique soil properties or differences in precipitation and evaporative demand between the sites during the summer.

As pointed out in Sect. 3.4, the results of the present study indicate that a direct dependence on leaf VPD in CLM's stomatal conductance model, rather than leaf RH, would lead to a more accurate representation of stomatal functioning at Wind River, but overall, for the period analyzed in the present study, such improvement is expected to be small. It is important to emphasize that this expectation refers to the results presented here only. In case of model predictions under future climate scenarios, in which atmospheric VPD is predicted to change while RH stays the same (as discussed in Sato et al., 2015), a direct dependence on leaf VPD in the stomatal conductance model becomes critical. The next CLM release (Version 5) is expected to replace the Ball-Berry model with the Medlyn model (Medlyn et al., 2011), which directly depends on leaf VPD. This modification is expected to be more relevant for climate change simulations. Note that the present analysis is based on a hindcast simulation using a stable climate.

## 4.2 Calibration of CLM

Substantial calibration of model parameters was necessary to simulate the observed energy and carbon dynamics at Wind River, an old-growth (~500-years-old) coniferous forest site dominated by Douglas-fir and western hemlock trees and characterized by Mediterranean climate. This is not surprising given that the default parameters used in CLM are intended

for global simulations, so model performance at particular sites is expected to vary greatly, requiring site-specific calibration in order to adequately simulate the observations. This is also demonstrated in the studies by Raczka et al. (2016) and Mao et al. (2016). Raczka et al. (2016) investigated the performance of CLM at the Niwot Ridge AmeriFlux site, a ~110-years-old subalpine coniferous forest site in Colorado, USA, consisting of lodgepole pine, Engelmann spruce, and subalpine fir, while Mao et al. (2016) evaluated CLM in a 10-year-old loblolly pine stand in Tennessee, USA. In both cases significant site-specific specification and calibration of model parameters were also necessary. Note that these sites fall into the same PFT category as Wind River (NETT PFT). Despite the significant differences between the 3 sites, the results presented here and in Raczka et al. (2016) and Mao et al. (2016) converge in respect to the calibration of the Ball-Berry stomatal conductance slope, $m_{bb}$. It is promising that despite the range in stand age and climate conditions amongst these sites, there appears to be a consensus that reduced stomatal conductance is required across all sites. This bodes well when upscaling to regional simulations.

A reduction of $m_{bb}$ from 9 (default) to 6 was necessary to simulate the observed GPP, LE, and $\delta^{13}$C values (leaf, SOM) at Wind River. This aligns with the results by Mao et al. (2016), as they were able to simulate the observations at their Tennessee site, including biomass $\delta^{13}$C values, with an optimized $m_{bb}$ of 5.6. However, as discussed in Appendix A9, the present results show that the significant reduction of $m_{bb}$ from 9 to 6 may represent a tradeoff with model representation of nitrogen limitation. When using CLM's default nitrogen limitation scheme and $m_{bb}$ value, Raczka et al. (2016) found significant overestimation of $^{13}$C discrimination at Niwot Ridge due to excessive stomatal conductance, similar to the present study. When using an alternative nitrogen limitation scheme based on $V_{cmax25}$ down-regulation, maintaining the coupling between net leaf assimilation and $g_s$, Raczka et al. (2016) found significant improvement in the simulations. This alternative scheme was also tested here while keeping the default $m_{bb}$ value, and the results were similar compared to the model run with default nitrogen limitation scheme and $m_{bb} = 6$ (Appendix A9).

The results in the present study indicate that it is possible to account for the partial coupling between net leaf assimilation and stomatal conductance in CLM through the adjustment of $m_{bb}$ to achieve reasonable carbon and energy exchange behavior, including $^{13}$C discrimination. This is also supported by the results in Mao et al. (2016). A more detailed evaluation of model skill in simulating $^{13}$C discrimination with this approach, in comparison with the $V_{cmax25}$ down-regulation approach (fully-coupled CLM), would depend on high-frequency observations of $\delta^{13}$C$_{GPP}$ as in Raczka et al. (2016). These data were not available at Wind River. Note that $^{13}$C discrimination at Wind River was inferred from $\delta^{13}$C measurements of leaves and soil organic matter.

The results in Raczka et al. (2016), Mao et al. (2016), and in the present study indicate that $m_{bb} = 9$ is "excessive" when the default nitrogen limitation implementation is used in the simulations, with the latter two studies indicating that $m_{bb} \approx 6$ is a more appropriate value to simulate the site observations. This agreement at 3 very distinct sites is encouraging and suggests that CLM could possibly benefit from a revised $m_{bb}$ value of 6 for the NETT PFT, keeping in mind that such adjustment to improve model skill would also account for structural error. At the same time, the results presented here and in

Raczka et al. (2016) indicate that the default $m_{bb} = 9$ is reasonable for simulations when the $V_{cmax25}$ down-regulation scheme is implemented in the model, although Raczka et al. (2016) still found a small overestimation of [13]C discrimination at Niwot Ridge suggesting that a smaller $m_{bb}$ value would better simulate the site dynamics. It is important to point out, however, that the experimental literature indicates generally lower $m_{bb}$ values for coniferous forests (see for example the

surveys by Williams et al., 2004, Table 6.3, and Miner et al., 2017, Fig. 1). The Simple Biosphere Model (Sellers et al., 1996), for instance, uses $m_{bb} = 6$ for conifers and $m_{bb} = 9$ for other $C_3$ plants, while CLM uses $m_{bb} = 9$ for all $C_3$ plants. Further investigation of the applicability of the revised $m_{bb}$ value (or the current default value while using the $V_{cmax25}$ down-regulation scheme as in Raczka et al., 2016) at other NETT PFT sites is recommended for future studies.

### 4.3 Recommendations for Structural Improvement within CLM

The results of the present study demonstrates that $\delta^{13}$C observations can be used to constrain stomatal conductance and iWUE in CLM as an alternative to eddy covariance flux measurements, leveraging the recent implementation of photosynthetic [13]C discrimination within the model. The adjustments made on the parameters controlling stomatal conductance within the model, originally aiming to improve the simulation of evapotranspiration, were critical to simulate the observed photosynthetic [13]C discrimination at Wind River, inferred from $\delta^{13}$C measurements of leaves and soil organic

matter. As discussed in Sect. 4.2, these adjustments to improve model skill interacted strongly with the nitrogen limitation scheme. A possible interpretation of results from this and other recent studies is that growth limitation due to restricted nitrogen availability does not operate instantaneously upon photosynthesis (e.g., through nitrogen downscaling in the default version of CLM 4.5) but is accounted for further "downstream" during the allocation of carbon.

For example, Metcalfe et al. (2017) proposed a revised model structure in which GPP is not instantaneously down-

regulated during photosynthesis, but the excess photosynthate, which cannot be allocated to structural pools due to insufficient nitrogen supply, is allocated to a new nonstructural carbohydrate storage pool within the model. Carbon from this pool is able to return to the atmosphere via the inclusion of a single additional respiration term within the model. This new model structure provides a solution for the issue regarding the partial coupling between net leaf assimilation and stomatal conductance. Alternatively, a foliar nitrogen model could be used to account for nitrogen limitation directly within

the estimation of photosynthetic capacity (Ghimire et al., 2016), removing the requirement for nitrogen downscaling. A similar approach is planned to be included in the next release of CLM (version 5.0).

The use of $\delta^{13}C_{ER}$ observations as a strong constraint upon CLM is hindered by the lack of an explicit representation of carbohydrate storage pools within the model to support autotrophic respiration (Fig. 4). The results from the [13]C-labeling study by Mao et al. (2016) also illustrate the issue and highlight the need of structural improvements in CLM's carbon

allocation scheme. One implication of this issue is that it prevents a more direct use of $\delta^{13}C_{ER}$ observations —which are easier to obtain and more frequently available than $\delta^{13}C_{GPP}$ observations— for evaluation of [13]C discrimination in CLM. It may also limit the applicability of CLM for global atmospheric [13]C budget studies focusing on land-ocean flux partitioning (e.g., van der Velde et al., 2013), as errors in the simulation of the land isotopic disequilibrium ($\delta^{13}C_{ER} - \delta^{13}C_{GPP}$) can

propagate to the estimation of the land-ocean partitioning and the estimation of variability of each sink (van der Velde et al., 2014). Van der Velde et al. (2014) were able to reasonably simulate mean observed $\delta^{13}C_{ER}$ values for a selection of sites from the Biosphere-Atmosphere Stable Isotope Network (BASIN; Pataki et al., 2003) using a modified version of the SiB-CASA model including representation of $^{13}C$ isotopes and modified carbon storage pools. The original SiB-CASA model (Schaefer et al., 2008) has a single storage pool representing sugars and starch, with only the sugar portion being readily available for plant growth and maintenance. The effective pool turnover rate in this configuration is ~70 days. In the modified model, sugar and starch allocation are simulated separately with 2 distinct pools, with prescribed turnover rates of 7 days (sugar to starch) and 63 days (starch to sugar). Van der Velde et al. (2014) found significant improvement in the simulation of $\delta^{13}C_{ER}$ with the new carbon allocation approach. We recommend that CLM adopt a similar carbon allocation scheme, moving away from the deficit-based accounting scheme (Sect. 3.3.1) towards an explicit representation of carbohydrate storage pools such as in the SiB-CASA model (van der Velde et al., 2014).

Another shortcoming in CLM is the fact that mesophyll conductance ($g_m$) is not simulated, i.e., intracellular and intercellular $CO_2$ are assumed to be equal. As demonstrated here and in Raczka et al. (2016) and Mao et al. (2016), CLM is able to reasonably simulate $^{13}C$ discrimination by either adjusting the stomatal conductance slope parameter or using an alternative nitrogen limitation scheme ($V_{cmax25}$ down-regulation), but the impact of not including $g_m$ in the simulations must be investigated. Mesophyll conductance was recently incorporated in CLM by Sun et al. (2014), however it still has to be linked to the carbon isotope submodel. This is another front where $^{13}C$ observations can be used for model evaluation and development.

## 5 Conclusions

After substantial calibration of model parameters, CLM was able to simulate energy and carbon fluxes, leaf area index, and carbon stocks at an old-growth coniferous forest (Wind River AmeriFlux site) in general agreement with site observations. Overall, the calibrated CLM was able to simulate the observed response of canopy conductance to atmospheric vapor pressure deficit and soil water content, reasonably capturing the impact of water stress on ecosystem functioning. Key model adjustments to simulate observed flux and carbon stock patterns included 1) parameters controlling the variation of specific leaf area through the forest canopy (SLA$_0$, $m$), with significant impact on GPP, 2) parameters controlling stomatal conductance ($m_{bb}$, $b_{bb}$), with significant impact on the simulated latent heat flux and water use efficiency, and 3) soil hydraulic parameters, with impact on soil water content and on the soil moisture stress parameter, $\beta_t$.

The calibrated CLM was able to simulate carbon isotope ratios of leaves and soil organic matter at Wind River, in general agreement with site observations. The adjustments made on the parameters controlling stomatal conductance within the model, originally aiming to improve the simulation of evapotranspiration, were critical to simulate the observed photosynthetic $^{13}C$ discrimination at the site, inferred from $\delta^{13}C$ measurements of leaves and soil organic matter. This demonstrates that stable carbon isotopes can serve as an alternative to eddy covariance flux measurements for constraining

stomatal conductance. The simulation of nocturnal $\delta^{13}C_{ER}$ was found to be inconsistent with site observations, with results showing little sensitivity to recent photosynthetic $^{13}C$ discrimination. The inclusion of explicit carbohydrate storage pools within CLM (and removal of the current deficit-based carbon accounting system) to support the maintenance respiration demand from live plant tissues would improve the simulation of $\delta^{13}C_{ER}$.

We found an optimized stomatal slope value ($m_{bb} = 6$) was necessary at Wind River, consistent with previous CLM experiments from distinct needleleaf evergreen temperate forest sites. This suggests that this parameterization could apply to broader scale simulations of this PFT. We also found a tradeoff between adjustment of stomatal slope and changes to the nitrogen limitation scheme. The best long term solution may be to replace this nitrogen scheme with alternative approaches.

The hydrology submodel within CLM and its parameterization deserve special attention because the simulation of soil water content has a direct impact on $\beta_t$, and thus on stomatal conductance. Wind River required a unique calibration to achieve reasonable soil moisture, that was not consistent across other sites. This suggests that simulation of soil moisture in regional studies should be used with caution.

       The recent inclusion of the photosynthetic $^{13}C$ discrimination functionality in CLM opens a new opportunity for
model testing and development. The results presented here demonstrate that carbon isotopes can expose structural weaknesses in the model, such as the deficit-based accounting system in CLM's carbon allocation scheme and the partial coupling between net leaf photosynthesis and stomatal conductance caused by the nitrogen limitation scheme. $\delta^{13}C$ observations provide a key constraint that may guide future CLM development.

**Appendix A: CLM Calibration**

Most of the adjustments were performed on parameters particular to the needleleaf evergreen temperate tree plant functional type in CLM. For brevity, this plant functional type is referred to as NETT PFT in the following Sections.

**A1 Carbon Allocation Ratios**

       By default, CLM uses a dynamic new-stem-carbon-to-new-leaf-carbon allocation ratio ($A_{s:l}$, gC gC$^{-1}$) for the NETT PFT, which rises with increasing net primary production. A survey by White et al. (2000) indicates an average $A_{s:l}$ of
25 $2.2 \pm 0.89$ gC gC$^{-1}$ for needleleaf evergreen forests. Measurements reported by Hudiburg et al. (2013) for a region close to the Wind River site and characterized by forests of similar species composition vary approximately between 1 and 3.5 gC gC$^{-1}$ (their Fig. A1 – Mesic sites). A fixed value of $A_{s:l} = 2$ gC gC$^{-1}$ (value also used by Thornton et al., 2002 in their BIOME-BGC simulations for the Wind River site) was found to improve the simulated forest biomass and was adopted in this study for the NETT PFT.

The new-fine-root-carbon-to-new-leaf-carbon allocation ratio parameter ($A_{fr:l}$, gC gC$^{-1}$) for the NETT PFT was also changed based on observations at the Wind River site reported in the AmeriFlux database indicating $A_{fr:l} = 0.385$

gC gC$^{-1}$ rather than the default value of 1 gC gC$^{-1}$. The change meant a significantly greater carbon investment to leaves, helping to increase the modeled GPP towards the site observations.

## A2  Carbon:Nitrogen Ratios

Leaf-litter C:N ratio ($CN_{llit}$, gC gN$^{-1}$) for the NETT PFT was adjusted based on measurements at the Wind River site (Klopatek, 2007) to 76.4 gC gN$^{-1}$ (mean observed value). Based on the mean observed $CN_{llit}$ and assuming a nitrogen retranslocation efficiency of 50% (survey by Parkinson, 1983 indicates efficiencies around 50% for conifer trees and 36–69% for Douglas-fir in particular), the leaf C:N ratio ($CN_l$, gC gN$^{-1}$) for NETT PFT was adjusted to 38.2 gC gN$^{-1}$. The updated parameters differ little from the default values ($CN_{llit} = 70$ gC gN$^{-1}$, $CN_l = 35$ gC gN$^{-1}$).

Fine-root C:N ratio ($CN_{fr}$, gC gN$^{-1}$) for the NETT PFT was also adjusted based on measurements at the Wind River site (Klopatek, 2007). The value was adjusted from 42 gC gN$^{-1}$ (default) to 64.7 gC gN$^{-1}$ (mean observed value), meaning a significantly smaller nitrogen investment in fine roots resulting in more nitrogen for investment in leaves. This change helped to increase the modeled GPP towards the site observations.

## A3  Leaf Longevity

Measurements reported by Hudiburg et al. (2013) for a region near the Wind River site and characterized by forests of similar species composition indicate leaf longevity ($\tau_l$) of 5 yrs. This value was adopted for the NETT PFT, replacing the default value of 3 yrs. This change contributed particularly to an increase in the modeled leaf area index.

## A4  Specific Leaf Area

In CLM, specific leaf area (SLA, m$^2$ leaf gC$^{-1}$) is assumed to be linear with canopy depth $x$ (expressed as overlying leaf area index, m$^2$ leaf m$^{-2}$ ground) (Thornton and Zimmermann, 2007):

$$\text{SLA}(x) = \text{SLA}_0 + mx \tag{A1}$$

where SLA$_0$ is the specific leaf area at the top of canopy and $m$ is a linear coefficient (m$^2$ ground gC$^{-1}$). Integrating this equation over the canopy, a relationship can be established where leaf area index (LAI, m$^2$ leaf m$^{-2}$ ground) is calculated as a function of leaf carbon ($C_l$, gC m$^{-2}$ ground), knowing the parameters SLA$_0$ and $m$ (Thornton and Zimmermann, 2007):

$$\text{LAI} = \frac{\text{SLA}_0(e^{mC_l}-1)}{m} \tag{A2}$$

The default NETT PFT values in CLM for SLA$_0$ and $m$ are 0.01 m$^2$ leaf gC$^{-1}$ and 0.00125 m$^2$ ground gC$^{-1}$, respectively. These values were found to be too large for the Wind River site. Using them in Eq. (A2) with a $C_l$ of 941 gC m$^{-2}$ ground  (mean observation at the Wind River site reported in the AmeriFlux database) results in an LAI of $\approx 18$ m$^2$ leaf m$^{-2}$ ground, instead of $\approx 9$ m$^2$ leaf m$^{-2}$ ground according to the observations at the Wind River site (AmeriFlux database).

In CLM, the maximum rate of carboxylation at 25°C ($V_{cmax25}$) is proportional to the area-based leaf nitrogen concentration defined as $N_a = 1/(CN_l \text{SLA}_0)$, i.e., $V_{cmax25} \propto 1/\text{SLA}_0$. Using the default NETT PFT values for $\text{SLA}_0$ and $m$ led to the development of large and thin leaves with reduced $N_a$ and $V_{cmax25}$, resulting in excessive LAI and significant down-regulation of GPP. Smaller $\text{SLA}_0$ values were attempted (manual trial and error), with $m$ values constrained by Eq.

(A2), the $\text{SLA}_0$ value, and the site observations of LAI and $C_l$ mentioned above, aiming to minimize model errors in the simulation of GPP and LAI. $\text{SLA}_0 = 0.006$ m$^2$ leaf gC$^{-1}$ and $m = 0.000985$ m$^2$ ground gC$^{-1}$ were found to significantly improve the simulations and were adopted instead of the default values. Measurements reported by Woodruff et al. (2004) indicate that the ratio of leaf dry mass to leaf area reaches 263 g m$^{-2}$ leaf near the canopy top at Wind River (their Fig. 6). Assuming the mass of carbon is 50% of the dry mass, the observed value corresponds to 131.5 gC m$^{-2}$ leaf, i.e., an $\text{SLA}_0$

value of 0.0076 m$^2$ leaf gC$^{-1}$, indicating that the optimized $\text{SLA}_0$ value moved in the right direction from the default NETT PFT value (0.0100 down to 0.0060 m$^2$ leaf gC$^{-1}$), but ended up slightly lower than the observed value.

## A5 Tree Mortality

Results reported by van Mantgem et al. (2009) indicate an increasing trend of plant mortality rates ($M$, yr$^{-1}$) for Pacific Northwest forests, with $M$ growing from $\approx 1\%$ yr$^{-1}$ in 2000 towards $\approx 1.5\%$ yr$^{-1}$ in 2010. In CLM, a default rate of

$M = 2\%$ yr$^{-1}$ is used for all vegetation types, which was found to be excessive at Wind River, leading to a reduced modeled forest biomass. $M = 1.5\%$ yr$^{-1}$ was found to yield results closer to site observations and was therefore adopted in this study.

## A6 Temperature Sensitivity Coefficient ($Q_{10}$)

The effect of temperature on maintenance respiration (component of autotrophic respiration) in CLM is calculated via a $Q_{10}$ formulation, where the base rate of maintenance respiration is multiplied by $Q_{10}^{(T_a - T_{ref})/10}$, where $Q_{10}$ is a

temperature sensitivity coefficient, $T_a$ is air temperature, and $T_{ref}$ is a reference temperature. For the maintenance respiration cost for live fine roots, soil temperature at the respective soil layer ($T_{s,i}$) is used instead of $T_a$. Similarly, the effect of temperature on decomposition (and therefore on heterotrophic respiration) is also calculated via a $Q_{10}$ formulation, where the base rates of decomposition are multiplied by $Q_{10}^{(T_{s,i} - T_{ref})/10}$. In CLM, a default $Q_{10}$ of 1.5 is used for both maintenance respiration and decomposition. However, nighttime $CO_2$ flux measurements above the canopy at Wind River, which would

include the sum of autotrophic and heterotrophic respiration, indicate a $Q_{10}$ of 2.49 (Misson et al., 2007). By adjusting CLM's $Q_{10}$ to 2.5 for both maintenance respiration and decomposition, the seasonal behavior of ecosystem respiration better corresponded with observed values. This was especially the case for heterotrophic respiration, reducing the model overestimation during winter and the model underestimation during summer.

## A7  Soil Hydraulic Properties

Initial runs indicated poor performance of CLM in the simulation of soil water content at the Wind River site (strong dry bias), which resulted in an unrealistic down-regulation of GPP due to soil moisture stress late in the dry, summer season. When using the original soil hydraulic properties from CLM v.4.0 the results were greatly improved, with a wetter soil and a reduction of the unrealistic soil moisture stress. The observed improvement was likely related to a smaller subsurface runoff in CLM v.4.0 and consequently greater water retention in the soil. In CLM, subsurface runoff is proportional to a term representing the maximum drainage when the water table depth is at the surface ($q_{drai}^{max}$). In CLM v.4.0, $q_{drai}^{max} = 0.0055$ kg m$^{-2}$ s$^{-1}$, while in CLM v.4.5 $q_{drai}^{max} = 10\sin\beta$ kg m$^{-2}$ s$^{-1}$, where $\beta$ is the mean grid cell topographic slope. Even for a small $1°$ slope, $q_{drai}^{max}$ is significantly larger than in CLM v.4.0 (0.1745 kg m$^{-2}$ s$^{-1}$). The soil hydraulic properties from CLM v.4.0 were therefore used in this study.

## A8  Root Distribution

In CLM, root distribution over soil depth is calculated as in Eq. (4). Root fraction ($r_i$) in combination with a plant wilting factor ($w_i$, Eq. 5) for each soil layer $i$ are used to calculate an integrated soil moisture stress parameter in CLM, $\beta_t$ (Eq. 3), which downregulates stomatal conductance in the model (Eq. 2).

Shaw et al. (2004) provides a good description of rooting depth at Wind River: "Plant roots are concentrated above 50 cm in soil profiles; however, roots as deep as 2.05 m have been observed in younger forests growing on nearly identical soils (T. Hinckley personal communication). Many coarse roots of Douglas-fir extend to depths greater than 1.0 m. Tip-up mounds of windthrown western hemlock trees typically have a classic flat root plate indicative of shallow rooting" (Douglas-fir and western hemlock are the dominant species at the site). With the default NETT PFT root distribution parameters in Eq. (4) ($r_a = 7$ m$^{-1}$ and $r_b = 2$ m$^{-1}$), the total root fraction in the top 46 and 130 cm of soil is 78% and 96%, respectively (note the small fraction of roots at depths below 1.3 m (4%)). The above site description (Shaw et al. 2004) suggests that the default parameters are inadequate at Wind River, resulting in a "too-shallow" rooting profile.

In this study the NETT PFT $r_b$ parameter was changed to 1 m$^{-1}$ (default CLM value for broadleaf evergreen temperate tree PFT), shifting roots towards deeper soil layers, in order to make water stored at deeper soil layers available to the trees and, along with the changes in the soil hydraulic properties discussed in Appendix A7, reduce the excessive late-summer soil moisture stress and downregulation of GPP in the model. With the adjusted $r_b$ parameter, the total root fraction in the top 46 and 130 cm of soil is 67% and 86%, respectively (14% below 1.3 m), which seems more reasonable based on Shaw et al. (2004) and the fact that Douglas-fir trees at the site are about 500 years old and 40–65-m tall. The adjustment of soil moisture stress in CLM via root distribution was therefore physically justified.

The plant wilting factor, $w_i$, offers an additional path for adjustment of the simulated soil moisture stress, but it was not investigated in this study.

## A9 Stomatal Conductance

In CLM, leaf stomatal conductance is calculated based on the Ball-Berry model as described by Collatz et al. (1991) and implemented by Sellers et al. (1996) in the SiB2 model (see Eq. 2). The default values set for the parameters $m_{bb}$ and $b_{bb}$ in CLM for $C_3$ plants (9 and 10 mmol m$^{-2}$ leaf s$^{-1}$, respectively) were found to be inadequate at Wind River, leading to a significant overestimation of latent heat fluxes due to excessive plant transpiration (after the adjustments discussed in the aforementioned Sections which resulted in higher forest productivity). These default parameter values were established based on the values used in the SiB2 model (Sellers et al., 1996). In SiB2, however, a distinction was made for coniferous forests ($m_{bb} = 6$) but was not carried over to CLM. Observations reported in the literature support this lower $m_{bb}$ value for conifers (see for example the survey by Williams et al., 2004, Table 6.3, and Miner et al., 2017, Fig. 1). On the other hand, $b_{bb}$ values reported in the literature are highly variable (1–400 mmol m$^{-2}$ leaf s$^{-1}$ in the survey by Barnard and Bauerle, 2013 for a broad range of plant species). In CLM v.4.0, the default $b_{bb}$ for $C_3$ plants is significantly smaller than in CLM v.4.5 (2 vs. 10 mmol m$^{-2}$ leaf s$^{-1}$) (Oleson et al., 2010). $m_{bb} = 6$ and $b_{bb} = 5$ mmol m$^{-2}$ leaf s$^{-1}$ were found to greatly improve the modeled latent heat fluxes at the Wind River site, and were therefore adopted in this study. The updated values also resulted in a great improvement in the simulation of $\delta^{13}$C of leaves, SOM, and GPP. Figures A1 and A2 illustrate the impact of the stomatal conductance parameters on model performance, particularly in regards to latent heat fluxes and photosynthetic $^{13}$C discrimination.

It is important to highlight that the default nitrogen limitation scheme was used in the simulations. As discussed in Sect. 2.1, this scheme makes CLM a partially-coupled model in respect to net leaf photosynthesis and stomatal conductance: while the actual GPP is down-regulated in response to nitrogen availability, stomatal conductance remains consistent with potential net leaf photosynthesis ($A_n$). With this structure, CLM is expected to overestimate plant transpiration and photosynthetic $^{13}$C discrimination. The above discussed calibration of the Ball-Berry stomatal conductance parameters, especially the significant reduction of $m_{bb}$ from 9 to 6, must also have compensated for this structural issue within the model. Note that nitrogen down-regulation is significant at Wind River, peaking at ~0.25 (GPP/GPP$_{pot}$ = 0.75) in May (Fig. A3).

When using the default nitrogen limitation scheme in CLM, the modeled $^{13}$C discrimination values reported by Raczka et al. (2016) for the Niwot Ridge AmeriFlux site (also a coniferous forest site) were significantly overestimated, i.e., $\delta^{13}$C values of GPP and biomass were significantly smaller than observations. To improve the simulation, Raczka et al. (2016) removed the post-photosynthetic nitrogen down-regulation of $A_n$ and GPP$_{pot}$ ($d = 0$; see Eq. 9) and included a foliar nitrogen-limiting factor in the calculation of $V_{cmax25}$, making the model fully coupled in respect to net leaf photosynthesis and stomatal conductance. With this configuration, their simulation of $^{13}$C discrimination improved significantly, but the values still presented a small overestimation in respect to the site observations. According to Raczka et al. (2016), overestimation of $g_s$ due to an inadequate $m_{bb}$ value (too high) could be a reason for the mismatch (they used the default value of 9 in their simulation).

The alternative nitrogen limitation scheme (via $V_{cmax25}$ down-regulation, as in Raczka et al., 2016) was also investigated here. The simulation of LE, GPP, and $^{13}$C discrimination when using this configuration and the default $m_{bb}$ value of 9 was found to be similar to the results when using the default nitrogen limitation scheme and $m_{bb} = 6$ (Fig. A4). The results in Fig. A4 indicate that the calibration of $m_{bb}$ from 9 to 6 represents a tradeoff with the approach to nutrient limitation, compensating for elevated, nitrogen-unlimited (potential) net leaf photosynthesis used in the calculation of $g_s$.

## A10  CLM Performance: Default vs. Calibrated Parameters

In order to illustrate the effect of altering the model parameters discussed in this Appendix (see summary of changes in Table 1), Figs. A5 and A6 compare the performance of CLM for key model outputs when using "out-of-the-box" parameters and calibrated parameters. Note the significant improvement in the simulation of LAI, biomass, and $CO_2/H_2O$ fluxes.

## Acknowledgments

This research was supported by the U.S. Department of Energy's Office of Science, Terrestrial Ecosystem Science Program, under award number DE-SC0010624. BMR and DRB were supported by the U.S. Department of Energy's Office of Science, Terrestrial Ecosystem Science Program, under award number DE-SC0010625. PET was supported by the U.S. Department of Energy's Office of Science, Biological and Environmental Research, Accelerated Climate Modeling for Energy project. We acknowledge the Wind River Field Station AmeriFlux site (US-Wrc, PIs: Kenneth Bible, Sonia Wharton) for its data records. Funding for AmeriFlux data resources was provided by the U.S. Department of Energy's Office of Science. Data and logistical support were also provided by the U.S. Forest Service Pacific Northwest Research Station. We gratefully acknowledge the comments and suggestions of the editor and three anonymous reviewers, which have greatly improved our paper.

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

**Table 1.** Summary of changes in CLM parameters during the calibration process. The parameters listed, excluding $M$, $Q_{10}$, $m_{bb}$, $b_{bb}$, and soil hydraulic parameters, correspond to the needleleaf evergreen temperate tree plant functional type (NETT PFT).

| Parameter | Description | CLM name | Default CLM value | Calibrated CLM value |
|---|---|---|---|---|
| $A_{s:l}$ | New stem C: new leaf C ratio (gC gC$^{-1}$) | *stem_leaf* | Dynamic | 2 |
| $A_{fr:l}$ | New fine root C: new leaf C ratio (gC gC$^{-1}$) | *froot_leaf* | 1 | 0.385 |
| $CN_l$ | Leaf C:N ratio (gC gN$^{-1}$) | *leafcn* | 35 | 38.2 |
| $CN_{llit}$ | Leaf litter C:N ratio (gC gN$^{-1}$) | *lflitcn* | 70 | 76.4 |
| $CN_{fr}$ | Fine root C:N ratio (gC gN$^{-1}$) | *frootcn* | 42 | 64.7 |
| $\tau_l$ | Leaf longevity (yr) | *leaf_long* | 3 | 5 |
| $r_b$ | Root distribution parameter (m$^{-1}$) | *rootb_par* | 2 | 1 |
| SLA$_0$ | Specific leaf area at canopy top (m$^2$ leaf gC$^{-1}$) | *slatop* | 0.010 | 0.006 |
| $m$ | SLA($x$) slope (m$^2$ ground gC$^{-1}$) | *dsladlai* | 0.00125 | 0.000985 |
| $M$ | Plant mortality rate (% yr$^{-1}$) | *am* | 2 | 1.5 |
| $Q_{10}$ | Temperature sensitivity coefficient of maintenance respiration and decomposition (–) | *q10* | 1.5 | 2.5 |
| Soil hydraulic parameters | Version used | *origflag* (namelist variable) | 0 (CLM4.5) | 1 (CLM4.0) |
| $m_{bb}$ | Ball-Berry Eq. slope (–) | *mbbopt* | 9 | 6 |
| $b_{bb}$ | Ball-Berry Eq. intercept (µmol m$^{-2}$ leaf s$^{-1}$) | *bbbopt* | 10000 | 5000 |

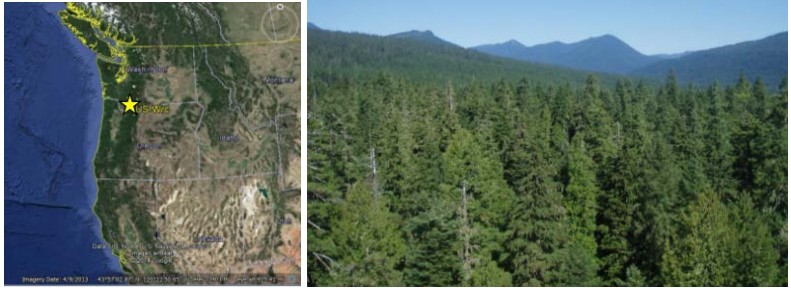

**Figure 1.** Location and view of the Wind River AmeriFlux site, US-Wrc (satellite image from Google Earth).

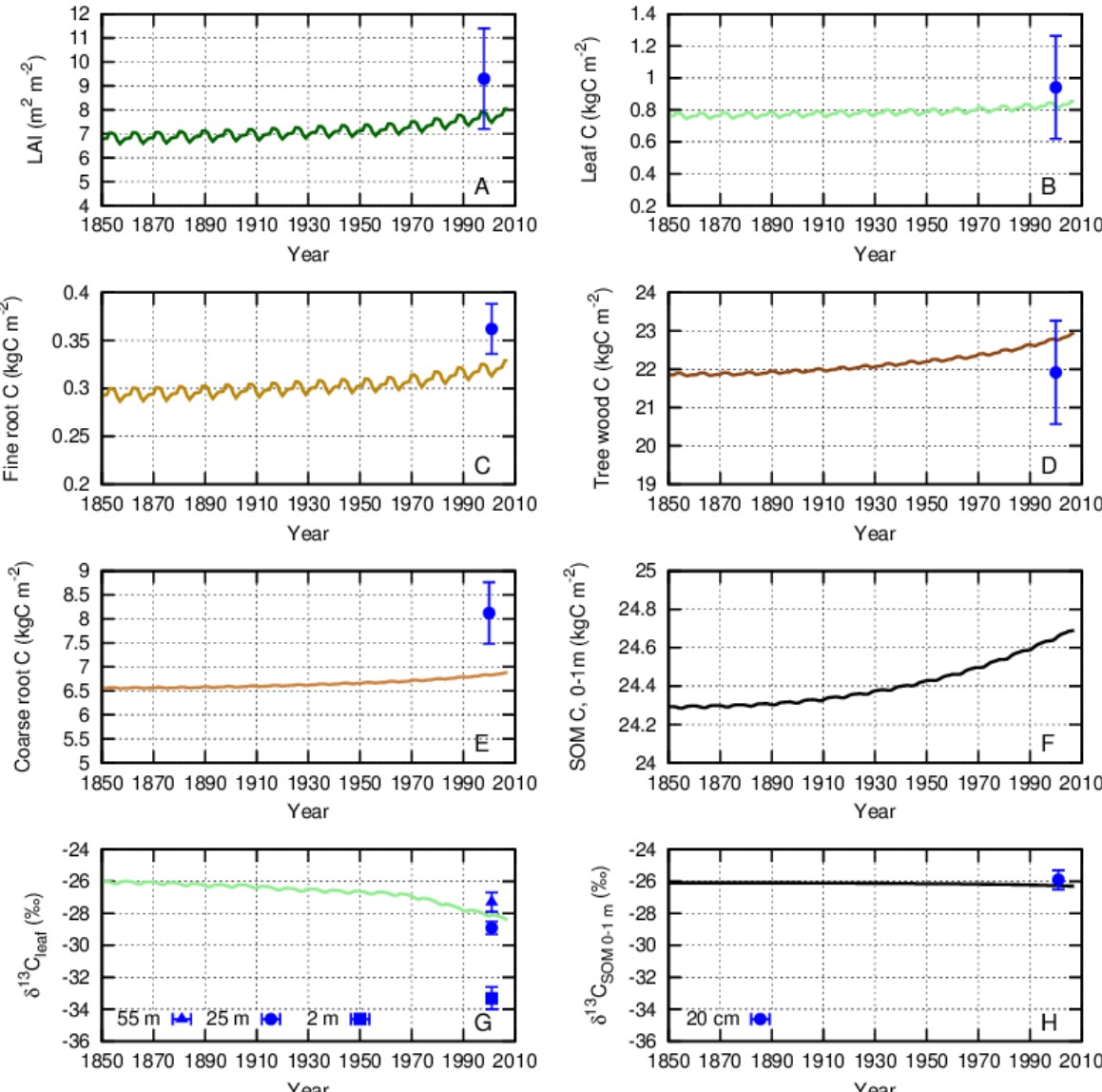

**Figure 2.** Modeled leaf area index (A), carbon stocks (B–F), and $\delta^{13}C$ of leaf (G) and soil organic matter (H) during the transient run (lines) compared against site observations (points and error bars). Modeled values in panels A–F correspond to annual averages. Modeled $\delta^{13}C$ values in panels G and H were calculated from annual averages of the respective $^{13}C$ and $^{12}C$ pools. Observations in panels A–E (average ± std. dev.) are from the AmeriFlux database (based on Thomas and Winner, 2000 and Harmon et al., 2004). Observations in panels G and H correspond to the average ± std. dev. of the measurements reported by Fessenden and Ehleringer (2003) in their Figs. 2b and 3 (leaf $\delta^{13}C$ at canopy top (55 m), middle (25 m), and bottom (2 m) and SOM $\delta^{13}C$ at 20 cm depth).

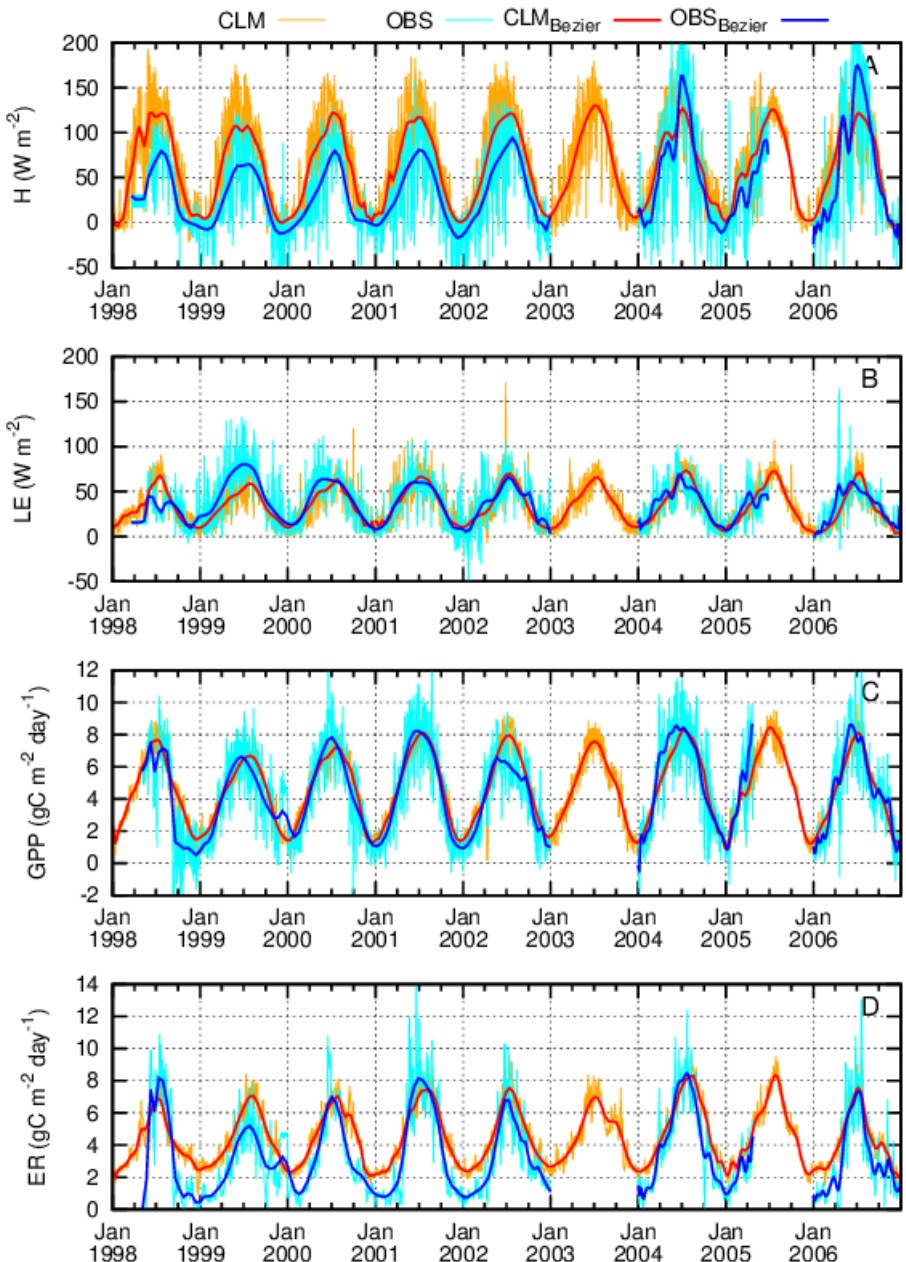

**Figure 3.** Modeled sensible heat flux (A), latent heat flux (B), gross primary production (C), and ecosystem respiration (D) vs. site observations. Orange/red and cyan/blue lines correspond to modeled and observed values, respectively. For a clearer visualization, the daily averages (thin lines) were smoothed with a Bézier algorithm (thick lines).

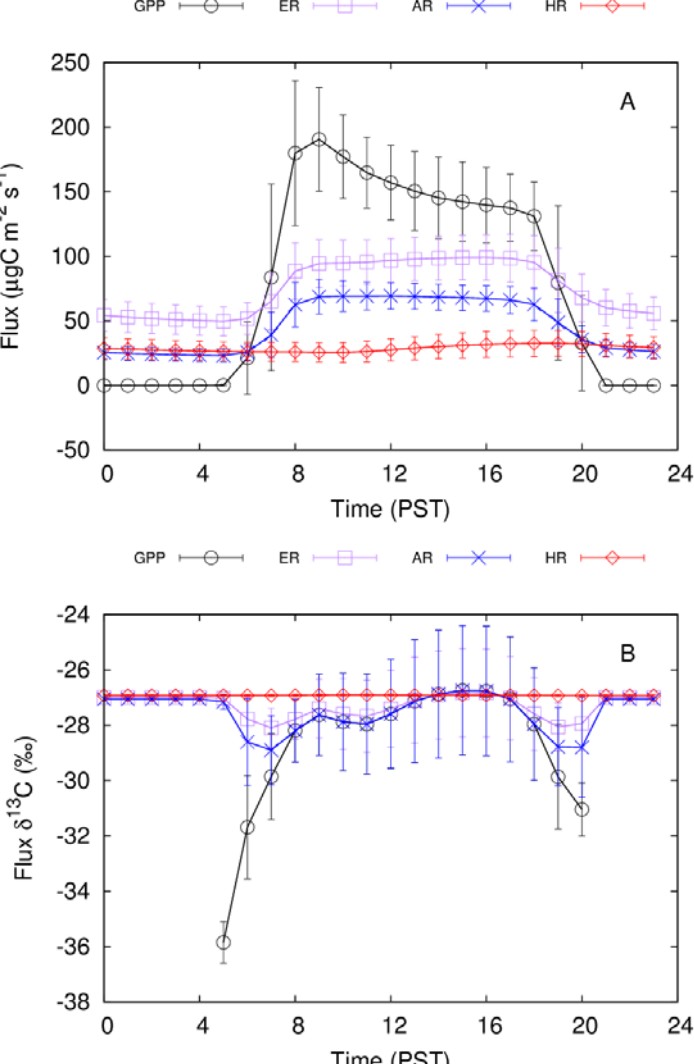

**Figure 4.** Mean diurnal cycle of modeled carbon fluxes (A) and their respective carbon isotope ratios (B) for the summer months (June–September) of years 1998–2006. Fluxes include gross primary production (black circles), ecosystem respiration (purple squares), autotrophic respiration (blue crosses), and heterotrophic respiration (red diamonds). Bars correspond to ±1 standard deviation.

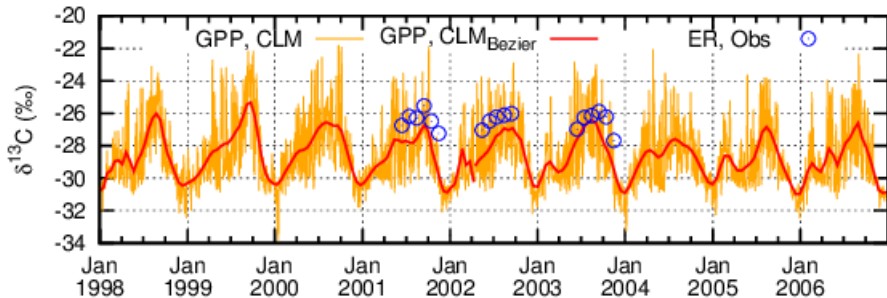

**Figure 5.** Modeled $\delta^{13}C$ of gross primary production (lines) and observed $\delta^{13}C$ of ecosystem respiration (circles). Thin orange line corresponds to daily averages using 10:00–16:00 data only. For a clearer visualization, this curve was smoothed with a Bézier algorithm (thick red line). Blue circles correspond to site observations (monthly averages) reported by Lai et al. (2005).

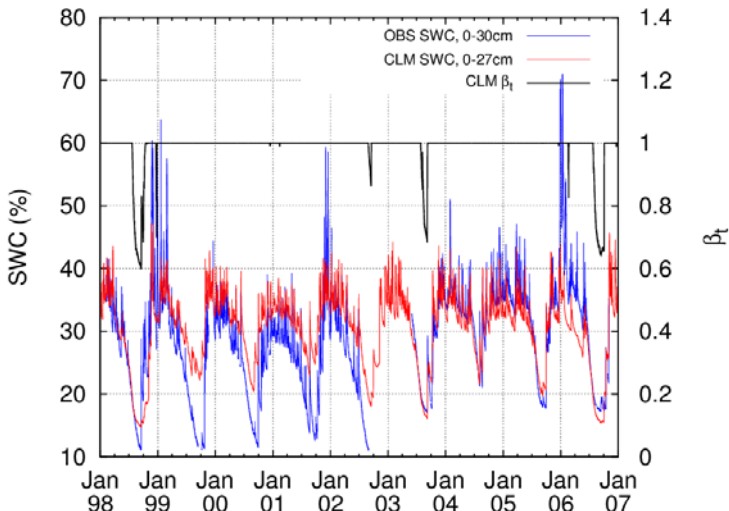

**Figure 6.** Hourly soil water content and CLM's soil moisture stress parameter, $\beta_t$ (black line). Observed SWC (blue line) corresponds to the integrated value for the top 30 cm of soil, while modeled SWC (red line) corresponds to the integrated value for the top 5 soil layers in CLM (0–27 cm). At Wind River, SWC at permanent wilting point and at field capacity is 14% and 30%, respectively (Wharton et al., 2009).

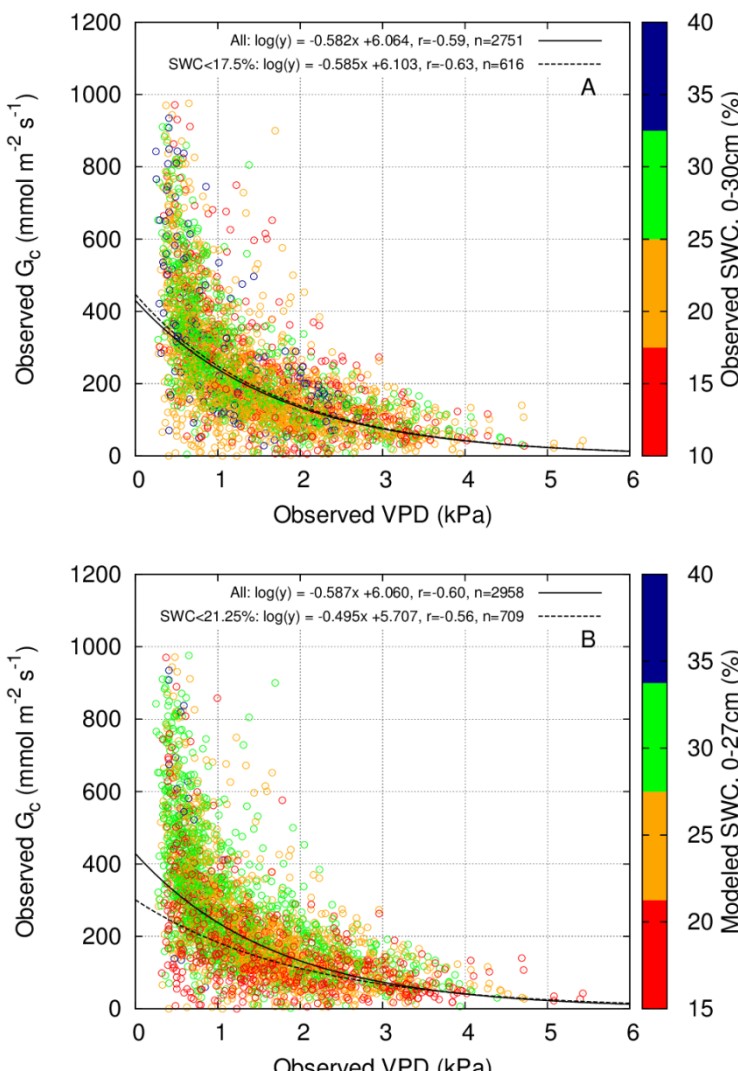

**Figure 7.** Hourly observed canopy conductance vs. observed VPD for the summer months (June–September) of years 1999–2006, restricted to 10:00–16:00 PST (additional restrictions were imposed to the calculation of $G_c$, see Sect. 2.7). Years 1998 and 2005 were not included due to missing data. Data points were segregated according to observed SWC in panel A and according to modeled SWC in panel B (see Fig. 6). Lines correspond to the linear regression between $\log G_c$ and VPD using all data points (solid lines) and using only points within the lowest SWC bin (red circles, dashed lines).

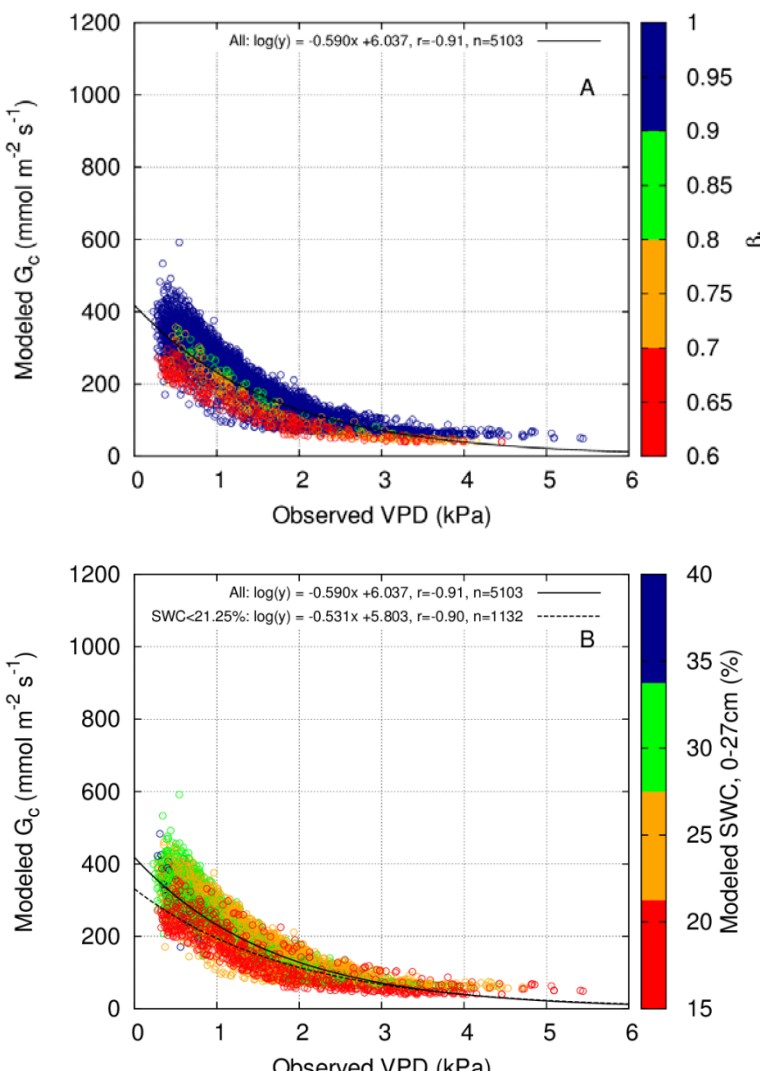

**Figure 8.** Hourly modeled canopy conductance vs. observed VPD for the summer months (June–September) of years 1999–2006, restricted to 10:00–16:00 PST (additional restrictions were imposed to the calculation of $G_c$, see Sect. 2.7). Note that observed air temperature and relative humidity were used to drive CLM. Years 1998 and 2005 were not included for consistency with Fig. 7. Data points were segregated according to the soil moisture stress parameter $\beta_t$ in panel A and according to modeled SWC in panel B (see Fig. 6). Lines correspond to the linear regression between $\log G_c$ and VPD using all data points (solid lines) and using only points within the lowest SWC bin (red circles, dashed line).

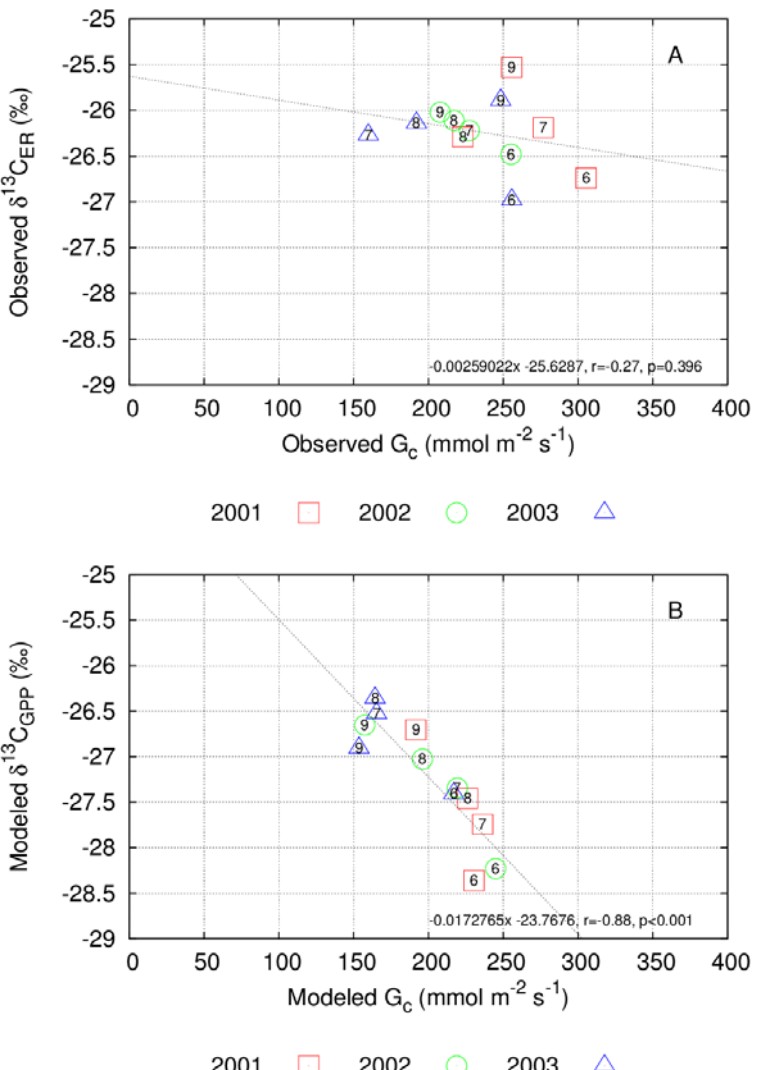

**Figure 9.** Observed $\delta^{13}C$ of ecosystem respiration vs. observed canopy conductance (A) and modeled $\delta^{13}C$ of gross primary production vs. modeled canopy conductance (B) for the summer months of 2001–2003. Except for the observed $\delta^{13}C_{ER}$, data points correspond to monthly averages of daytime (10:00–16:00) data (additional restrictions were imposed to the calculation of $G_c$, see Sect. 2.7). Observed $\delta^{13}C_{ER}$ corresponds to the monthly averages reported by Lai et al. (2005). Numbers at the center of each point indicate the month.

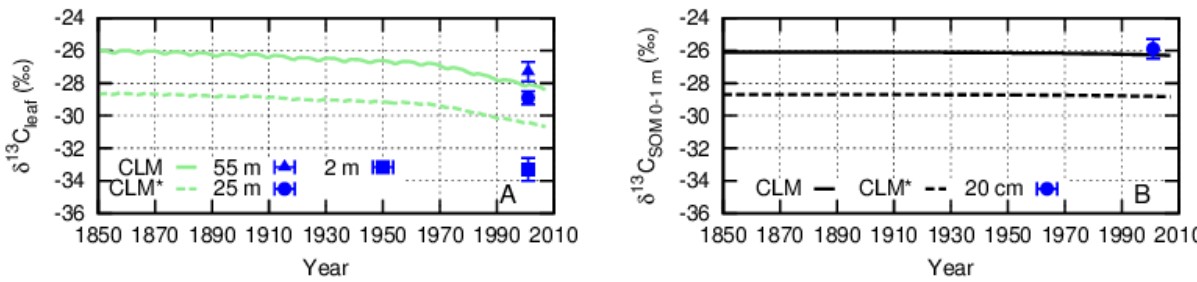

**Figure A1.** Modeled $\delta^{13}$C of leaf (A) and soil organic matter (B), calculated from annual averages of the respective $^{13}$C and $^{12}$C pools during the transient run (lines). Results from two model configurations are presented: CLM (calibrated model, solid lines) and CLM\* (calibrated model using the default stomatal conductance parameters ($m_{bb}$ and $b_{bb}$; see Table 1), dashed lines). Site observations (average ± std. dev., blue points and error bars) are also shown (see caption of Fig. 2 for details).

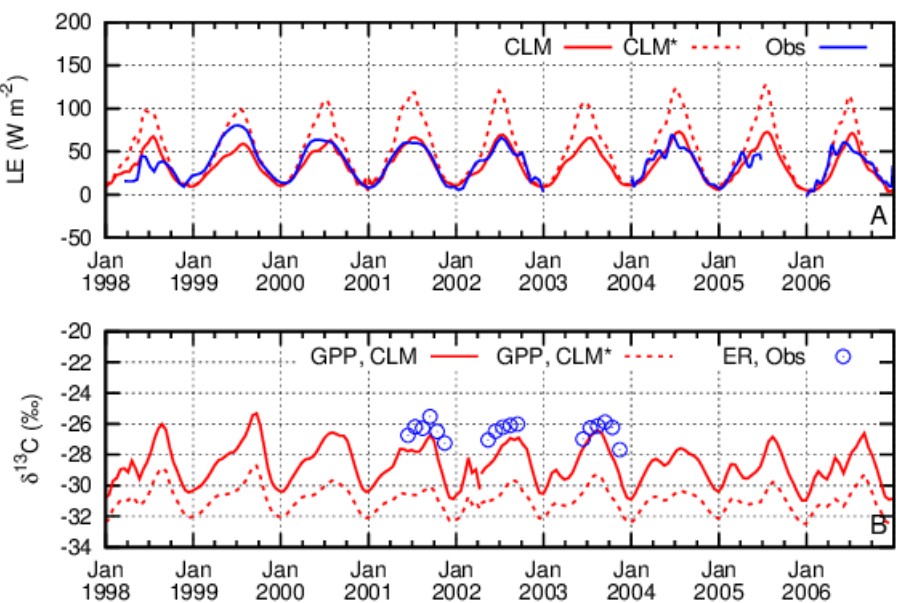

**Figure A2.** Modeled latent heat flux (A) and $\delta^{13}C$ of gross primary production (B, lines) for 1998–2006. The curves presented correspond to Bézier-smoothed daily averages as in Figs. 3 and 5. Results from two model runs are presented: CLM (calibrated model, solid red lines), and CLM* (calibrated model using the default stomatal conductance parameters ($m_{bb}$ and $b_{bb}$; see Table 1), dashed red lines). The blue line and circles correspond to site observations. The circles in panel B are the monthly averages of $\delta^{13}C_{ER}$ reported by Lai et al. (2005).

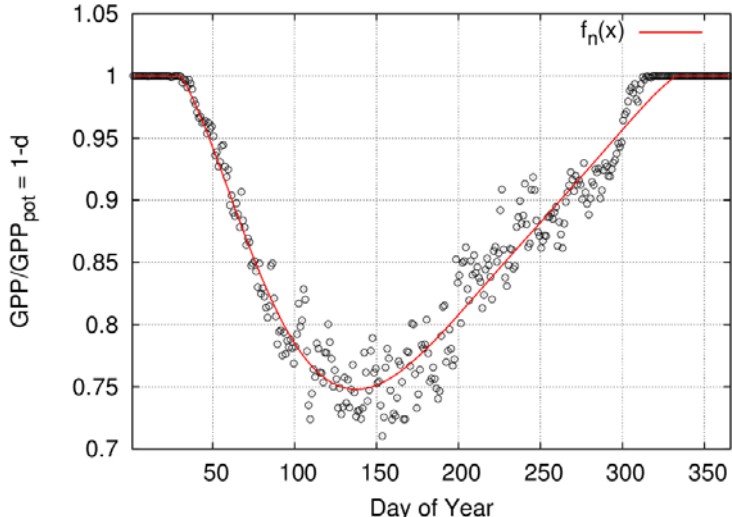

**Figure A3.** Modeled fraction of potential GPP (GPP/GPP$_{\text{pot}}$). Data points correspond to daily means averaged over 1850–2006 (calibrated CLM simulation). The fitted curve is $f_n(x = \text{day of year}) = -1.39697\times10^{-14}x^6 +1.71948\times10^{-11}x^5 -8.26883\times10^{-9}x^4 +1.90682\times10^{-6}x^3 -1.97639\times10^{-4}x^2 +0.0055728x +0.966272$ for $31 \leq x \leq 332$ and $f_n(x) = 1$ elsewhere. Note

5   that GPP/GPP$_{\text{pot}} = 1-d$, where $d$ is the nitrogen down-regulation factor as defined in Eq. (9) within text.

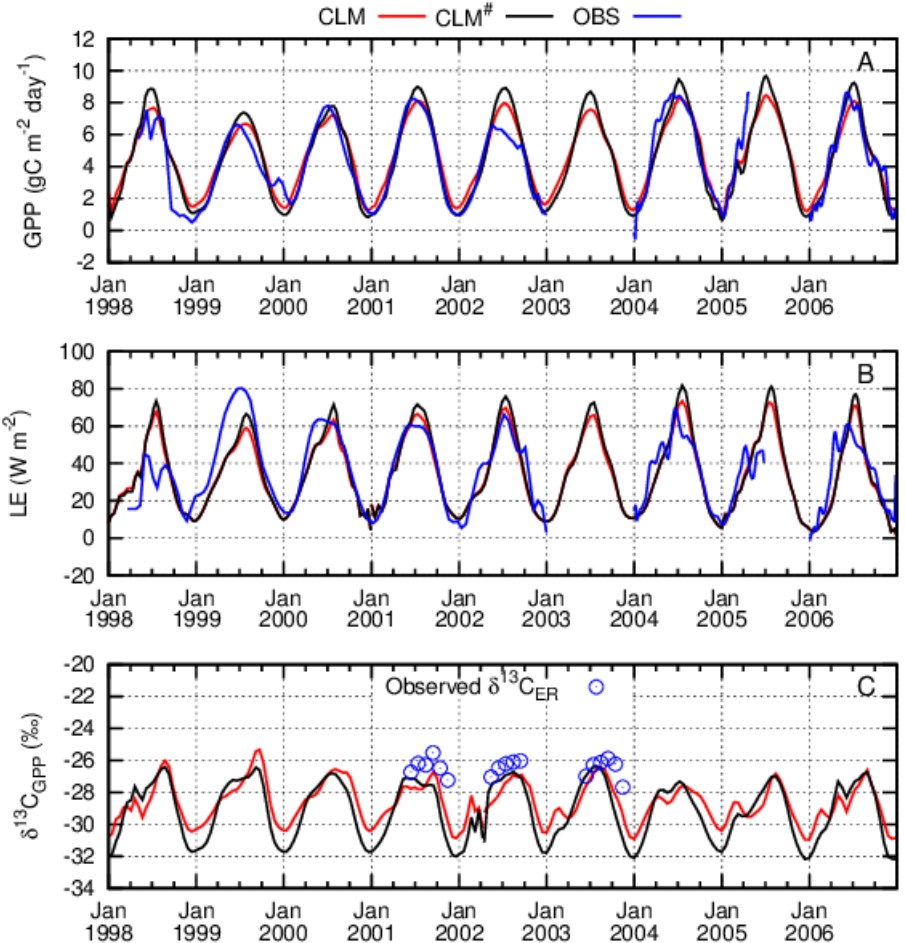

**Figure A4.** Modeled gross primary production (A), latent heat flux (B), and $\delta^{13}$C of gross primary production (C) for 1998–2006. The curves presented correspond to Bézier-smoothed daily averages as in Figs. 3 and 5. Results from two model runs are presented: CLM (calibrated model, red lines), and CLM$^{\#}$ (calibrated model using $m_{bb} = 9$ and the alternative nitrogen limitation scheme discussed in Appendix A9, black lines). In CLM$^{\#}$, $V_{cmax25}$ was multiplied by a seasonally-varying nitrogen down-regulation factor, calculated based on the mean (1850–2006) seasonal cycle of GPP/GPP$_{pot} = 1 - d$ in the CLM run ($f_n(x)$ in Fig. A3) subtracted by 0.35 (manual adjustment applied to avoid excessive productivity during the transient simulation). Blue lines and circles correspond to site observations. The circles in panel C are the monthly averages of $\delta^{13}$C$_{ER}$ reported by Lai et al. (2005).

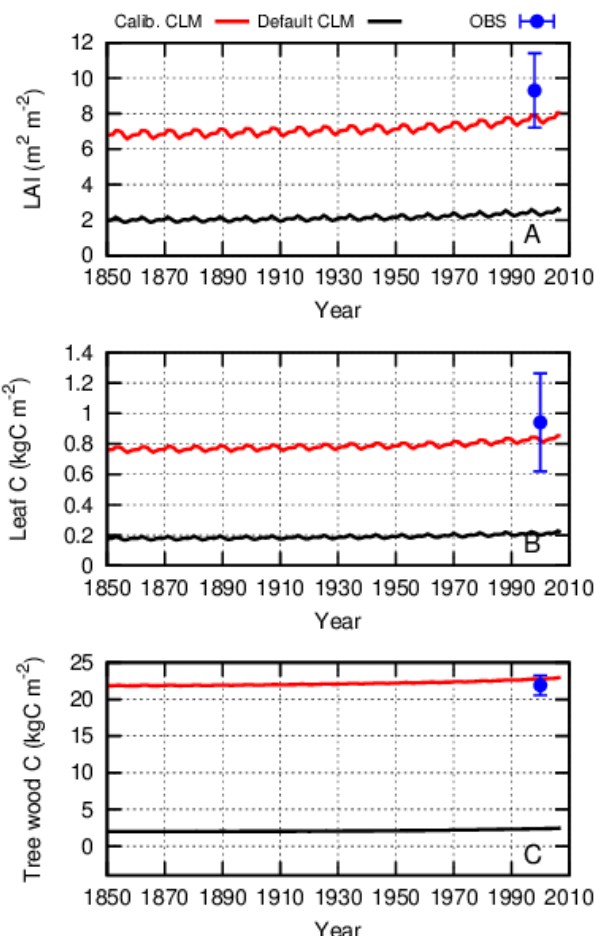

**Figure A5.** Comparison of CLM performance at Wind River when using default, "out-of-the-box" parameters (black lines) and calibrated parameters (red lines). Modeled values correspond to annual averages. Observations (average ± std. dev., blue points and error bars) are from the AmeriFlux database (based on Thomas and Winner, 2000 and Harmon et al., 2004).

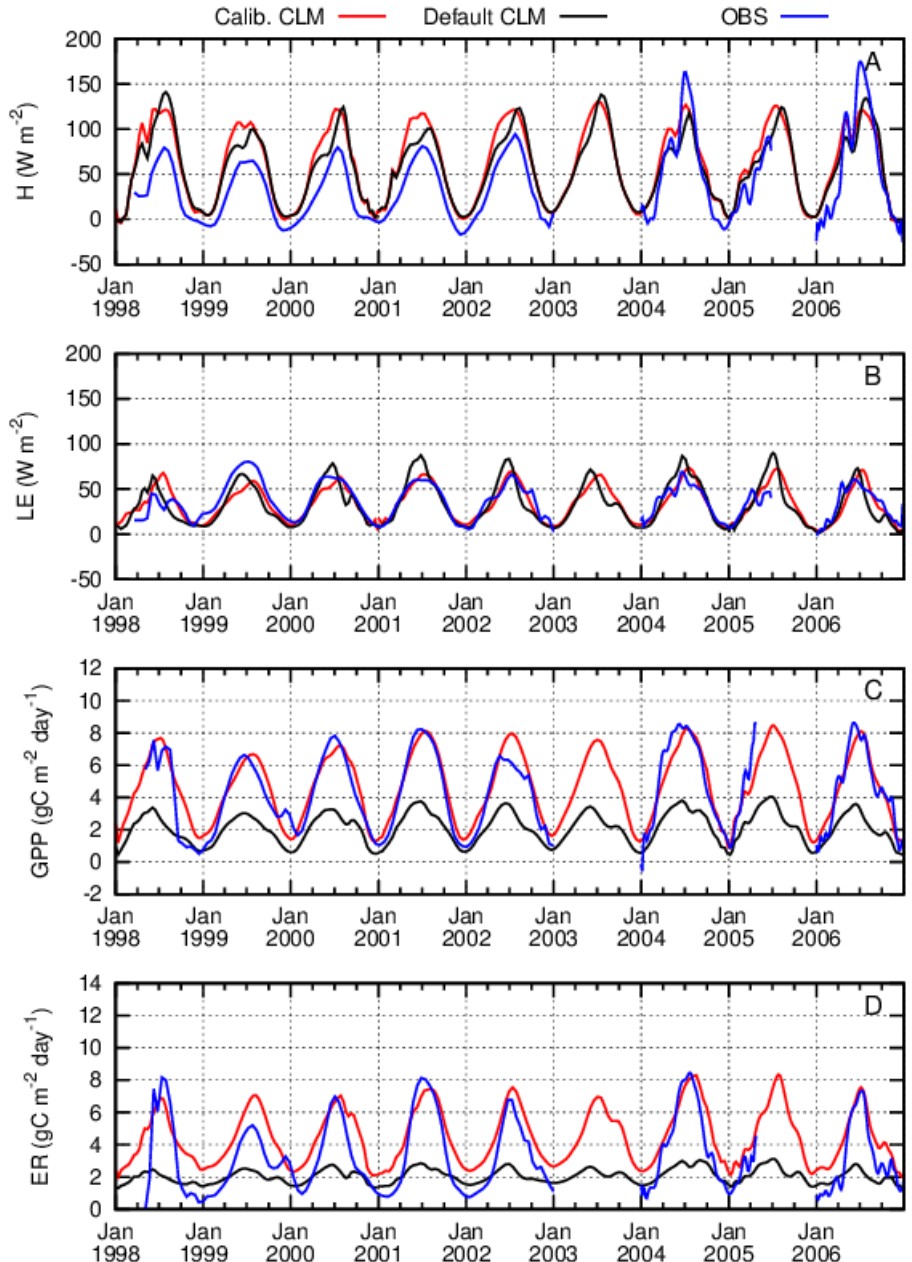

**Figure A6.** Comparison of CLM performance at Wind River when using default, "out-of-the-box" parameters (black lines) and calibrated parameters (red lines). Observations (blue lines) are from the AmeriFlux database. For a clearer visualization, the data presented correspond to Bézier-smoothed daily averages as in Fig. 3.

