# Peer review of "Evaluating the Community Land Model (CLM 4.5) at a Coniferous Forest Site in Northwestern United States Using Flux and Carbon-Isotope Measurements"

_Biogeosciences, 2016_

## Referee Comment (RC1) · Anonymous Referee #1 · 16 Nov 2016

This paper describes an application of the CLM model to the Wind River flux tower site.

The first point to note is that model parameterization is done in the good old-fashioned way. When applied out of the box with standard PFT parameters, the model does not fit very well. Hence, the model needs to be calibrated. This calibration is done by adjusting parameter values manually, based on the literature and some trial-and-error, until the fit to the data is not too bad. Most groups are moving away from this approach to parameterization, to a more rigorous statistical framework such as Bayesian calibration, which yields more defensible parameter values. I don't think it's essential that the

authors do this, but it would be good if they could give at least some justification for sticking with the traditional method of parameterization.

The second striking thing about this manuscript was the almost complete lack of reference to the literature in the Discussion. The Results and Discussion are combined into the one section – never a good idea in my view. Here, there is almost no discussion of the results, and no attempt to place the results in the context of the literature. Overall, I came away with a strong "so what" feeling: the authors do not do a good job of articulating why they want to calibrate CLM for this site, nor what we get out of it. There is little in the Introduction to motivate the study, and nothing in the conclusions about how this work advances the field in general. I very strongly suggest that the authors - Better motivate the study in the Introduction, with an expectation of the kinds of questions that this work can address - Separate the Results from the Discussion - Focus the Discussion on what we learn from this study, and ensure that it is placed in the broader context of the literature with appropriate citations.

Some comments on the methods:

The drought stress factor should be more clearly defined: I'd like to see the equation for the plant wilting factor, which apparently depends on both soil water potential (state variable) and the plant dependent response to water stress.

I don't understand the use of the factor 'd' in equations 5 & 6. As I understand it, the relationship $A = gs/1.6 (Cs – Ci)$ is a physical description of the diffusion process through the stomata. How can this be modified by nitrogen limitation? Or is this something that affects the "isotopic" $Ci/Ca$ only?

I note that mesophyll conductance also affects the isotopic ratio – is this accounted for in this model?

Please add a description of how the model scales from leaf to canopy. As all of the comparisons are with canopy-scale GPP, LE and Gs, it is important for the reader to

know the principal assumptions underlying this scaling. How is leaf isotopic composition modeled for the whole canopy? How is leaf conductance scaled to the canopy?

The model is evaluated against gap-filled flux data. In my view that's not acceptable: evaluating a model against gap-filled data is comparing one model against another. The model should only be evaluated against non-gap-filled data.

Please describe more clearly the process used for calibration. For example, p16 says that SLA0 is optimized by aiming to minimize model errors in site observations of LAI and CI – was this done using a solver function, or simply by manual trial and error?

On the results:

Figure 2 could show observations as well as model output, making it easier to visualize the model-data correspondence. Please indicate in Table 2 what the errors refer to (+/- SE? 95% CI? Range?)

I was unsure how to evaluate the leaf isotopic data. Are the modeled values to be compared with the top, bottom, or average of the canopy? See note above about how isotope discrimination is scaled to the whole canopy.

It would have been good to see the model performance with the parameters out of the box, as well as model performance with calibrated parameters, in order to visualize the effect of altering model parameters.

Please discuss the lack of energy balance closure at this site. The model assumes the energy balance is closed; if the data show a lack of closure the model must show a bias in its predictions of either LE or H. How large is the lack of closure at this site, and how does it affect the model comparison to data?

What is the average rooting depth? The SWC data shown are only to 30 cm – how much deeper than this do the roots penetrate? Is the lack of response to low SWC a function of only considering the very top soil? The demonstration that the model over-estimates the effect of low SWC in the topsoil is interesting, but difficult to interpret without the rooting depth and the formulation for soil moisture stress being given. Nothing is said about how the model might be improved based on this observation – it would be good if the authors could identify the root cause for this mismatch and suggest how it could be addressed.

Finally, a very important point, something that needs to be said: why on earth is CLM still using the Ball-Berry stomatal model? The Ball-Berry model is physiologically incorrect (see Aphalo & Jarvis 1991, and pretty much every stomatal physiology paper since). It is consistently outperformed by models based on VPD. It will incorrectly predict stomatal behaviour in future climates, when VPD is predicted to change but RH stay the same (see Sato et al. 2015 JGR). It is quite odd to read here the justification that "such improvement is expected to be small". I think it is well past time that CLM moved on from the Ball-Berry model.

Very finally, debating whether to mention this or not, but: I was also struck by the extreme gender imbalance of the authorship list. Ten male authors! I hope and trust that the PIs here are actively taking steps to address this imbalance in their group of collaborators.

---

## Referee Comment (RC2) · Anonymous Referee #2 · 3 Jan 2017

Review: Duarte et al., Evaluating the Community Land Model (CLM4.5) at a coniferous forest site in northwester United States using flux and carbon-isotope measurements.

This manuscript (MS) presented a difficult review. After the first couple of readings, I found the paper easy to read, with no obvious shortcomings. This should have prompted a favorable review, right? However, the MS does read somewhat like 'we ran a model, here is what happened' which is bothersome to me. The use of isotopes in a model is no longer novel; you need to go beyond just saying you put isotopes in a model and ran it, and say something about how this capability and our inclusion of it in models informs our understanding of natural systems. With more careful reading, more issues emerged for me.

1. The first paragraph talks about drought, and drought stress is mentioned throughout the MS. However, drought is an *anomalous* reduction in precipitation from climatological values. If, climatologically, 5% of annual precipitation occurs in JJA, then a dry summer is just…normal. Drought in this part of the country will likely manifest under climate change as a longer growing season and higher summertime temperatures and VPD when compared to the present-day. If meteorological forcing from the period 1998-2006 is cycled multiple times to simulate the period 1850-present, you're not going to capture any slow secular changes. Really, what's being done here is an evaluation of CLM's ability to capture a *mean seasonal cycle* that includes a dry summer. This site has a 50+ meter tall canopy, and fluxes (LE/H) are in phase with Boreal summer. LE fluxes appear to peak in June or July, and H peaks one month or so later, from a quick inspection of Figure 3. That suggests an ecosystem that may experience some stress in late summer, but is not water-limited on the whole. Costa et al. (2010) and da Rocha et al. (2009) discuss the notion of environmental and physiological stress, albeit in Amazonian forests. At Wind River, I suspect an ecosystem that is environmentally-limited for large parts of the year (winter, obviously) and may experience some physiological stress in late summer. There may be some value in studying interannual variability (IAV), as observed GPP in 2002 (a known western drought that year) and 2006 both taper off rather quickly following an early peak. CLM does not capture this behavior in 2002, but does in 2006. But the authors do not discuss IAV, but rather concentrate on the idea of annual drought stress. I'm not buying it; you're not going to have a 50-meter canopy in a region that is water stressed. Furthermore, $\beta t$ drops below 1 in less than half of the years, and then only to a minimum of 0.6. That's not a whole lot of stress. I strenuously object to the use of the word drought for a dry summer that occurs every year.

2. As I think more and more about the paper, the notion of equifinality (multiple solutions in parameter or process space that result in a single

model outcome) keeps arising. A strength of CLM is that there is a large community of researchers banging on the model, and just about any process that one can imagine is included in the model and can be turned on or off (sunlit and shaded leaves; explicit nutrient cycles; hydraulic redistribution; multiple hydrology schemes; diagnostic canopy vs. DGVM; etc., etc.). This is also a CLM weakness. There are so many knobs to turn, how can you really be sure you are turning the right ones? With this in mind, I returned to Raczka et al. (2016, hereafter R16), which simulated Niwot Ridge (NR1, another site with similar vegetation type, but dissimilar climate). Since Raczka is the second author on this MS, and Duarte is the second author on R16, these papers offer an opportunity to present a body of work that compares and contrasts CLM behavior at two sites that have a similar dominant PFT but dissimilar climate (and soil, I presume). Imagine my surprise when I find that R16 claims that the nitrogen limitation scheme is a critical component, but is not considered in sensitivity tests in this MS! R16 is happy using CLM4.5 hydrology, while the current MS states that CLM4.0 hydrology was necessary to capture realistic behavior. I was encouraged to see that both papers suggest a change from standard BB C3 slope of 9 to 6 in the evergreen needleleaf PFT, and to see that both papers see excessive discrimination of $^{13}$C. But I think there is an opportunity being missed here, to compare and contrast the results at two similar-but-not-identical sites. This is an opportunity to say something about how evergreen needleleaf forests behave, as informed by dissimilar CLM simulations. The first 2 authors are at the same institution, if not in the same department, and they need to be talking to each other. I think closer coordination/comparison/contrast of the results presented in R16 and here is required. Currently, I find myself more confused than anything. Can we trust CLM at all in evergreen needleleaf forest without extensive tuning from site-specific observations?

3. The authors question the veracity of the observations twice, with regard to H for 1998-2003, and SWC for 1998-2002. I urge extreme caution here, and sincerely hope the authors have corresponded with Dr. Wharton to express their concerns and make these statements with the understanding and approval of the site PI. This is the kind of situation that can foster distrust and animosity between the observational and modeling communities. I most strongly recommend that the authors verify that there may be uncertainty in these observational records, and the listing of Dr. Wharton as a coauthor would legitimize the claims as stated in the MS and confirm that she has participated in discussions of this issue.

4. Isotopes are difficult. They are difficult to explain, and difficult to understand for many (if not most) readers. The provenance of the treatment of carbon isotopes in CLM is poorly summarized in this MS. Oleson et al. (2013) does

not mention them; is Mao (2016) the seminal paper? What about Randerson et al. (2002) or Suits et al. (2005) which investigated isotopes using SiB, or the work of Van der Velde et al. (2013, 2014) which studies isotopes in SiB-CASA? Did isotopes in CLM build from that work, or from an independent source? R16 is cited, but elaboration is warranted, especially with regard to superposition of an annual cycle onto the larger trend in forcing data. The treatment of $\delta^{13}C_{ER}$ in models is extremely difficult to capture. Heterotrophic respiration is comprised of old, intermediate, and young components, and the $\delta^{13}C$ of each is difficult to constrain, as is the fractional contribution of each. The description of $\delta^{13}C_{ER}$ in section 3.3.1 is troubling; if I understand this section correctly, $\delta^{13}C_{ER}$ follows $\delta^{13}C_{GPP}$ in the daytime, then switches to follow $\delta^{13}C_{HR}$ at night. If the $\delta^{13}C$ of the $C_{XS}$ pool has no sensitivity to recent discrimination, then I assume the pool is large enough that the $\delta^{13}C$ of this pool reflects some previous state. Is this true? What is that state? Is it realistic? The MS states that this behavior "aligns with expectations", but is it realistic? Is this behavior observed at sites with more detailed observations? If this behavior is not observed anywhere, how can you trust the model results? Similarly, the authors state that "Autotrophic respiration at Wind River is likely fueled by a mixture of stored and recently-fixed carbon, as indicated by $^{14}C$ measurements...(and) cannot be appropriately modeled by CLM with the current allocation scheme..." so if I understand correctly, we can't trust the $\delta^{13}C_{HR}$ because CLM doesn't consider different contributions from differently-aged dead pools, and we can't trust the $\delta^{13}C_{AR}$ either. Both $\delta^{13}C_{AR}$ and $\delta^{13}C_{HR}$ influence the $\delta^{13}C$ of the canopy air, which will in turn have a strong influence on $\delta^{13}C_{GPP}$! At this point I'm left thinking that we have no confidence in any of the discrimination values of this version of CLM, and any resemblance to observations is a happy accident.

I do not recommend rejection, but this paper requires major revisions to be acceptable for publication. The differences between the findings at Wind River and those of R16 must be reconciled, and the problems with the isotope treatment must be resolved. Furthermore, the characterization of climatological dry summer as 'drought' is unacceptable.

Specific Comments

- The plant wilting factor, $w_i$, can be expressed in a multitude of ways, and could have serious implications for this site. Entekhabi and Eagleson (1989) suggest a linear reduction in $w_i$ from some point s* below field capacity (where $w_i$ = 1) to a value of 0 at wilt point, while Sellers et al (1996) and Colello et al (1998) promote a nonlinear equation for $w_i$ based on field data from FIFE. Baker et al (2008) demonstrate that in tropical forests, a direct linkage of $w_i$ to the vertical profile of root density can be problematic. What

form does $w_i$ take in CLM? Neither R16 nor this manuscript discuss this; was it investigated? This suggests yet another path that can be taken to tune CLM. More equifinality.

- Discussing equation 5, the authors state that since $g_b >> g_s$, $g_b$ can be neglected in the calculation of $c_i/c_a$, and therefore discrimination. In midday this is certainly true, but what about near sunrise/sunset, particularly sunrise? I can imagine that immediately after sunrise, the canopy is cool and the nocturnal inversion has not yet been broken. The canopy, however, is illuminated, and both temperature and humidity are favorable for stomatal conductance. In this situation I might expect that $g_s$ could be larger than $g_b$. This might also have some bearing on the large excursions in $\delta^{13}C^{GPP}$ values seen early and late in the day. Was this investigated?
- Line 31, page 6; should be 'resulting', not 'resulted'
- Why is the observed δ13C of bottom canopy leaves so much lower than elsewhere in the canopy? Does this inform the CLM treatment of isotopes?
- ET will be composed of transpiration, leaf evaporation and ground evaporation components. In a dense canopy like Wind River, I would expect ground evaporation to be low, but excessive leaf evaporation could influence the amount of infiltration and therefore the amount of water available for transpiration later in the season. What is the partition of these components at Wind River? I know it is impossible to quantify these components with a single ET observation, but the model partition may give insight into behavior.
- The authors mention a "spring-to-summer decrease in the contribution of root respiration towards total soil respiration". That makes me think that the observed signal could come about from one of 2 ways. 1) there is a decrease in discrimination as WUE increases, or 2) there is an increase in the HR component towards older material with a lower $\delta^{13}C$. Is this possible?
- When discussing soil moisture, fraction of saturation may be more useful than volumetric water content. The volumetric content at various important points (wilt point, field capacity, saturation) can vary significantly depending on soil character.
- Page 13, lines 14-17: "Observed $\delta^{13}C_{ER}$ was found to have a low negative correlation with observed $G_c$, but not statistically significant. The low correlation was likely a result of $\delta^{13}C_{ER}$ reflecting constraints of prior environmental drivers in comparison with the more rapid response of $G_c$ to more recent environmental drivers." This is not surprising, but doesn't it underscore the importance of getting a handle on all these drivers, in both your observations and model? Aren't you basically saying here that the model results are not to be trusted because the proper mechanistic pathways for the various isotopes are not simulated?

References:

Baker, I.T., L. Prihodko, A.S. Denning, M. Goulden, S. Milller, H. da Rocha, 2008: Seasonal Drought Stress in the Amazon: Reconciling Models and Observations. J.Geophys. Res., 113, G00B01, doi:10.1029/2007JG000644.

Colello, G.D. and Grivet, C., P.J. Sellers, J.A. Berry, 1998: Modeling of Energy, Water and CO2 Flux in a Temperate Grassland Ecosystem with SiB2: May-October 1987. Journal of the Atmospheric Sciences, 55, 1141- 1169, 01 April 1998.

Costa, M. H., M. C. Biajoli, L. Sanches, A. C. M. Malhado, L. R. Hutyra, H. R. da Rocha, R. G. Aguiar, and A. C. de Araújo (2010), Atmospheric versus vegetation controls of Amazonian tropical rain forest evapotranspiration: Are the wet and seasonally dry rain forests any different?, J. Geophys. Res., 115, G04021, doi:10.1029/2009JG001179.

da Rocha, H. R., et al. (2009), Patterns of water and heat flux across a biome gradient from tropical forest to savanna in Brazil, J. Geophys. Res., 114, G00B12, doi:10.1029/2007JG000640.

Entekhabi, D. and Eagleson, P.S., 1998: Land surface hydrology parameterization for atmospheric general circulation models including subgrid scale spatial variability. J. Clim., 2, 816-831.

Randerson, J. T., G. J. Collatz, J. E. Fessenden, A. D. Munoz, C. J. Still, J. A. Berry, I. Y. Fung, N. Suits, and A. S. Denning, A possible global covariance between terrestrial gross primary production and $^{13}$C discrimination: Consequences for the atmospheric $^{13}$C budget and its response to ENSO, Global Biogeochem. Cycles, 16(4), 1136, doi:10.1029/2001GB001845, 2002.

Sellers, P.J., D.A. Randall, G.J. Collatz, J.A. Berry, C.B. Field, D.A. Dazlich, C. Zhang, G.D. Collelo, and L. Bounoua,1996: A Revised Land Surface Parameteriztion (SiB2) for Atmospheric GCMs. Part I: Model Formulation. Journal of Climate, 9(4), 676-705

Suits, N.S., A.S. Denning, J.A. Berry, C.J. Still, J. Kaduk, J.B. Miller, I.T. Baker, 2005: Simulation of carbon isotope discrimination of the terrestrial biosphere. Global Biogeochem. Cy., 19(1), Art No. GB1017, Mar 5 2005.

Van der Velde, I.R., J.B. Miller, K. Schaefer, K.A. Masarie, S. Denning, J,W.C. White, P.P. Tans, M.C. Krol, W. Peters, 2013: Biosphere model simulations of interannual variability in terrestrial 13C/12C exchange. Global Biogeochemical Cycles, 27, 637-649, doi:10.1002/gbc.20048.

Van der Velde, I.R., J.B. Miller, K. Schaefer, G.R. van der Werf, M.C. Krol, W. Peters, 2014: Terrestrial cycling of 13CO2 by photosynthesis, respiration and biomass burning in SiBCASA. Biogeosciences, 11, 6553-6571, doi:10.5194/bg-11-6553-2014.

---

## Author Response (AR1)

**Authors' Response to Associate Editor and Reviewers**

Evaluating the Community Land Model (CLM 4.5) at a Coniferous Forest Site in Northwestern United States Using Flux and Carbon-Isotope Measurements
Henrique F. Duarte, Brett M. Raczka, Daniel M. Ricciuto, John C. Lin, Charles D. Koven, Peter E. Thornton, David R. Bowling, Chun-Ta Lai, Kenneth J. Bible, and James R. Ehleringer

We would like to thank the Associate Editor, Dr. Andreas Ibrom, and the two anonymous Reviewers for their comments, which greatly helped to improve this manuscript.

10 All comments are listed below in italic font and are numbered for easy reference. Our responses are in blue, regular font. The responses to Reviewers 1 and 2 are reproduced from the original document published in the discussion board, with the following modification: at the end of each response, where applicable, we indicate the changes that were implemented and their location (P[page_number]L[line number]) in the revised manuscript (blue, bold font). Cited page and line

15 numbers in the responses correspond to our published discussion paper. The cited literature is listed at the end of this document. Figures R1–3 correspond to the new figures included in this document.

On behalf of all authors,

Henrique F. Duarte
20 University of Utah
h.duarte@utah.edu

**Comments from Reviewer #1**

*This paper describes an application of the CLM model to the Wind River flux tower site.*

25 [RC1.1] *The first point to note is that model parameterization is done in the good old-fashioned way. When applied out of the box with standard PFT parameters, the model does not fit very well. Hence, the model needs to be calibrated. This calibration is done by adjusting parameter values manually, based on the literature and some trial-and-error, until the fit to the data is not too bad. Most groups are moving away from this approach to parameterization, to a more rigorous*
30 *statistical framework such as Bayesian calibration, which yields more defensible parameter values. I don't think it's essential that the authors do this, but it would be good if they could give at least some justification for sticking with the traditional method of parameterization.*

**A:** The adjusted parameters were primarily based on biological measurements at Wind River or at similar stands in the Pacific Northwest (Sect. 2.6, P7L5–16). Soil hydraulic parameter values were
35 switched back to the values in CLM 4.0 (via a configuration option in the model), leading to a more

accurate simulation of soil moisture at the site. Root distribution parameters were also changed in order to reduce the excessive late-summer soil moisture stress and gross primary production (GPP) down-regulation in the model. The default values for the needleleaf evergreen temperate tree PFT were replaced by the default values for broadleaf evergreen temperate tree PFT, shifting roots to deeper soil layers (this change was justified based on our physical understanding of the site – see further discussion in our response to RC1.13 and RC2.3). The $SLA_0$ parameter (specific leaf area at canopy top) was adjusted by manual trial and error (decreasing values were attempted), with $m$ (linear coefficient in Eq. A1) values constrained by Eq. (A2), the $SLA_0$ value, and the site observations of leaf area index (LAI) and leaf carbon, aiming to minimize model errors in the simulation of GPP and LAI (Sect. A4).

A Bayesian calibration approach would be complicated by the current lack of prior parameter distributions within CLM in order to create a model ensemble and the computational expense of running a calibration. Commonly used techniques such as Markov Chain Monte Carlo (MCMC) are prohibitively expensive with long CLM simulations, and more advanced techniques for calibration (e.g. using surrogate modeling approaches) are still under development. The simpler approach we used proved to be an effective method to improve model performance at the Wind River AmeriFlux site. Given our observation-based parameter adjustments, combined with careful trial and error adjustments to match the observed surface fluxes, it is unlikely that a Bayesian approach could provide a meaningfully better fit.

**Manuscript revision**

**Added in P10L1–7: "Bayesian parameter calibration is a common approach used in modelling studies to account for both the prior parameter distributions and more recent observations. In this case, a Bayesian calibration approach would be complicated by the current lack of prior parameter distributions within CLM in order to create a model ensemble and the computational expense of running a calibration. Commonly used techniques such as Markov Chain Monte Carlo (MCMC) are prohibitively expensive with long CLM simulations, and more advanced techniques for calibration (e.g. using surrogate modeling approaches) are still under development. The simpler approach used here proved to be an effective method to improve model performance at the Wind River AmeriFlux site."**

[RC1.2] *The second striking thing about this manuscript was the almost complete lack of reference to the literature in the Discussion. The Results and Discussion are combined into the one section – never a good idea in my view. Here, there is almost no discussion of the results, and no attempt to place the results in the context of the literature. Overall, I came away with a strong "so what" feeling: the authors do not do a good job of articulating why they want to calibrate CLM for this site, nor what we get out of it. There is little in the Introduction to motivate the study, and nothing in the conclusions about how this work advances the field in general. I very strongly suggest that the authors - Better motivate the study in the Introduction, with an expectation of the kinds of questions that this work can address - Separate the Results from the Discussion -Focus the Discussion on what we learn from this study, and ensure that it is placed in the broader context of the literature with appropriate citations.*

**A:** The key findings in our paper, summarized below, represent important contributions to the modeling community and are stated in the Conclusions. We intend to include suggested citations from Reviewer 2 to bolster the Introduction and more clearly lay out motivation to why work like this is important to the modeling community. We will also improve the discussion of our results in the context of the literature, especially Raczka et al. (2016) (nitrogen limitation in CLM; see response to RC2.1.2), Wharton et al. (2009) (observed physiological response to water stress at Wind River; see response to RC2.1.1), and previous studies involving the simulation of carbon isotopes with different land models, such as SiB and SiB-CASA (e.g., Randerson et al. 2002; Suits et al. 2005; Van der Velde et al. 2013,2014). We will also separate Results from Discussion, as suggested by the Reviewer.

Key points:

Here we assess the current version of the Community Land Model (CLM 4.5, Oleson et al. 2013), focusing primarily on the model skill in the simulation of stomatal conductance and its response to water stress (P3L5–8). The Wind River site was chosen for this study because of its climatology (dry summers) and long record of meteorological, biological, surface flux (energy and carbon), and carbon isotope measurements for model assessment (P4L20–22). We leveraged the inclusion of photosynthetic [13]C discrimination dynamics within CLM 4.5 to better diagnose the simulation of stomatal conductance at the site.

As discussed in P6L29–P7L4, our initial CLM simulations at Wind River using out-of-the-box parameters performed poorly, which is not quite a surprise given the fact that those parameters are based on global optimizations. It is also important to mention that Wind River is characterized by a unique old-growth forest, susceptible to stand age effects (Wharton et al. 2009). The default model parameters, likely based upon secondary-growth forests, may therefore not be appropriate. In this study the presentation of the results and discussion on model performance focus on the simulations using calibrated parameters. Note that, without calibration, we would not be able to properly test the new model structure, especially the photosynthetic [13]C discrimination scheme.

Overall, the calibrated CLM was able to simulate the observed response of canopy conductance to atmospheric VPD and soil water content, reasonably capturing the impact of water stress on ecosystem functioning (P12L29; P13L8; P13L25). The calibrated parameters we present in the paper may be of use for future modelling studies involving stands of similar age and composition under a similar climate regime (P13L30).

A critical adjustment was a significant reduction of the slope in the Ball-Berry stomatal conductance equation ($m_{bb}$) from 9 to 6, in alignment with observations reported in the literature for conifer trees and with the CLM results by Mao et al. (2016) for a loblolly pine stand in southeastern USA (Sect. A9). It also corroborates the recommendation of a lower $m_{bb}$ value by Raczka et al. (2016) based on CLM runs at a subalpine conifer forest site in Colorado, USA. Currently, CLM uses the same $m_{bb}$ value (9) for all C3 plants. Based on the results, we originally concluded that a future release of CLM would benefit from using a distinct $m_{bb}$ value (6) for conifers (P14L1). However, additional analysis indicates that the significant reduction in $m_{bb}$

mostly accounted for the partial coupling between net leaf photosynthesis ($A_n$) and stomatal conductance ($g_s$) resulting from the nitrogen limitation scheme in the model. This partial coupling between $A_n$ and $g_s$ is a shortcoming of CLM (Raczka et al. 2016; Metcalfe et al. 2016) which we will more formally address in the revised paper (see further discussion in our response to RC2.1.2).

5    We speculate that a lower $m_{bb}$ value for C3 plants other than conifers would also be necessary given the current nitrogen limitation scheme in CLM. Even so, we believe that a distinct parameterization for conifers, as done in other models such as SiB, would still benefit CLM.

As discussed in P11L11–23 and P14L12–17, our results show that carbon isotope measurements can be used to constrain stomatal conductance and intrinsic water-use efficiency in CLM, as an

10    alternative to eddy-covariance flux measurements. Our results also show that carbon isotopes expose a conceptual weakness in CLM's carbon allocation scheme (lack of an explicit representation of carbohydrate storage pools) and may guide future improvements in the model. The need for a better representation of carbohydrate storage pools within CLM is also highlighted by the $^{13}CO_2$-labeling study conducted by Mao et al. (2016) (P10L21).

15    The key findings discussed above represent important contributions to the modeling community and are stated in the Conclusions (the recent findings related to the nitrogen limitation scheme will be included in the revised paper).

**Manuscript revision**

**We carefully followed the recommendation from Reviewer 1:**

20    **1. Introduction was significantly revised. Additional citations were included, helping to more clearly lay out the motivation of our study.**

**2. Discussion of results was expanded as explained above. A separate "Discussion" section was created (new Sect. 4), where we focus on the key findings of our study.**

**Abstract and Conclusion were revised accordingly.**

25    *Some comments on the methods:*

*[RC1.3] The drought stress factor should be more clearly defined: I'd like to see the equation for the plant wilting factor, which apparently depends on both soil water potential (state variable) and the plant dependent response to water stress.*

**A:** In CLM Version 4.5, the plant wilting factor for soil layer $i$ is defined as:

$$w_i = \begin{cases} \dfrac{\Psi_c - \Psi_i}{\Psi_c - \Psi_o}\left[\dfrac{\theta_{sat,i} - \theta_{ice,i}}{\theta_{sat,i}}\right] \leq 1 & \text{for } T_i > T_f - 2 \text{ and } \theta_{liq,i} > 0 \\ 0 & \text{for } T_i \leq T_f - 2 \text{ or } \theta_{liq,i} \leq 0 \end{cases}$$

30    where $\Psi_i$ is the soil water matric potential, $\Psi_c$ and $\Psi_o$ are the soil water potential when stomata are fully closed or fully open, respectively ($\Psi_c = -255000$ mm and $\Psi_o = -66000$ mm for NETT PFT), $\theta_{sat,i}$ is the saturated volumetric water content, $\theta_{ice,i}$ is the volumetric ice content, $\theta_{liq,i}$ is

the volumetric liquid water content, $T_i$ is the soil layer temperature, and $T_f = 273.15$ K is the freezing temperature of water. $w_i$ and root fraction ($r_i$, Eq. A3) are used to calculate the soil moisture stress factor $\beta_t$ (Eq. 3). We will include the $w_i$ equation in the revised paper.

**Manuscript revision**

5     **We added the $w_i$ equation (Eq. 5)**

[RC1.4] *I don't understand the use of the factor 'd' in equations 5 & 6. As I understand it, the relationship A = gs/1.6 (Cs – Ci) is a physical description of the diffusion process through the stomata. How can this be modified by nitrogen limitation? Or is this something that affects the "isotopic" Ci/Ca only?*

10     **A:** The method in which CLM 4.5 incorporates nitrogen down-regulation of photosynthesis complicates the implementation of the above equation. Although initially the net leaf photosynthesis ($A_n$) and leaf stomatal conductance ($g_s$) are "fully coupled", meaning $A_n$ and $g_s$ are solved simultaneously, this is done initially without including the effects of nitrogen limitation upon assimilation rate. Therefore, the initial fully-coupled solution between $A_n$ and $g_s$ is the

15     "potential" assimilation rate that leads to a potential gross primary production (GPP$_{pot}$). CLM 4.5 then down-regulates GPP$_{pot}$ (and potential $A_n$) according to what nitrogen is available, and what nitrogen is required to allocate new carbon tissue based upon C:N ratio. This down-regulation is used to calculate actual GPP (and actual $A_n$). This is a weakness in CLM, because this downscaling de-couples $A_n$ from $g_s$, and is why CLM 4.5 is a "partially-coupled" model.

20     The nitrogen photosynthetic downregulation factor in Eqs. (5) and (6) is defined as

$$d = \frac{\text{CF}_{\text{avail\_alloc}} - \text{CF}_{\text{alloc}}}{\text{GPP}_{\text{pot}}}$$

where $\text{CF}_{\text{avail\_alloc}}$ is the carbon flux from photosynthesis which is available to new growth allocation and $\text{CF}_{\text{alloc}}$ is the actual carbon allocation to new growth (limited by nitrogen availability). Actual GPP is calculated as

$$\text{GPP} = \text{GPP}_{\text{pot}}(1 - d).$$

25     The net leaf photosynthesis term $A_n$ in Eqs. (5) and (6) corresponds to potential $A_n$. In order to make $c_i$ consistent with the actual, nitrogen limited GPP, $A_n$ is multiplied by $(1 - d)$ in Eqs. (5) and (6). Note, however, that $g_s$ in those equations is consistent with $A_n$, not $A_n(1 - d)$. We discuss the implications of this mismatch in our response to RC2.1.2. A more detailed description of CLM 4.5's nitrogen limitation scheme and its shortcomings is presented by Raczka et al. (2016).

30     We will include this information in the revised paper.

**Manuscript revision**

**The information discussed above was included in Sect. 2.1.**

[RC1.5] *I note that mesophyll conductance also affects the isotopic ratio – is this accounted for in this model?*

**A:** CLM 4.5 does not account for mesophyll conductance (intracellular $CO_2$ is assumed to be the same as intercellular $CO_2$). Raczka et al. (2016) hypothesizes that inclusion of mesophyll conductance could improve the magnitude of $^{13}C$ discrimination in the model. We will discuss this in the revised paper. Mesophyll conductance was recently incorporated and tested in CLM 4.5 (Sun et al., 2014), and in the future we could explore this by working to link mesophyll conductance to the carbon isotope submodel.

**Manuscript revision**

**This is addressed in P6L13 and P21L12.**

[RC1.6] *Please add a description of how the model scales from leaf to canopy. As all of the comparisons are with canopy-scale GPP, LE and Gs, it is important for the reader to know the principal assumptions underlying this scaling. How is leaf isotopic composition modeled for the whole canopy? How is leaf conductance scaled to the canopy?*

**A:** Leaf stomatal resistance and photosynthesis are solved separately for sunlit and shaded leaves. Canopy conductance is given by

$$G_s = \frac{1}{r_b + r_s^{sun}} \text{LAI}^{sun} + \frac{1}{r_b + r_s^{sha}} \text{LAI}^{sha}$$

and potential GPP by

$$\text{GPP}_{pot} = (A_n^{sun} + R_d^{sun})\text{LAI}^{sun} + \left(A_n^{sha} + R_d^{sha}\right)\text{LAI}^{sha}$$

where $r_b$ is the leaf boundary layer resistance, $r_s$ is the leaf stomatal resistance, LAI is the leaf area index, $A_n$ is the net leaf photosynthesis, and $R_d$ is leaf-level respiration ($sun$ and $sha$ superscripts denote sunlit and shaded leaves, respectively). Photosynthetic parameters such as $V_{cmax25}$ are solved separately for sunlit and shaded leaves and their canopy scaling scheme is detailed in Oleson et al. (2013, Sect. 8.3). Actual GPP is calculated from $\text{GPP}_{pot}$ and the nitrogen photosynthetic downregulation factor, $d$ (see our response to RC1.4). Note that, as discussed in Sect. 2.7, we opted to calculate a modeled canopy conductance using the Penman-Monteith method (Eq. 7) and modeled surface fluxes in order to allow a more direct comparison against observations (Penman-Monteith method using observed surface fluxes).

$\delta^{13}C_{GPP}$ is determined based on the carbon assimilation and photosynthetic $^{13}C$ discrimination by sunlit and shaded leaves and their respective leaf area indices. The carbon isotope ratio of newly allocated carbon is the same as $\delta^{13}C_{GPP}$. The $\delta^{13}C$ of the leaf carbon pool depends on the inward allocation flux and its $\delta^{13}C$ ($\delta^{13}C_{GPP}$), in addition to the turnover time of the pool.

We will incorporate these descriptions in the revised paper.

**The above descriptions were added in Sect. 2.1.**

[RC1.7] *The model is evaluated against gap-filled flux data. In my view that's not acceptable: evaluating a model against gap-filled data is comparing one model against another. The model*
5     *should only be evaluated against non-gap-filled data.*

**A:** Gap-filled data (AmeriFlux L4 data) were used for a general assessment of CLM in respect to the simulation of energy and carbon fluxes (Fig. 3). For the entire period of study (1998–2006), a reasonably small fraction of the half-hourly energy flux data actually corresponds to gap-filled data (22% for the sensible heat flux data and 24% for the latent heat flux data). In our view the L4 data
10    are sufficient for a general evaluation of seasonal patterns of CLM. Note that we did *not* use gap-filled flux data for calculations (e.g. canopy conductance) in our study, nor did we use them to calculate annual integrals of net ecosystem exchange (NEE) or gross primary production (GPP), that may be influenced by the gap-filling method.

The GPP and ecosystem respiration (ER) data, strictly speaking, do not correspond to observations
15    but to model products based on NEE measurements at the site, as pointed out in the text (P9L5). We opted to use these products as a reference in the evaluation of CLM. Note that comparing modeled output against partitioned GPP and ER flux tower data is common in the literature.

[RC1.8] *Please describe more clearly the process used for calibration. For example, p16 says that SLA0 is optimized by aiming to minimize model errors in site observations of LAI and Cl – was this*
20    *done using a solver function, or simply by manual trial and error?*

**A:** The adjusted parameters were primarily based on biological measurements at Wind River or at similar stands in the Pacific Northwest (Sect. 2.6, P7L5–16). Soil hydraulic parameter values were switched back to the values in CLM 4.0 (via a configuration option in the model), leading to a more accurate simulation of soil moisture at the site. Root distribution parameters were also changed in
25    order to reduce the excessive late-summer soil moisture stress and gross primary production (GPP) down-regulation in the model. The default values for the needleleaf evergreen temperate tree PFT were replaced by the default values for broadleaf evergreen temperate tree PFT, shifting roots to deeper soil layers (this change was justified based on our physical understanding of the site – see further discussion in our response to RC1.13 and RC2.3). The $SLA_0$ parameter (specific
30    leaf area at canopy top) was adjusted by manual trial and error (decreasing values were attempted), with $m$ (linear coefficient in Eq. A1) values constrained by Eq. (A2), the $SLA_0$ value, and the site observations of leaf area index (LAI) and leaf carbon, aiming to minimize model errors in the simulation of GPP and LAI (Sect. A4). We will clarify in the revised text that $SLA_0$ was adjusted by manual trial and error.

35    **Manuscript revision**

**Clarification that SLA$_0$ was adjusted by manual trial and error was added in P24L4.**

*On the results:*

[RC1.9] *Figure 2 could show observations as well as model output, making it easier to visualize the model-data correspondence. Please indicate in Table 2 what the errors refer to (+/-SE? 95% CI? Range?)*

**A:** We will add the observations from Table 2 in Fig. 2 as suggested. In Table 2, the observations from the AmeriFlux database are given as *mean ± standard deviation*. The values from Fessenden and Ehleringer (2003) correspond to the range of observed values in their Figs. 2b ($\delta^{13}$C leaf) and 3 ($\delta^{13}$C SOM). We will clarify this in Table 2.

**Manuscript revision**

**We added the observations from the original Table 2 in Fig. 2 and removed Table 2 from the revised manuscript. Observed $\delta^{13}C_{leaf}$ and $\delta^{13}C_{SOM}$ are now expressed as avg ± std. dev. instead of range. Information about the error bars is included in the figure caption.**

[RC1.10] *I was unsure how to evaluate the leaf isotopic data. Are the modeled values to be compared with the top, bottom, or average of the canopy? See note above about how isotope discrimination is scaled to the whole canopy.*

**A:** CLM is a two-big-leaf (sunlit and shaded leaves), single-canopy-layer model (a multi-layer option is available, but not supported). The modeled leaf $\delta^{13}$C output corresponds to the isotopic signature of the entire leaf carbon pool, which is calculated from both sunlit and shaded portions of the leaf canopy. Further details are provided in our response to RC1.6.

The observed leaf $\delta^{13}$C values in Table 2 correspond to measurements at canopy top (55 m), middle (25 m), and bottom (2 m). As pointed out by Fessenden and Ehleringer (2003), the decrease in the leaf $\delta^{13}$C values (i.e., increase in photosynthetic $^{13}$C discrimination) with canopy depth can be explained by light reduction within the canopy. The observed mid-canopy values are expected to better represent the isotopic composition of leaves for the whole canopy, in comparison with the observed values at the two canopy extremes, especially given the larger amount of leaf biomass in mid canopy. Therefore, the mid-canopy leaf $\delta^{13}$C values in Table 2 should provide a better reference for evaluating CLM, which does not explicitly resolve the leaf $\delta^{13}$C by canopy height. We will discuss this in the revised paper.

**Manuscript revision**

**We added the above discussion in the revised paper (P11L22–P12L2), with a few modifications:**

**"It is important to clarify that CLM has leaf properties that vary continuously with canopy depth, and that two leaf categories (sunlit and shaded leaves) are estimated dynamically on every time step, as a function of canopy structure and solar elevation angle (Thornton and Zimmerman, 2007). The modeled leaf $\delta^{13}$C output corresponds to the isotopic signature of the entire leaf carbon pool, which is calculated from both sunlit and shaded portions of the leaf canopy (see Sect. 2.1). The observed leaf $\delta^{13}$C values in Fig. 2g correspond to measurements at canopy top**

(55 m), middle (25 m), and bottom (2 m). As pointed out by Fessenden and Ehleringer (2003), the decrease in the observed leaf $\delta^{13}$C values (i.e., increase in photosynthetic $^{13}$C discrimination) with canopy depth can be explained by light reduction within the canopy. In principle, the observed mid-canopy values are expected to better represent the isotopic composition of leaves for the whole canopy, in comparison with the observed values at the two canopy extremes, especially given the larger amount of leaf biomass in mid canopy. However, considering how light is reduced within the canopy, the top-canopy $\delta^{13}$C value should still be representative of a significant fraction of the canopy as well, so the whole canopy $\delta^{13}$C is expected to lay somewhere in between the top- and mid-canopy values. As shown in Fig. 2g, the modeled $\delta^{13}$C of the leaf carbon pool was the average between the observed values at canopy top and middle."

[RC1.11] *It would have been good to see the model performance with the parameters out of the box, as well as model performance with calibrated parameters, in order to visualize the effect of altering model parameters.*

**A:** Figures R1 and R2 below compare the performance of CLM for key model outputs when using "out-of-the-box" parameters and calibrated parameters. We intend to add these figures in the Appendix of the revised paper. Note that the model performance improved substantially after calibration and allowed for a fair test of the photosynthetic $^{13}$C discrimination scheme in CLM.

**Manuscript revision**

**Figures R1 and R2 were added in the revised manuscript (Figs. A4 and A5, respectively).**

[Figure]

**Figure R1**.  Comparison of CLM performance at Wind River when using default, "out-of-the-box" parameters (black lines) and calibrated parameters (red lines). Observations (*mean ± std. dev.*, blue points and error bars) are from the AmeriFlux database.

[Figure]

**Figure R2**. Comparison of CLM performance at Wind River when using default, "out-of-the-box" parameters (black lines) and calibrated parameters (red lines) against site observations (blue lines). For clearer visualization, the data presented correspond to Bézier-smoothed daily averages as in Fig. 3.

[RC1.12] *Please discuss the lack of energy balance closure at this site. The model assumes the energy balance is closed; if the data show a lack of closure the model must show a bias in its predictions of either LE or H. How large is the lack of closure at this site, and how does it affect the model comparison to data?*

**A:** We calculated the energy balance ratio, $\text{EBR} = (H + LE)/(R_n - G)$, using 30-min, L2 (V007) data from the AmeriFlux repository ($H$ is sensible heat flux, $LE$ is latent heat flux, $R_n$ is net radiation, and $G$ is soil heat flux). We used data from June–September, 10:00–14:00, and rejected

periods with rain or unfavorable wind direction. With the available data, we were able to calculate EBR for the years of 1998–2001, 2004, and 2006.

The overall mean EBR is 0.88. The energy balance closure for years 2004 and 2006 is high (mean EBR = 1.01 and 1.09, respectively). Note in Fig. 3a and b that the model bias in the estimation of $H$ and $LE$ is reduced in those years. In years 1998, 2000 and 2001, mean EBR is significantly lower (0.63, 0.69 and 0.76, respectively). Note in Fig. 3a the positive bias in modeled $H$ for those years. As discussed in the text (P9L13), the observed $H$ values in 1998–2003 are significantly smaller than in 2004–2006, while the $LE$ observations show approximately the same pattern over the years. The change in the pattern of observed $H$ was reported as a potential data issue in the Wind River site documentation (AmeriFlux repository[#]). The low EBR for years 1998, 2000 and 2001 supports this notion, and suggests that the observed $H$ values were too low in 1998–2003. Mean EBR in 1999 is relatively high (0.92), where the reduced $H$ values (Fig. 3a) are compensated by larger $LE$ values (Fig. 3b). In that year, modeled $H$ ($LE$) has a positive (negative) bias in respect to the observations.

[#]ftp://cdiac.ornl.gov/pub/ameriflux/data/Level4/Sites_ByName/Wind_River_Field_Station/history_changes.txt

**Manuscript revision**

**We included the above results in the revised manuscript (P12L22–31). The methodology used in the calculation of EBR was included in P8L3–7.**

[RC1.13] *What is the average rooting depth? The SWC data shown are only to 30 cm – how much deeper than this do the roots penetrate? Is the lack of response to low SWC a function of only considering the very top soil?*

**A:** Shaw et al. (2004) provides a good description of rooting depth at Wind River: "Plant roots are concentrated above 50 cm in soil profiles; however, roots as deep as 2.05 m have been observed in younger forests growing on nearly identical soils (T. Hinckley personal communication). Many coarse roots of Douglas-fir extend to depths greater than 1.0 m. Tip-up mounds of windthrown western hemlock trees typically have a classic flat root plate indicative of shallow rooting" (Douglas-fir and western hemlock are the dominant species at the site).

In CLM, root fraction as a function of soil depth is calculated using Eq. (A3). With the default root distribution parameters, the total root fraction in the top 46 and 130 cm of soil is 78% and 96%, respectively (note the small fraction of roots at depths below 1.3 m (4%)). The above site description (Shaw et al. 2004) suggests that the default parameters are inadequate at Wind River, resulting in a "too-shallow" rooting profile. As discussed in Sect. A8, we adjusted the $r_b$ root distribution parameter from 2 m$^{-1}$ to 1 m$^{-1}$ (based on the default CLM value for broadleaf evergreen temperate tree PFT), shifting roots to deeper soil layers, aiming to reduce the excessive late-summer soil moisture stress and downregulation of gross primary production (GPP) in the model. With the adjusted $r_b$ parameter, the total root fraction in the top 46 and 130 cm of soil is 67% and 86%, respectively (14% below 1.3 m), which seems more reasonable based on Shaw et al.

(2004) and the fact that Douglas-fir trees at the site are about 500 years old and 40–65-m tall. Our adjustment of soil moisture stress in CLM via root distribution is therefore physically justified. This adjustment is further discussed in our response to RC1.14.

With regards to the third question, we believe the reviewer is referring to the apparent lack of
5   response of the observed canopy conductance ($G_c$) to observed soil water content (SWC) (Fig. 7a). As we point out in the text (see Sect. 3.4, second and fourth paragraphs), this is likely associated with a negative bias in the observed SWC data in 1999–2002 due to a different instrumentation setup. Excluding this period in which the SWC data may be biased, CLM was able to match the observed SWC values reasonably well, especially during the summer months (note that observed
10   precipitation was used to drive CLM). In Fig. 7b, note that the data points ($G_c$ vs. VPD) correspond to *observations*, but those are binned according to *modeled* SWC. The impact of soil moisture on the observed $G_c$ then becomes evident. The result supports the suspicion of a negative bias in the observed SWC data in 1999–2002.

**Manuscript revision**

15   **The above discussion on root distribution at Wind River was added in the revised manuscript (Appendix A8).**

[RC1.14] *The demonstration that the model over-estimates the effect of low SWC in the topsoil is interesting, but difficult to interpret without the rooting depth and the formulation for soil moisture stress being given. Nothing is said about how the model might be improved based on this*
20   *observation – it would be good if the authors could identify the root cause for this mismatch and suggest how it could be addressed.*

**A:** See our response to RC1.13. Note that the result in Fig. 7a is likely impacted by a negative bias in the observed soil water content (SWC) data from 1999 to 2002. Please compare Fig. 7b with Fig. 8b, the former showing observed canopy conductance ($G_c$) on the $y$ axis and the latter showing
25   modeled $G_c$. Both figures show observed vapor pressure deficit (VPD) on the $x$ axis and use modeled SWC (0–27cm) values to segregate the data points into different SWC regimes. The linear regression between log $G_c$ and VPD considering all data points (regardless of SWC value) are virtually identical in both figures. When considering only data points within the lowest SWC bin, the linear regression is similar in both figures, with CLM apparently presenting a small
30   *underestimation* of the effect of low SWC on $G_c$ (i.e., small overestimation of $G_c$). It is important to point out, however, that modeled SWC was used to segregate the observations in Fig. 7b, due to the suspicion of a negative bias in part of the SWC observations, as discussed above. Overall, the results indicate that the calibrated CLM was able to reasonably simulate the observed response of canopy conductance to VPD and SWC.

35   Note that CLM's soil moisture stress factor, $\beta_t$, is defined in Eq. 3. It is a function of root fraction, $r_i$, and a plant wilting factor, $w_i$. The former is defined in Eq. (A3), and the latter has been included in this document (see response to RC1.3). Without any adjustments in the model, soil moisture stress was excessive, resulting in an unrealistic down-regulation of gross primary production (GPP)

during late summer. The excessive soil moisture stress was especially due to an inaccurate simulation of soil moisture at the site (strong dry bias). In addition to the adjustment of soil hydraulic parameters (resolving the soil moisture issue), we also adjusted $r_i$, shifting roots to deeper soil layers (a physically justified approach – see further discussion in our response to

5    RC1.13). We did not adjust $w_i$, but this could be a supplementary approach to better simulate $\beta_t$. This is discussed in our response to RC2.3.

[RC1.15] *Finally, a very important point, something that needs to be said: why on earth is CLM still using the Ball-Berry stomatal model? The Ball-Berry model is physiologically incorrect (see Aphalo & Jarvis 1991, and pretty much every stomatal physiology paper since). It is consistently*

10    *outperformed by models based on VPD. It will incorrectly predict stomatal behaviour in future climates, when VPD is predicted to change but RH stay the same (see Sato et al. 2015 JGR). It is quite odd to read here the justification that "such improvement is expected to be small". I think it is well past time that CLM moved on from the Ball-Berry model.*

**A:** The CLM development team is in a better position to justify the use of the Ball-Berry stomatal

15    model, but it is important to highlight here that the Ball-Berry model is also used in many other major land surface models (LSMs) (Sato et al. 2015; see their Appendix S3). The Ball-Berry model (based on RH) is simpler than the Leuning model (based on VPD), involving a smaller number of parameters. Note that major LSMs are designed to work in a variety of situations where simplicity of formulation can actually be a strength. The LSM intercomparison in the NACP site-level interim

20    synthesis study by Schaefer et al. (2012), for instance, does not support the suggestion that the Ball-Berry approach is clearly inferior. The Reviewer offers Sato et al. (2015) as an example of why the Leuning model would be preferred, however, this is given in context with future climates in which the trend of RH deviates from VPD. Note that this is not the case of our study (we are hindcasting from prescribed meteorology).

25    In P12L25–29, we wrote: "The correlation between modeled $G_c$ and RH was found to be slightly higher than between modeled $G_c$ and VPD, while observed $G_c$ correlated slightly better with VPD than RH (results not shown). The results indicate that a direct dependence on leaf VPD in CLM's stomatal conductance model, rather than leaf RH, would lead to a more accurate representation of stomatal functioning at Wind River, but overall, such improvement is expected to be small. The

30    general dependence of modeled canopy conductance on VPD was very similar in comparison with observations, as indicated by the linear regression curve between $\log G_c$ and VPD in Fig. 8 using all data points ($\log G_c = -0.59\text{VPD} + 6.04$; compare with $\log G_c = -0.59\text{VPD} + 6.06$ in Fig. 7b)."

Note that our conclusions regarding the potential improvement in the simulations by having a direct dependence on leaf VPD, rather than leaf RH, in CLM's stomatal conductance model refer to

35    the results presented in our study only.

As the Reviewer points out, in the case of model predictions under different climate scenarios, in which atmospheric VPD is predicted to change while RH stays the same, a direct dependence on leaf VPD in the stomatal conductance model becomes critical. This is a point that hopefully will be addressed in a future release of CLM, and would be more relevant for climate change simulation

(our analysis uses a stable climate and is a hindcast). We intend to add this remark in the revised paper.

**Manuscript revision**

**We added this discussion in the revised manuscript (P18L21–29): "As pointed out in Sect. 3.4, the results of the present study indicate that a direct dependence on leaf VPD in CLM's stomatal conductance model, rather than leaf RH, would lead to a more accurate representation of stomatal functioning at Wind River, but overall, for the period analyzed in the present study, such improvement is expected to be small. It is important to emphasize that this expectation refers to the results presented here only. In case of model predictions under future climate scenarios, in which atmospheric VPD is predicted to change while RH stays the same (as discussed in Sato et al., 2015), a direct dependence on leaf VPD in the stomatal conductance model becomes critical. The next CLM release (Version 5) is expected to replace the Ball-Berry model with the Medlyn model (Medlyn et al., 2011), which directly depends on leaf VPD. This modification is expected to be more relevant for climate change simulations. Note that the present analysis is based on a hindcast simulation using a stable climate."**

[RC1.16] *Very finally, debating whether to mention this or not, but: I was also struck by the extreme gender imbalance of the authorship list. Ten male authors! I hope and trust that the PIs here are actively taking steps to address this imbalance in their group of collaborators.*

**A:** We share the Reviewer's concerns about diversity in science. We are indeed actively engaged in recruiting, mentoring, and collaborating with female scientists at all career stages.

**Comments from Reviewer #2**

[RC2.1] *This manuscript (MS) presented a difficult review. After the first couple of readings, I found the paper easy to read, with no obvious shortcomings. This should have prompted a favorable review, right? However, the MS does read somewhat like 'we ran a model, here is what happened' which is bothersome to me. The use of isotopes in a model is no longer novel; you need to go beyond just saying you put isotopes in a model and ran it, and say something about how this capability and our inclusion of it in models informs our understanding of natural systems. With more careful reading, more issues emerged for me.*

**A:** We agree that the use of isotopes in a land model is not novel, but we would like to clarify that the simulation of $^{13}C$ isotopes is a new feature in *CLM*. This implementation was made available in the latest release of the model (CLM 4.5) and is evaluated in our paper. Our results show that, thanks to this implementation, carbon isotope measurements can be used to constrain stomatal conductance and intrinsic water-use efficiency in CLM, as an alternative to eddy-covariance flux measurements. Our results also show that carbon isotopes expose a conceptual weakness in CLM's carbon allocation scheme (lack of an explicit representation of carbohydrate storage pools)

and may guide future improvements in the model. In the revised paper, we plan to further discuss these results in the context of other land models.

**Manuscript revision**

**The Introduction Section was substantially revised and now includes a better overview of**
5 **previous modelling studies using carbon isotopes (particular attention was given to studies using $^{13}$C observations as a constraint to modeled stomatal conductance). The discussion of results was expanded and better placed in the context of the literature. Following the recommendation of Reviewer 1 ([RC1.2]), we created a separate Discussion Section (Sect. 4) where we focus on the key findings of our study.**

10 *1.  [RC2.1.1] The first paragraph talks about drought, and drought stress is mentioned throughout the MS. However, drought is an anomalous reduction in precipitation from climatological values. If, climatologically, 5% of annual precipitation occurs in JJA, then a dry summer is just…normal. Drought in this part of the country will likely manifest under climate change as a longer growing season and higher summertime temperatures and VPD when compared to the*
15 *present---day. If meteorological forcing from the period 1998--- 2006 is cycled multiple times to simulate the period 1850---present, you're not going to capture any slow secular changes. Really, what's being done here is an evaluation of CLM's ability to capture a mean seasonal cycle that includes a dry summer. This site has a 50+ meter tall canopy, and fluxes (LE/H) are in phase with Boreal summer. LE fluxes appear to peak in June or July, and H peaks one month or*
20 *so later, from a quick inspection of Figure 3. That suggests an ecosystem that may experience some stress in late summer, but is not water---limited on the whole. Costa et al. (2010) and da Rocha et al. (2009) discuss the notion of environmental and physiological stress, albeit in Amazonian forests. At Wind River, I suspect an ecosystem that is environmentally---limited for large parts of the year (winter, obviously) and may experience some physiological stress in late*
25 *summer. There may be some value in studying interannual variability (IAV), as observed GPP in 2002 (a known western drought that year) and 2006 both taper off rather quickly following an early peak. CLM does not capture this behavior in 2002, but does in 2006. But the authors do not discuss IAV, but rather concentrate on the idea of annual drought stress. I'm not buying it; you're not going to have a 50---meter canopy in a region that is water stressed. Furthermore,*
30 βt *drops below 1 in less than half of the years, and then only to a minimum of 0.6. That's not a whole lot of stress. I strenuously object to the use of the word drought for a dry summer that occurs every year.*

**A:** We thank the Reviewer in alerting us to the potential confusion between "drought stress", "meteorological drought", and "water stress" our original text may have caused. In the paper,
35 we evaluate the capability of CLM to model the ecosystem response to seasonal water stress associated with changes in vapor pressure deficit (VPD) and soil water content (SWC). The old-growth forest at Wind River is subject to water stress each summer, which is accentuated during periods of meteorological drought (precipitation below climatological normal).

The expressions "drought stress" and "water stress" are often used interchangeably in the literature. For example, the expression "seasonal drought stress" appears often in plant physiology studies carried out at sites with Mediterranean climate (also found at Wind River) (e.g., Mooney 1987; Kurpius et al. 2003; Niinemets and Keenan 2014). At Wind River, Wharton et al. (2009) investigated the ecophysiological responses of forest stands of contrasting age to "seasonal drought". In order to avoid confusion with meteorological drought (precipitation below climatological normal), we will use the expression "water stress" in the revised paper.

The major reason for the summer water stress at Wind River is the elevated VPD. Canopy conductance values at the site strongly decrease at moderate VPD levels, regardless of soil moisture conditions. This is shown in our paper and is more extensively discussed in Wharton et al. (2009). As pointed out in their study, "Even under moderate VPD levels, foliage at the tops of tall evergreen conifer trees often reach near critical values for cavitation due to a long path distance between the water table and the hydraulic capacity of the xylem, and as a result shut their stomata frequently (Ryan and Yoder 1997)".

Soil moisture depletion is usually not limiting at the site because the mature trees are capable to tap water from deeper soil layers (Wharton et al. 2009). Our simulation results align with this observation, as the $\beta_t$ values were equal to 1 (no soil moisture stress) throughout most of the period of study (Fig. 6; note that we adjusted the root distribution in CLM – see our response to RC1.13). However, our results show that more severe SWC depletions can impose a reasonable limitation to canopy conductance under low VPD conditions (Fig. 7b). We will further discuss these points in the revised paper.

After a careful review we found that the words "drought stress" were used ambiguously in the text, in some instances meaning water stress in the broader sense (i.e., due to decreased SWC *and* increased VPD), and in some stances meaning soil moisture stress in particular. We will revise the text to improve clarity. Most importantly, we will change the title of Sect. 3.4 from "Ecosystem Drought Response" to "Ecosystem Response to Water Stress", and inside that section, we will change "drought stress" to "soil moisture stress". We will also expand the discussion on VPD-related water stress.

As pointed out by the Reviewer, we cycled meteorological forcing data from 1998 to 2006 in our transient simulation (1850–2006), so the impact of any slow secular changes in the forcing data (especially in precipitation, incident short(long)wave radiation, air temperature and relative humidity) was not captured in our simulation. We will mention this limitation in the revised paper. However, it is important to note that we did use 1850–2006 transient data for atmospheric $CO_2$, $\delta^{13}CO_2$, and nitrogen deposition.

As discussed above, we indeed focused on evaluating CLM's capability to capture the observed mean seasonal cycle of energy and carbon fluxes, marked by seasonal water stress periods during summer. Despite our focus being on the general performance of the model (across all study years), we intend to add discussion on inter-annual variability in the revised paper, as suggested by the Reviewer.

1. Consistency in water stress terminology throughout the text:  the expression 'drought stress' is now avoided; 'Water stress' is used in the context of VPD and SWC control on stomatal
5        conductance; 'soil moisture stress' is used in the context of SWC control on stomatal conductance.

2. Further discussion on canopy conductance dependency on VPD and SWC at Wind River (observed and modeled results):  included in Sect. 4.1.

3. Discussion on inter-annual variability:  included in Sect. 3.4, P15L9–18.

10    *2.* [RC2.1.2] *As I think more and more about the paper, the notion of equifinality (multiple solutions in parameter or process space that result in a single model outcome) keeps arising. A strength of CLM is that there is a large community of researchers banging on the model, and just about any process that one can imagine is included in the model and can be turned on or off (sunlit and shaded leaves; explicit nutrient cycles; hydraulic redistribution; multiple*
15      *hydrology schemes; diagnostic canopy vs. DGVM; etc., etc.). This is also a CLM weakness. There are so many knobs to turn, how can you really be sure you are turning the right ones? With this in mind, I returned to Raczka et al. (2016, hereafter R16), which simulated Niwot Ridge (NR1, another site with similar vegetation type, but dissimilar climate). Since Raczka is the second author on this MS, and Duarte is the second author on R16, these papers offer an opportunity*
20      *to present a body of work that compares and contrasts CLM behavior at two sites that have a similar dominant PFT but dissimilar climate (and soil, I presume). Imagine my surprise when I find that R16 claims that the nitrogen limitation scheme is a critical component, but is not considered in sensitivity tests in this MS! R16 is happy using CLM4.5 hydrology, while the current MS states that CLM4.0 hydrology was necessary to capture realistic behavior. I was*
25      *encouraged to see that both papers suggest a change from standard BB C3 slope of 9 to 6 in the evergreen needleleaf PFT, and to see that both papers see excessive discrimination of $^{13}$C. But I think there is an opportunity being missed here, to compare and contrast the results at two similar---but---not---identical sites. This is an opportunity to say something about how evergreen needleleaf forests behave, as informed by dissimilar CLM simulations. The first 2*
30      *authors are at the same institution, if not in the same department, and they need to be talking to each other. I think closer coordination/comparison/contrast of the results presented in R16 and here is required. Currently, I find myself more confused than anything. Can we trust CLM at all in evergreen needleleaf forest without extensive tuning from site---specific   observations?*

        **A:** We agree with the Reviewer that the modeling community should make an effort to
35      understand how applicable a parameterization at a single site is to other sites or regions. Although Niwot Ridge and Wind River fall into the same PFT category (needleleaf evergreen temperate tree) there are many local, atmospheric, topographical and physiological differences between the sites that put them at opposite sides of the PFT spectrum. These

include differences in age: Wind River is an old growth forest (~500 years old), and it has been shown that stand age effects (Wharton et al. 2009) can lead to a higher sensitivity to environmental conditions (stomatal response), whereas Niwot Ridge is a much younger, secondary growth forest, not subject to same effects. The site soil characteristics are also much different with a deeper soil layer (>1 m) at Wind River, allowing for deep penetration of roots, whereas Niwot Ridge has generally a shallow soil layer and root structure (< 1 m). Niwot Ridge is highly dependent on spring snowmelt for photosynthesis, and is completely dormant during the winter, whereas Wind River is in a milder climate, photosynthesizing during most times of the year and relying on deep root structure for soil moisture in the dry season.

The default parameters used in CLM are optimized for global simulations, so model performance at particular sites is expected to vary greatly, requiring site-specific calibration in order to adequately simulate the observed energy and carbon dynamics. This is demonstrated in the simulations at Wind River and Niwot Ridge. Soil hydraulic parameters from CLM 4.5, in particular, were found to perform well at Niwot Ridge (simulated and observed soil moisture gave reasonable agreement) and did not require adjustments. On the other hand, these parameters were found to be problematic at Wind River (soil moisture was much lower than observations, leading to excessive late-summer water stress and down-regulation of gross primary production), requiring adjustments (soil hydraulic parameters from CLM 4.0 brought soil moisture values in general agreement with observations).

We agree with the reviewer that a closer coordination of the results presented in Rackza et al. (2016) and here is needed, especially in the context of nitrogen limitation. We intend to add the following discussion in the revised paper.

Nitrogen limitation

As discussed in our response to RC1.4, CLM calculates potential GPP ($GPP_{pot}$) based on potential net leaf photosynthesis (potential $A_n$), which is not restricted by nitrogen availability. Leaf stomatal conductance, $g_s$, is calculated based on potential $A_n$ (Eq. 2):

$$g_s = m_{bb} \frac{A_n(\beta_t)}{c_s / P_{atm}} h_s + b_{bb} \beta_t$$

After the calculation of potential $A_n$ and $g_s$ (and $GPP_{pot}$), actual GPP is obtained by down-regulating $GPP_{pot}$ based on nitrogen availability (i.e., $GPP = GPP_{pot}(1 - d)$; see response to RC1.4). This down-regulation occurs in a decoupled fashion within the model, as the change in $GPP_{pot}$ (potential $A_n$) does not feedback into the calculation of $g_s$, i.e., stomatal conductance is consistent with the *potential* net leaf photosynthesis rather than the actual, nitrogen-limited net leaf photosynthesis. This makes CLM-CN a partially-coupled model in regards to $A_n$ and $g_s$ (Raczka et al. 2016).

As discussed in Raczka et al. (2016), the partial coupling between $A_n$ and $g_s$ causes issues in the calculation of photosynthetic $^{13}$C discrimination in CLM (Eqs. 4 and 5 in Duarte et al.). Note that in CLM's $c_i/c_a$ equation (Eq. 5),

$$\frac{c_i}{c_a} = 1 - \frac{A_n(1-d)}{c_a}\left[\frac{1.4}{g_b} + \frac{1.6}{g_s}\right],$$

$A_n$ (potential value) is down-regulated by the $(1-d)$ factor in order to make $c_i$ consistent with the actual, nitrogen limited GPP. However, $g_s$ is overestimated as it is consistent with potential $A_n$. When using this default nitrogen limitation scheme, the modeled $^{13}$C discrimination values reported by Raczka et al. (2016) for the Niwot Ridge site were significantly overestimated, i.e., δ$^{13}$C values of GPP and biomass significantly smaller than observations. To improve the simulation, Raczka et al. (2016)'s approach was to remove the posteriori nitrogen down-regulation of potential $A_n$ and GPP$_{pot}$ ($d=0$) and include a nitrogen-limiting factor directly in CLM's $V_{cmax25}$ equation, making the model fully coupled in respect to $A_n$ and $g_s$. With this configuration, their simulation of $^{13}$C discrimination improved significantly, but the values still presented a small overestimation in respect to the site observations. According to Raczka et al. (2016), an overestimation of $g_s$ due to inadequate parameter values in the Ball-Berry equation (e.g., too large slope value) could be a reason for the mismatch.

Here, we adopted a different approach. We used the default nitrogen limitation scheme in CLM, but also calibrated $g_s$ (via parameters $m_{bb}$ and $b_{bb}$ in Eq. 2) based on latent heat flux (LE) observations at the site. Mostly important, we found that a significant reduction of $m_{bb}$ from 9 (default) to 6 was necessary. This reduction, however, must also have compensated for the potential $A_n$ value used in Eq. 2.

In our simulation, we found an average nitrogen downregulation peak of $d=0.25$ during late spring (GPP/GPP$_{pot}$ = $(1-d)$ = 0.75). The first term on the r.h.s. of Eq. 2, with the calibration of $m_{bb}$, is equal to $6A_n$. We could interpret this term as $8(0.75A_n)$, where 8 is the value of $m_{bb}$ and $0.75A_n$ is an approximation of the actual net leaf photosynthesis.

Even though $A_n$ and $g_s$ were not fully coupled in our simulation, by calibrating $g_s$ we were able to significantly reduce the mismatch between actual photosynthesis and stomatal conductance and satisfactorily simulate the observed GPP, LE, and δ$^{13}$C values (leaf, SOM) at the site. We expect a fully coupled simulation using a nitrogen limitation downscaling factor of 0.75 directly in the $V_{cmax25}$ equation and $m_{bb}=8$ to lead to similar results. The downscaling factor is consistent with the value reported by Oleson et al. (2008) (0.72; Appendix G and Table G1 in their paper) for the needleleaf evergreen temperate tree PFT in CLM. The $m_{bb}$ parameter value (8) is smaller than the default in CLM (9).

When using the default nitrogen limitation scheme in CLM, a stronger reduction of $m_{bb}$, from 9 to 6, was necessary to simulate the carbon and energy dynamics at Wind River. This aligns with the results by Mao et al. (2016) for a loblolly pine stand in southeastern U.S. They were

able to simulate the site dynamics, including biomass $\delta^{13}$C values, with an optimized $m_{bb}$ of 5.6. As discussed in Appendix A9, CLM uses a default $m_{bb}$ value of 9 for all C3 plants, while the experimental literature indicates reasonably lower values for conifers. However, as discussed above, our results indicate that the significant reduction of $m_{bb}$ from 9 to 6 was mostly necessary to compensate for the potential $A_n$ value used in Eq. 2. The default $m_{bb}$ value of 9 seems more adequate for fully coupled simulations using the $V_{cmax25}$ downscaling approach.

In summary, our results indicate that it is possible to account for partial coupling in CLM through the adjustment of stomatal slope to achieve reasonable carbon and energy exchange behavior (including $^{13}$C discrimination). This is also supported by the results in Mao et al. (2016). The caveat of this approach is that $m_{bb}$ is adjusted beyond its intended range, compensating for structural error within the model. Overall, this approach is expected to lead to similar results in comparison with the $V_{cmax25}$ down-regulation/$m_{bb}$ calibration approach (fully-coupled CLM), but is still not a perfect solution for the partial coupling issue. The approach we used is sufficient for the purpose of our study, but we intend to run an additional simulation in "fully-coupled mode" to provide results for comparison in the revised paper.

The shortcomings of the current nitrogen limitation scheme in CLM are also discussed in Metcalfe et al. (2016). They propose a new scheme in which GPP is not down-regulated and the excess photosynthate is allocated to a new nonstructural carbohydrate storage pool within the model. Testing $^{13}$C discrimination with this new scheme is something we recommend for future studies.

**Manuscript revision**

**We now clearly address the nitrogen limitation issue in the revised manuscript, incorporating the above discussion in the text. Changes can be observed throughout the manuscript, especially in these Sections: Introduction; Methodology/Model Description (Sect. 2.1); Discussion/Calibration of CLM (Sect. 4.2); Discussion/Recommendations for Structural Improvement within CLM (Sect. 4.3); Appendix A9 (calibration of stomatal conductance).**

**Note: we tested the alternative nitrogen limitation scheme ($V_{cmax25}$ downscaling; Raczka et al., 2016) at Wind River and were able to verify the tradeoff between adjustment of stomatal slope and changes to the nitrogen limitation scheme, discussed above. However, the hypothesis that using $0.75V_{cmax25}$ and $m_{bb} = 8$ in a fully-coupled simulation would lead to similar results at Wind River could not be verified. Excessive productivity was found during the 1850–2006 transient simulation, indicating that a smaller $V_{cmax25}$ downscaling factor was necessary. We were able to get similar model results with the alternative nitrogen limitation scheme (in comparison with the default scheme and $m_{bb} = 6$) using $m_{bb} = 9$ and a seasonally-varying $V_{cmax25}$ downscaling factor calculated based on the mean (1850–2006) seasonal cycle of $\mathrm{GPP/GPP_{pot}} = 1 - d$ (from simulation using default nitrogen limitation scheme and $m_{bb} = 6$) subtracted by 0.35 (manual adjustment to avoid excessive productivity during the transient simulation). The results are included in the revised paper (Fig. A3).**

3. [RC2.1.3] *The authors question the veracity of the observations twice, with regard to H for 1998---2003, and SWC for 1998---2002. I urge extreme caution here, and sincerely hope the authors have corresponded with Dr. Wharton to express their concerns and make these statements with the understanding and approval of the site PI. This is the kind of situation that can foster distrust and animosity between the observational and modeling communities. I most strongly recommend that the authors verify that there may be uncertainty in these observational records, and the listing of Dr. Wharton as a coauthor would legitimize the claims as stated in the MS and confirm that she has participated in discussions of this issue.*

**A:** As mentioned in P9L11–15, our remarks on the sensible heat flux ($H$) observations are based on publicly available information posted on the AmeriFlux database. For clarity, we will add the full URL in the revised text: ftp://cdiac.ornl.gov/pub/ameriflux/data/Level4/Sites_ByName/Wind_River_Field_Station/history_changes.txt

Our remarks on the soil water content (SWC) data (P12L1–5 and P12L13–22) are based on discussions we carried out with Dr. Ken Bible, the site PI responsible for data management and who was directly involved with the installation of the ground instrumentation. Dr. Bible is a co-author in our paper.

**Manuscript revision**

**We noticed that the document cited above is no longer available online after migration of AmeriFlux to LBNL, so the web link was not included in the revised text.**

4. [RC2.1.4a] *Isotopes are difficult. They are difficult to explain, and difficult to understand for many (if not most) readers. The provenance of the treatment of carbon isotopes in CLM is poorly summarized in this MS. Oleson et al. (2013) does not mention them; is Mao (2016) the seminal paper? What about Randerson et al. (2002) or Suits et al. (2005) which investigated isotopes using SiB, or the work of Van der Velde et al. (2013, 2014) which studies isotopes in SiB--- CASA? Did isotopes in CLM build from that work, or from an independent source? R16 is cited, but elaboration is warranted, especially with regard to superposition of an annual cycle onto the larger trend in forcing data.*

**A:** Note that Oleson et al. (2013) does indeed describe the treatment of carbon isotopes in CLM 4.5 (see their Chapter 25, p. 391–397). We cite this technical report in P3L21. As stated in P4L6, the implementation of C3 photosynthetic $^{13}$C discrimination in CLM follows the model proposed by Farquhar and Richards (1984) and consists of Eqs. (4) and (5). CLM 4.5 does not include any representation of post-photosynthetic discrimination. The rest of the isotopic dynamics is described through the handling of the accounting and conservation of the isotopic pools, with the one exception of nighttime autotrophic respiration, as described in more detail below. The original implementation of $^{13}$C in CLM was done in consultation with Neil Suits. We will acknowledge this in the revised paper with the Reviewer's suggested citation (Suits et al. 2005).

With regards to the transient $\delta^{13}CO_2$ forcing data, as mentioned in P6L22–25, we used values based on ice-core and flask measurements reported by Francey et al. (1999) (annual values in their spline fitting from 1850 to 1981) and flask measurements in Mauna Loa (annual averages from 1981 to 2006) by the Scripps $CO_2$ program (Keeling et al., 2005), following a similar methodology as in Raczka et al. (2016). It is important to clarify that, unlike in Raczka et al. (2016), here we did not superimpose a seasonal cycle onto the time series. We will add this clarification in the revised paper.

**Manuscript revision**

**We acknowledged Suits et al. (2005) in P6L7 and added the clarification on the transient $\delta^{13}CO_2$ forcing data in P9L11.**

[RC2.1.4b] *The treatment of $\delta^{13}C_{ER}$ in models is extremely difficult to capture. Heterotrophic respiration is comprised of old, intermediate, and young components, and the $\delta^{13}C$ of each is difficult to constrain, as is the fractional contribution of each. The description of $\delta^{13}C_{ER}$ in section 3.3.1 is troubling; if I understand this section correctly, $\delta^{13}C_{ER}$ follows $\delta^{13}C_{GPP}$ in the daytime, then switches to follow $\delta^{13}C_{HR}$ at night. If the $\delta^{13}C$ of the $C_{XS}$ pool has no sensitivity to recent discrimination, then I assume the pool is large enough that the $\delta^{13}C$ of this pool reflects some previous state. Is this true? What is that state? Is it realistic? The MS states that this behavior "aligns with expectations", but is it realistic? Is this behavior observed at sites with more detailed observations? If this behavior is not observed anywhere, how can you trust the model results?*

**A:** As shown in Fig. 4, autotrophic respiration (AR) was the major component of total ecosystem respiration (ER), so $\delta^{13}C_{ER}$ exhibited a similar behavior compared to $\delta^{13}C_{AR}$. $\delta^{13}C_{AR}$ followed $\delta^{13}C_{GPP}$ during daytime as daytime maintenance respiration in CLM is fueled by newly assimilated carbon (supplemented by the $CS_{xs}$ pool if GPP is insufficient to meet the demand). During nighttime, $\delta^{13}C_{AR}$ followed values close to $\delta^{13}C_{HR}$ (heterotrophic respiration), with little sensitivity to recent $^{13}C$ discrimination.

In CLM, carbon from the $CS_{xs}$ pool is used to fuel the maintenance respiration at night (and other periods when GPP is insufficient to meet the maintenance respiration demand, e.g. winter). This is called "excess maintenance respiration", or XSMR, in the model. The nighttime $\delta^{13}C_{AR}$ therefore follows $\delta^{13}C_{XSMR}$. However, the isotopic signature of XSMR is not taken from $CS_{xs}$, but from the bulk vegetation tissues (total vegetation carbon, TOTVEGC). This is done because $CS_{xs}$ is not a physical quantity, but a construct of CLM. Note that XSMR "borrows" carbon from the $CS_{xs}$ pool, which is allowed to run a deficit state. This "debt" is paid in the future with the replenishment of the $CS_{xs}$ pool with newly assimilated carbon. This construct makes the $\delta^{13}C$ of $CS_{xs}$ non-physical, therefore, the approximation that $\delta^{13}C_{XSMR}=\delta^{13}C_{TOTVEGC}$ is more physically realistic.

In summary, the approximation done in the model ($\delta^{13}C_{XSMR}=\delta^{13}C_{TOTVEGC}$) makes the nocturnal $\delta^{13}C_{AR}$ to follow $\delta^{13}C_{TOTVEGC}$. This explains the low sensitivity of the nocturnal $\delta^{13}C_{AR}$ to recent

[13]C discrimination in our results (Fig. 4b). We will revise our explanation given in P10L12–16 with this information.

In P10L15 we mention that "the simulation results in Fig. 4b align with these expectations". Please note that the "expectations" which we refer to are related to the understanding of the carbon allocation structure in the model, not to observations. Observations at Wind River indicate that AR is likely fueled by a mixture of stored and recently-fixed carbon (P10L17), a process that cannot be appropriately modeled with the current carbon allocation scheme in CLM, given the lack of an explicit representation of carbohydrate storage pools to support maintenance respiration. In P10L17–22 we discuss this limitation in the model and highlight the need for a better representation of carbohydrate storage pools within CLM. In P11L20 we also point out that the implementation of carbon isotopes in CLM opens an interesting opportunity for future model development, as isotopes expose a limitation of the carbon allocation scheme. In future efforts, carbon isotope data can be used to guide a restructuring of the model, moving away from the deficit-based accounting scheme towards an explicit representation of carbohydrate storage pools.

**Manuscript revision**

**We revised the explanation of the $CS_{xs}$ pool and its implications to the isotopic signature of autotrophic respiration, as discussed above (P13L27–P14L3).**

[RC2.1.4c] *Similarly, the authors state that "Autotrophic respiration at Wind River is likely fueled by a mixture of stored and recently---fixed carbon, as indicated by $^{14}C$ measurements…(and) cannot be appropriately modeled by CLM with the current allocation scheme…" so if I understand correctly, we can't trust the $\delta^{13}C_{HR}$ because CLM doesn't consider different contributions from differently---aged dead pools, and we can't trust the $\delta^{13}C_{AR}$ either. Both $\delta^{13}C_{AR}$ and $\delta^{13}C_{HR}$ influence the $\delta^{13}C$ of the canopy air, which will in turn have a strong influence on $\delta^{13}C_{GPP}$! At this point I'm left thinking that we have no confidence in any of the discrimination values of this version of CLM, and any resemblance to observations is a happy accident.*

**A:** CLM does not consider different age classes of decomposing material. Instead, it treats different classes of litter and soil organic matter quality, with characteristic decomposition rates. It is possible to track a labeled pulse through these various pools because they are connected in a cascade structure, but it is not possible to isolate a specific age-class or cohort of same-aged decomposing material without rather elaborate "numerical labeling" experiments.

The main issue (or at least the most obvious one) is not with the representation of $\delta^{13}C_{HR}$ (heterotrophic respiration), but with the representation of $\delta^{13}C_{AR}$ (autotrophic respiration), for the reasons discussed in our response above. The isotopic signature of the total ecosystem respiration (ER=AR+HR) is obviously affected. However, in CLM, there is no feedback onto the canopy air space, because there is no prognostic canopy air space to feed back onto. This is

unlike the SiB model, for instance, which does have a prognostic canopy airspace. Therefore, in CLM, $\delta^{13}C_{GPP}$ is not affected. We will add this discussion in the revised paper.

**Manuscript revision**

**We added in the revised manuscript (P14L10–12): "It is important to highlight that, unlike models such as SiB (Sellers et al., 1996; Vidale and Stöckli, 2005), CLM does not have a prognostic canopy airspace where $\delta^{13}CO_2$ is impacted by photosynthetic and respiratory fluxes, so the simulation of $\delta^{13}C_{GPP}$ is not affected by the above described limitations in the simulation of $\delta^{13}C_{ER}$."**

[RC2.2] *I do not recommend rejection, but this paper requires major revisions to be acceptable for publication. The differences between the findings at Wind River and those of R16 must be reconciled, and the problems with the isotope treatment must be resolved. Furthermore, the characterization of climatological dry summer as 'drought' is unacceptable.*

**A:** We addressed these topics in our responses above.

*Specific Comments*

- [RC2.3] *The plant wilting factor, $w_i$, can be expressed in a multitude of ways, and could have serious implications for this site. Entekhabi and Eagleson (1989) suggest a linear reduction in $w_i$ from some point s\* below field capacity (where $w_i$ = 1) to a value of 0 at wilt point, while Sellers et al (1996) and Colello et al (1998) promote a nonlinear equation for $w_i$ based on field data from FIFE. Baker et al (2008) demonstrate that in tropical forests, a direct linkage of $w_i$ to the vertical profile of root density can be problematic. What form does $w_i$ take in CLM? Neither R16 nor this manuscript discuss this; was it investigated? This suggests yet another path that can be taken to tune CLM. More equifinality.*

  **A:** In our answer to RC1.3 we describe how the plant wilting factor ($w_i$) is calculated in CLM. We agree that $w_i$ offers a path for adjustment of the soil moisture stress factor ($\beta_t$). In our study we opted to adjust the root distribution ($r_i$) instead, shifting it to deeper soil layers (a physically justified approach – see further discussion in our response to RC1.13). We have not investigated the $w_i$ factor in our study, but will include its formulation and point out that it offers an additional path that can be taken to adjust $\beta_t$.

**Manuscript revision**

**We added the equation for $w_i$ in the revised manuscript (Eq. 5). We also included: "The plant wilting factor, $w_i$, offers an additional path for adjustment of the simulated soil moisture stress, but it was not investigated in this study." (P25L30–31).**

- [RC2.4] *Discussing equation 5, the authors state that since $g_b \gg g_s$, $g_b$ can be neglected in the calculation of ci/ca, and therefore discrimination. In midday this is certainly true, but what about near sunrise/sunset, particularly sunrise? I can imagine that immediately after sunrise, the canopy is cool and the nocturnal inversion has not yet been broken. The canopy, however,*

*is illuminated, and both temperature and humidity are favorable for stomatal conductance. In this situation I might expect that $g_s$ could be larger than $g_b$. This might also have some bearing on the large excursions in $\delta^{13}C_{GPP}$ values seen early and late in the day. Was this investigated?*

**A:** We would like to clarify that Eq. (5) is the actual equation used in the model, and Eq. (6) is a simplification we presented for discussion purposes. We did check the modeled values of leaf boundary-layer conductance ($g_b$) against the modeled values of leaf stomatal conductance ($g_s$). We found $g_b$ to be much larger than $g_s$, even around sunrise and sunset.

- **[RC2.5]** *Line 31, page 6; should be 'resulting', not 'resulted'*

  **A:** We will make the correction in the text.

**Manuscript revision**

**Correction made.**

- **[RC2.6]** *Why is the observed $\delta^{13}C$ of bottom canopy leaves so much lower than elsewhere in the canopy? Does this inform the CLM treatment of isotopes?*

  **A:** Please see our answer to RC1.10.

- **[RC2.7]** *ET will be composed of transpiration, leaf evaporation and ground evaporation components. In a dense canopy like Wind River, I would expect ground evaporation to be low, but excessive leaf evaporation could influence the amount of infiltration and therefore the amount of water available for transpiration later in the season. What is the partition of these components at Wind River? I know it is impossible to quantify these components with a single ET observation, but the model partition may give insight into behavior.*

  **A:** Overall, the simulated ground evaporation (FGEV) was negligible in respect to the simulated canopy transpiration (FCTR) and canopy evaporation (FCEV) (Fig. R3). FCTR was very low during winter and peaked during summer. On the other hand, FCEV was a major component of ET during the wet season (~October–May) but typically presented very small values during the dry season in summer. The results in Fig. R3 (year 2005) illustrate the overall pattern.

  In this study we were able to reasonably simulate the observed soil water content (SWC) values during summer with the adjustment of the soil hydraulic parameters (see P11L30 and Sect. A7). Perhaps the dry bias we found in SWC during summer when using the original soil hydraulic parameters could be in part related to excessive interception of precipitation by the canopy (and reduced infiltration), as pointed out by the reviewer. The observations we have do not allow us to evaluate if the simulated FCEV values were excessive, though. A sensitivity test to see how the simulated summer SWC values are impacted by the modeling of FCEV and eventual adjustments using this path are out of scope here, but are suggested for future studies.

[Figure]

Fig. R3: Partitioning of evapotranspiration at Wind River, as modeled by CLM. FCTR is canopy transpiration, FCEV is canopy evaporation, and FGEV is ground evaporation. Lines correspond to 1-hourly data, smoothed via a Bézier algorithm, for year 2005.

- [RC2.8] *The authors mention a "spring‑‑‑to‑‑‑summer decrease in the contribution of root respiration towards total soil respiration". That makes me think that the observed signal could come about from one of 2 ways. 1) there is a decrease in discrimination as WUE increases, or 2) there is an increase in the HR component towards older material with a lower $\delta^{13}C$. Is this possible?*

**A:** Assuming the Reviewer is referring to the observed $\delta^{13}C_{ER}$ (ecosystem respiration) signal and meant "higher $\delta^{13}C$" in explanation 2 ($^{13}C:^{12}C$ ratio is higher for older carbon given the Suess effect), that interpretation is correct. In fact, we already discuss this in the paper (P10L31–P11L3):

"The seasonal pattern in the observed $\delta^{13}C_{ER}$ (Fig. 5) could be partially attributed to an eventual spring-to-summer decrease in AR:ER ratio (assuming $\delta^{13}C_{HR} > \delta^{13}C_{AR}$). $^{14}C$ measurements from below-ground respiration components at Wind River reported by Taylor et al. (2015) do indicate a spring-to-summer decrease in the contribution of root respiration (RR) towards total soil respiration (SR=RR+HR). The similarity of the seasonal patterns of observed $\delta^{13}C_{ER}$ and modeled $\delta^{13}C_{GPP}$ suggests that stomatal response to drought[#] could also be driving the seasonal pattern in the observed $\delta^{13}C_{ER}$ at the site".

Note that a decrease in AR:ER is equivalent to say an increase in HR:ER. Note also that by "stomatal response to drought[#]" we mean a decrease in stomatal conductance leading to higher water use efficiency and lower photosynthetic $^{13}C$ discrimination.

[#]Will be reworded to "water stress" in the revised paper, avoiding confusion with meteorological drought (precipitation below climatological normal). See our response to RC2.1.1.

- [RC2.9] *When discussing soil moisture, fraction of saturation may be more useful than volumetric water content. The volumetric content at various important points (wilt point, field capacity, saturation) can vary significantly depending on soil character.*

    **A:** At Wind River, volumetric soil water content (SWC) at permanent wilting point is 14%, and SWC at field capacity is 30% (Wharton et al. 2009). We will include this information in the discussion.

**Manuscript revision**

**Information included in P15L23–24 and in the caption of Fig. 6.**

- [RC2.10] *Page 13, lines 14---17: "Observed $\delta^{13}C_{ER}$ was found to have a low negative correlation with observed $G_c$, but not statistically significant. The low correlation was likely a result of $\delta^{13}C_{ER}$ reflecting constraints of prior environmental drivers in comparison with the more rapid response of $G_c$ to more recent environmental drivers." This is not surprising, but doesn't it underscore the importance of getting a handle on all these drivers, in both your observations and model? Aren't you basically saying here that the model results are not to be trusted because the proper mechanistic pathways for the various isotopes are not simulated?*

    **A:** Indeed, the simulated $\delta^{13}C_{ER}$ (ecosystem respiration) values are impacted by the lack of an explicit representation of carbohydrate storage pools within CLM to support the maintenance respiration demand (see our answer to RC2.1.4). Note that the $\delta^{13}C_{ER}$ observations correspond to nocturnal measurements. During nighttime, our modeled $\delta^{13}C_{ER}$ values were found to be virtually constant throughout the entire study period (1998—2006), with values similar to $\delta^{13}C_{HR}$ (heterotrophic respiration) (Fig. 4b). During daytime, modeled $\delta^{13}C_{ER}$ was similar to $\delta^{13}C_{GPP}$ (gross primary production). Note that in Fig. 9b we plotted modeled $\delta^{13}C_{GPP}$, not modeled $\delta^{13}C_{ER}$, versus modeled canopy conductance ($G_c$) in order to have a more informative result for comparison against Fig. 9a (observed $\delta^{13}C_{ER}$ vs. observed $G_c$; see discussion in P13L12–21).

**Comments from Associate Editor**

*Dear Authors*
*Thank you for your very careful response to the reviews of the first version of the manuscript. At this stage I would recommend to go ahead with providing a revision that carefully addresses the many critical points the referees have raised.*

*When you work on the revision please keep specifically in mind:*
*1. That trial and error model tuning is never superior to Bayesian calibration. In accepting that calibrating the model properly was not possible, you accept that the information to constrain the complex model is not sufficient and, thus, the information on the generated parameters meaningless, i.e. one of potentially many possible solutions with no information on the parameter*

*uncertainty and correlation with other parameters. In your response you seem to neglect this deficiency when writing: "The calibrated parameters we present in the paper may be of use for future modelling studies". How can parameters that have been generated in a trial and error fashion be acceptable for the modelling community?*

5     **A:** We no longer include those statements. In the revised manuscript we focus on the adjustment of the Ball-Berry stomatal conductance slope at Wind River and compare our results against those from other recent CLM studies at sites characterized by the same plant functional type as Wind River (needleleaf evergreen temperate tree), but significant differences in stand composition/age and climatology. The comparison shows an agreement across the sites, suggesting that CLM could
10     benefit from a revised (lower) stomatal slope for that PFT.

*2. A very important concern is that both reviewers have expressed their impression that you didn't motivate the study well enough. When you prepare the revision please make sure that the objectives of your study can be easily understood and are well reasoned. Clarification of the objectives and the achieved scientific progress will be an essential requirement for the acceptance*
15     *of the manuscript. A statement 'represent important contributions to the modeling community' is not explicit enough.*

    **A:** We carefully followed the recommendation from Reviewer 1:

    1. Introduction was significantly revised. Additional citations were included, helping to more clearly lay out the motivation of our study.

20     2. Discussion of results was expanded. A separate "Discussion" section was created (Sect. 4), where we focus on the key findings of our study.

    Abstract and Conclusion were revised accordingly.

    *With kind regards,*
    *Andreas Ibrom*

[revised manuscript text omitted]

Comment [HFD41]: [RC1.11]

---

## Author Response (AR2)

**Authors' Response to Reviewers**

Evaluating the Community Land Model (CLM 4.5) at a Coniferous Forest Site in Northwestern United States Using Flux and Carbon-Isotope Measurements

Henrique F. Duarte, Brett M. Raczka, Daniel M. Ricciuto, John C. Lin, Charles D. Koven, Peter E.
5   Thornton, David R. Bowling, Chun-Ta Lai, Kenneth J. Bible, and James R. Ehleringer

**Report #1, Referee #2**

*The authors have addressed the comments and suggestions of both reviewers in their revised*
10   *manuscript to my satisfaction. I therefore recommend publication with a few minor technical*
*revisions. They are*

*1) P8L6: Should the american 'unfavorable' be used here?*

*2) P17L24: I think the figure cited should be 7a*

*3) P19L17: "similar"*

15   A: Text corrected as suggested, except for item 2. The citation to Fig. 7b is correct.

**Report #2, Referee #3**

*I'm glad that the authors carefully discussed the fact that the effect of the change in the Ball-Berry*
*slope parameter could also have been effected with a change to a fully coupled stomatal model. It*
*does leave one wondering, however, whether the change in the BB parameter is really needed or*
20   *not. One thing I would really like to see is the value of "d" ie the nitrogen limitation factor. Is it*
*possible to make a graph showing "d" at the Wind River site? This would be helpful to understand*
*whether the need to change the BB parameter arises from the nutrient limitation, or the fact that*
*the stomatal slope parameter really is lower at this site.*

A: For clarification, we added a figure (new Fig. A3) showing the mean seasonal cycle of
25   $GPP/GPP_{pot} = 1-d$ at Wind River (calibrated CLM simulation).

Please note in Appendix A9, P26L17–24:

"It is important to highlight that the default nitrogen limitation scheme was used in the simulations. As discussed in Sect. 2.1, this scheme makes CLM a partially-coupled model in respect to net leaf photosynthesis and stomatal conductance: while the actual GPP is down-regulated in
30   response to nitrogen availability, stomatal conductance remains consistent with potential net leaf photosynthesis ($A_n$). With this structure, CLM is expected to overestimate plant transpiration and photosynthetic $^{13}$C discrimination. The above discussed calibration of the Ball-Berry stomatal conductance parameters, especially the significant reduction of $m_{bb}$ from 9 to 6, must also have compensated for this structural issue within the model." We added at the end of this paragraph

the following text: "Note that nitrogen down-regulation is significant at Wind River, peaking at ~0.25 (GPP/GPP$_{pot}$ = 0.75) in May (Fig. A3)."

Please also note in Appendix A9, P27L1–5:

"The alternative nitrogen limitation scheme (via $V_{cmax25}$ down-regulation, as in Raczka et al., 2016) was also investigated here. The simulation of LE, GPP, and [13]C discrimination when using this configuration and the default $m_{bb}$ value of 9 was found to be similar to the results when using the default nitrogen limitation scheme and $m_{bb} = 6$ (Fig. A4). The results in Fig. A4 indicate that the calibration of $m_{bb}$ from 9 to 6 represents a tradeoff with the approach to nutrient limitation, compensating for elevated, nitrogen-unlimited (potential) net leaf photosynthesis used in the calculation of $g_s$."

Additional changes to the text (very minor edits):
P4L3, P20L5, P27L12–13, P27L18–19, P48L6–8 (see annotated document)

[revised manuscript text omitted]
 = $ day of year$) = -1.39697 \times 10^{-14} x^6 + 1.71948 \times 10^{-11} x^5 - 8.26883 \times 10^{-9} x^4 + 1.90682 \times 10^{-6} x^3 - 1.97639 \times 10^{-4} x^2 + 0.0055728x + 0.966272$ for $31 \le x \le 332$ and $f_n(x) = 1$ elsewhere. Note that GPP/GPP$_{pot} = 1-d$, where $d$ is the nitrogen down-regulation factor as defined in Eq. (9) within text.

[Figure]

**Figure A43.** Modeled gross primary production (A), latent heat flux (B), and $\delta^{13}C$ of gross primary production (C) for 1998–2006. The curves presented correspond to Bézier-smoothed daily averages as in Figs. 3 and 5. Results from two model runs are presented: CLM (calibrated model, red lines), and CLM# (calibrated model using $m_{bb} = 9$ and the alternative nitrogen limitation scheme discussed in Appendix A9, black lines). In CLM#, $V_{cmax25}$ was multiplied by a seasonally-varying nitrogen down-regulation factor, $ calculated based on the mean (1850–2006) seasonal cycle of $GPP/GPP_{pot} = 1 - d$ in the CLM run $(f_n(x)$ in Fig. A3) subtracted by 0.35 (manual adjustment applied to avoid excessive productivity during the transient simulation). Blue lines and circles correspond to site observations. The circles in panel C are the monthly averages of $\delta^{13}C_{ER}$ reported by Lai et al. (2005).

[Figure]

**Figure A54.** Comparison of CLM performance at Wind River when using default, "out-of-the-box" parameters (black lines) and calibrated parameters (red lines). Modeled values correspond to annual averages. Observations (average ± std. dev., blue points and error bars) are from the AmeriFlux database (based on Thomas and Winner, 2000 and Harmon et al., 2004).

[Figure]

**Figure A65.** Comparison of CLM performance at Wind River when using default, "out-of-the-box" parameters (black lines) and calibrated parameters (red lines). Observations (blue lines) are from the AmeriFlux database. For a clearer visualization, the data presented correspond to Bézier-smoothed daily averages as in Fig. 3.